# Fixing It in Post: A Comparative Study of LLM Post-Training Data Quality and Model Performance

**Aladin Djuhera**
Technical University Munich
aladin.djuhera@tum.de

**Swanand Ravindra Kadhe**
IBM Research
swanand.kadhe@ibm.com

**Syed Zawad**
IBM Research
szawad@ibm.com

**Farhan Ahmed**
IBM Research
farhan.ahmed@ibm.com

**Heiko Ludwig**
IBM Research
hludwig@ibm.com

**Holger Boche**
Technical University Munich
boche@tum.de

## Abstract

Recent work on large language models (LLMs) has increasingly focused on post-training and alignment with datasets curated to enhance instruction following, world knowledge, and specialized skills. However, most post-training datasets used in leading open- and closed-source LLMs remain inaccessible to the public, with limited information about their construction process. This lack of transparency has motivated the recent development of open-source post-training corpora. While training on these open alternatives can yield performance comparable to that of leading models, systematic comparisons remain challenging due to the significant computational cost of conducting them rigorously at scale, and are therefore largely absent. As a result, it remains unclear how specific samples, task types, or curation strategies influence downstream performance when assessing data quality. In this work, we conduct the first comprehensive side-by-side analysis of two prominent open post-training datasets: Tulu-3-SFT-Mix and SmolTalk. Using the Magpie framework, we annotate each sample with detailed quality metrics, including turn structure (single-turn vs. multi-turn), task category, input quality, and response quality, and we derive statistics that reveal structural and qualitative similarities and differences between the two datasets. Based on these insights, we design a principled curation recipe that produces a new data mixture, **TuluTalk**, which contains 14% fewer samples than either source dataset while matching or exceeding their performance on key benchmarks. Our findings offer actionable insights for constructing more effective post-training datasets that improve model performance within practical resource limits. To support future research, we publicly release both the annotated source datasets and our curated TuluTalk mixture.

## 1 Introduction

As large language models (LLMs) models continue to grow in complexity, so do their training requirements, necessitating ever-larger datasets with each new model iteration [1–3]. While pretraining LLMs on large, general-purpose corpora is now well understood [1–6], recent work has shifted toward *post-training*, which includes supervised fine-tuning (SFT), reinforcement learning (RL), and task-specific fine-tuning such as domain adaptation [7–10].

Carefully curated *post-training datasets* play a critical role in ensuring high downstream task performance, instruction following, and advanced reasoning. Nevertheless, the majority of post-training corpora remain proprietary, restricted by commercial licensing or intellectual-property concerns, and are therefore unavailable for public scrutiny and reuse. This has motivated state-of-the-art research

39th Conference on Neural Information Processing Systems (NeurIPS 2025) Track on Datasets and Benchmarks.

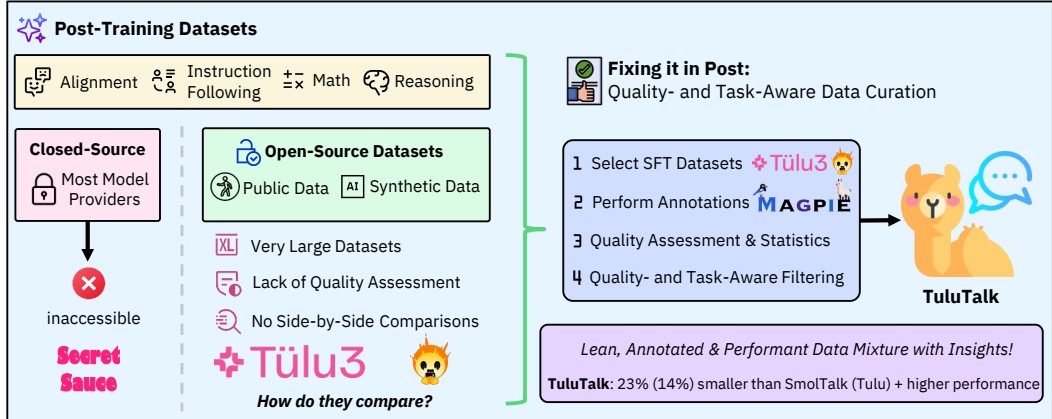

Figure 1: More effective post-training datasets through quality- and task-aware curation. We annotate and filter open-source SFT datasets (Tulu, SmolTalk) using Magpie to create TuluTalk, a leaner SFT data mixture (-23% vs. SmolTalk and -14% vs. Tulu) with improved benchmark performance.

on synthetic data generation, the development of open-source large-scale post-training datasets, and the design of effective post-training recipes [7, 11–16].

Yet, a major barrier to fully leveraging these datasets lies in the lack of systematic comparisons between them. Further, current literature employs a wide variety of model architectures, training hyperparameters, and data mixtures, resulting in considerable methodological heterogeneity across studies. Without a standardized frame of reference, it remains unclear which post-training datasets provide substantial benefits, and in what specific contexts. This lack of clarity hampers progress by obscuring the optimal direction for future research.

Another challenge with these datasets is the lack of transparent documentation regarding their curation processes. General steps are typically addressed briefly and critical details, particularly those concerning the creation of dataset mixtures, are often vaguely described. While recent works [7, 13] take giant strides in enhancing the transparency of post-training datasets and recipes, obscurity still remains for several crucial aspects. For instance, there is often a lack of details on which exact ablations were conducted to design mixture ratios, rendering the approach neither fully replicable nor sufficiently insightful to guide future dataset curation efforts.

Thus, in this paper, we adopt a principled and openly insightful approach to developing and evaluating post-training datasets, an effort that, to the best of our knowledge, is the first of its kind. To this end, we focus on two of the largest openly available SFT mixtures[1] from recent works: Tulu-3-SFT-Mix [7] (referred to as Tulu) and SmolTalk [9]. Our key contributions (see Fig. 1) are as follows:

- **Performance Evaluation**: We conduct the first side-by-side comparison of recent open-source SFT data mixtures: Tulu [7], SmolTalk [13], and Orca-AgentInstruct [16]. By fixing the model architecture and training hyperparameters, we enable a clean comparison of dataset performance across 14 LLM benchmarks, including those from popular Open LLM Leaderboards [17, 18]. We identify key differences and performance gaps for specialized skills such as coding, math, and instruction following, illuminating strengths and weaknesses of each mixture.

- **Quality Annotations**: To systematically drive data mixture decisions, we require detailed and standardized annotations of samples, a practice currently uncommon among the open-source post-training community. To this end, we leverage the Magpie framework [14] and annotate each Tulu and SmolTalk data sample along multiple dimensions, including conversational structure, prompt and response quality, and task categorization. These annotations provide concrete insights into dataset composition and support informed decision making for performant data mixtures.

- **Quality-Based and Task-Aware Data Curation**: Leveraging our extensive annotations, we design a simple yet principled curation recipe that selects high-quality and task-diverse samples from Tulu

---

[1]While post-training typically includes both SFT and RL, we restrict our attention to SFT in this paper. See App. A for more details.

and SmolTalk. The resulting mixture, **TuluTalk**, contains *14% fewer samples than Tulu* and *23% fewer samples than SmolTalk*, offering a leaner post-training corpus while achieving *comparable or better performance* on key benchmarks.

Our analysis provides actionable insights for curating effective post-training datasets from open-source corpora. We publicly release our annotation code[2], the corresponding annotated versions of Tulu[3] and SmolTalk[4], as well as our curated data mixture **TuluTalk**[5], to facilitate further research and enable reproducible studies on LLM post-training and data quality.

## 2   Background and Motivation

To the best of our knowledge, a direct side-by-side dissection of two flagship post-training datasets such as Tulu and SmolTalk has not been previously conducted, primarily due to significant compute requirements, particularly for large-scale SFT and extensive data annotations. We specifically select Tulu and SmolTalk (see App. B for more details on dataset compositions) due to their widespread adoption and the strong empirical performance demonstrated by their respective post-trained LLMs in recent benchmarks [7, 9].

**Tulu.** Lambert et al. [7] designed Tulu to advance broad-spectrum reasoning in medium-sized Llama models. They begin by filtering existing instruction corpora for (i) diverse real-user requests (e.g., WildChat [8], OpenAssistant [19]), and (ii) specialized skills (e.g., OpenMath-Instruct [10]). Residual gaps in *instruction following*, *math*, *coding*, and *safety* are filled with GPT-4o [20] generated prompts produced through persona-based prompting [21]. After n-gram decontamination and heuristic quality filtering, the final release contains 0.94 million high-quality pairs spanning seven broad domains (knowledge, math, reasoning, coding, safety, instruction following, and multilingual).

**SmolTalk.** Allal et al. [9] pursue a complementary objective to Tulu by building *small* models that deliver rich, multi-turn conversations without requiring large compute budgets. Their SFT mixture therefore focuses on *conversational depth* and *pragmatic rewriting*, and comprises roughly 1.04 million examples. In their curation process, they first mix Magpie-Ultra [14] which provides high-quality multi-turn prompts generated via an enhanced two-step Magpie procedure on a stronger teacher model. Second, three synthetic task-oriented subsets (Smol-Constraint, Smol-Summarization, and Smol-Rewrite [9]) are produced with targeted system prompts on Qwen2.5-72B-Instruct [22]. Third, equation-heavy math corpora [23, 24] are added. Finally, they add code-alignment and long-context resources [12, 25, 26] to the mixture. All subsets undergo a similar deduplication, quality filtering, and n-gram decontamination as in Tulu.

In addition to Tulu and SmolTalk, we also acknowledge earlier, similar-sized influential post-training mixtures such as **Orca** [15], which similarly covers tasks ranging from creative writing and text editing to coding and reading comprehension. However, initial results indicate that the more recent Tulu and SmolTalk datasets consistently outperform Orca across all evaluated tasks. Thus, we omit further investigation into Orca.

We present the corresponding SFT results in Table 1, where we fine-tune Llama-3.1-8B [27] and SmolLM2-1.7B [9] on Tulu, SmolTalk, and Orca, respectively, and evaluate both models on popular OpenLLM Leaderboard benchmarks (see App. E for the detailed fine-tuning and evaluation setup). We select these two models because they were used by Lambert et al. [7] to train the Llama-3.1-Tulu-3-8B model on the Tulu dataset and by Allal et al. [9] to train the SmolLM2-1.7B-Instruct model on SmolTalk. This choice allows us to validate our pipelines and ensure parity with prior work.

We evaluate performance across 12 tasks drawn from Open LLM Leaderboard V1 and Leaderboard V2, as well as two code generation tasks (HumanEval and HumanEval+), using the LM Evaluation Harness framework [28]. We report the average scores for Leaderboard V1 and Leaderboard V2 along with the overall average score across all 14 benchmarks. For Llama, SFT on SmolTalk outperforms Tulu on both LLM leaderboard benchmarks, however, falls behind in code benchmarks, where Tulu slightly pulls ahead on both benchmarks. Overall, fine-tuning with either dataset yields substantial

---

[2]Annotation code available at: github.com/aladinD/Magpie-single-and-multi-turn

[3]Annotated Tulu dataset: huggingface.co/datasets/aladinDJ/tulu-3-sft-mix-annotated

[4]Annotated SmolTalk dataset: huggingface.co/datasets/aladinDJ/smoltalk-annotated

[5]Annotated TuluTalk dataset: huggingface.co/datasets/aladinDJ/tulutalk-annotated

Table 1: SFT results for Llama-3.1-8B and SmolLM2-1.7B base models fine-tuned on Tulu, SmolTalk, and Orca, and evaluated on the Open LLM Leaderboards (averaged) and code benchmarks. The overall average is across all benchmarks. Best scores (row-wise) are in **bold**.

| | Llama-3.1-8B | | | | SmolLM2-1.7B | | | |
|---|---|---|---|---|---|---|---|---|
| Benchmark | Base | Tulu | SmolTalk | Orca | Base | Tulu | SmolTalk | Orca |
| *Leaderboards* | | | | | | | | |
| Open LLM Leaderboard 1 | 58.98 | 62.63 | **65.19** | 60.03 | 48.29 | 50.77 | **51.82** | 47.78 |
| Open LLM Leaderboard 2 | 27.84 | 37.47 | **38.24** | 36.05 | 24.14 | **30.66** | 30.39 | 27.67 |
| *Code* | | | | | | | | |
| HumanEval (pass@1) | 34.76 | **58.54** | 54.51 | 51.37 | 0.61 | **1.83** | **1.83** | 0.61 |
| HumanEval+ (pass@1) | 28.66 | **45.37** | 44.27 | 40.29 | 0.61 | **1.83** | **1.83** | 0.61 |
| *Overall* | 41.74 | 50.32 | **51.38** | 47.72 | 31.13 | 35.16 | **35.49** | 32.42 |

improvements compared to the baseline. For SmolLM, both SmolTalk and Tulu achieve similar performance on the OpenLLM Leaderboard benchmarks. However, performance is equally low on both code benchmarks, reflecting SmolLM's smaller size and its design focus on conversational rather than coding tasks. As noted earlier, both Tulu and SmolTalk consistently outperform Orca across all benchmarks, reflecting stronger data curation and higher corresponding task coverage.

These results prompt several initial research questions: Considering their distinct dataset compositions, what is the precise impact of SmolTalk's conversation-centric approach on fact-based benchmarks, such as math, reasoning, and code? To what extent do multi-turn conversations influence performance in these specific task categories? Lastly, how can we optimally combine Tulu and SmolTalk into a dataset mixture that enhances performance both in coding tasks and across benchmarks more broadly?

To address these questions, we conduct a detailed analysis of both post-training datasets in the following section, allowing us to make informed decisions about effective dataset combinations.

## 3 Quality Analysis of Tulu and SmolTalk via Magpie Annotations

We perform a unified diagnostic of the Tulu and SmolTalk datasets. Using the Magpie framework, a customizable self-synthesis annotation pipeline that leverages an LLM as a judge and specialized prompt templates, each data sample is systematically labeled for task category, conversation depth, instruction quality, response reward, and safety. These fine-grained annotations enable us to quantify both instruction fidelity and response adequacy, revealing precisely where these flagship corpora overlap, diverge, and, crucially, complement each other. Furthermore, this detailed characterization provides a principled basis for informed decision making regarding optimal dataset mixtures.

### 3.1 Unified Magpie Annotations

To enable direct comparability between Tulu and SmolTalk, we annotate (*tag*) every data sample using Magpie with Llama-3.3-70B-Instruct [27] as the judge model (see App. C.1). We find that Llama-generated annotations are reliable and align with human judgment (see App. C.1.3 and App. C.1.4). Magpie annotates each sample with structured tags for *Task Category* (12 classes), *Input Quality* (rated from very poor to excellent), *Response Quality* (termed *Instruct Reward*, rated from 0 to 5 for multi-turn and as a real number for single-turn), *Safety* (assessed via Llama-Guard 2 [29]), *Language*, and query *Difficulty*. We further extend Magpie's original annotation set by explicitly capturing the conversation structure (single-turn vs. multi-turn) and retain important metadata from the original datasets (e.g., unique sample identifiers), making our annotated versions reusable for future research.

In addition, we introduce two essential extensions to Magpie (see App. C.1.2): (1) To account for inconsistent or free-form outputs, we employ an error-tolerant JSON parser and include in-context examples in each prompt, resulting in up to 15% lower post-processing errors. (2) Further, as Magpie originally evaluates only the initial user-assistant interaction, we adapt its prompts to ingest entire conversation histories for multi-turn dialogues and utilize a larger context window to prevent truncation issues and tagging failures for longer conversations. With these adaptations, we limit the annotation failure rate, i.e., samples that could not be parsed or tagged due to inconsistent formatting or residual errors, to below 3%, ensuring that at least 97% of the original dataset is reliably tagged.

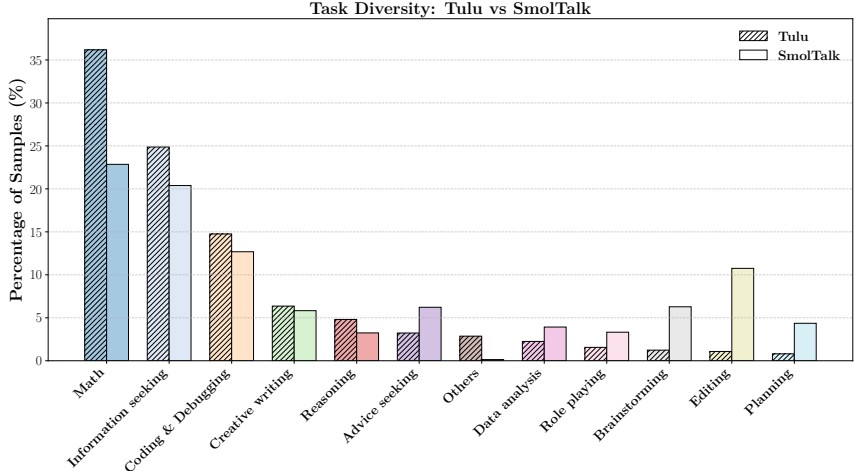

Figure 2: Task diversity in Tulu and SmolTalk as annotated by Magpie. Bars show the fraction of each dataset devoted to different tasks (e.g., math, coding/debugging). Tulu is dominated by structured, code-centric, and mathematical prompts, whereas SmolTalk features a substantial amount of conversational tasks such as editing, information seeking, and brainstorming, alongside math.

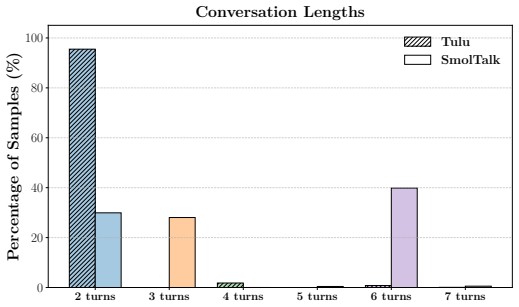

(a) Distribution of conversation lengths: Tulu is predominantly single-turn structured (95%), whereas SmolTalk is mostly multi-turn (70%).

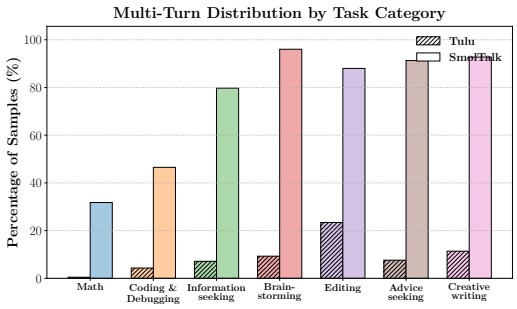

(b) Distribution of multi-turn samples: SmolTalk emphasizes rich multi-turn interactions for editing, creative writing, brainstorming, and advice seeking tasks.

Figure 3: Analysis of conversational turn structure: (a) Distribution of conversation lengths (single-turn vs. multi-turn). (b) Breakdown of multi-turn samples by task category.

## 3.2 Task Categories and Turn Structure

Our task annotations in Fig. 2 reveal clear contrasts between Tulu and SmolTalk. In particular, Tulu demonstrates a strong STEM-oriented bias. Over one-third (36%) of its samples focus on math, a quarter (25%) addresses information seeking (e.g., scientific fact checking), and coding covers 15%. In contrast, conversational and creative tasks, such as editing, creative writing, brainstorming, and advice seeking, are notably underrepresented, collectively accounting for only about 10% of the data. This composition directly aligns with Tulu's primary design objective of maximizing instruction following and structured reasoning, particularly in math and code.

Conversely, SmolTalk exhibits a more conversation-centric distribution. Editing, creative writing, brainstorming, and advice seeking constitute around 30% of the dataset, significantly more than in Tulu. Although SmolTalk maintains substantial math (23%) and coding (13%) segments, its overall emphasis clearly lies in open-domain interactions, aligning with its goal of training conversationally fluent yet compact ("*smol*") chat models.

These differences are also reflected in the conversational turn structures shown in Fig. 3a. Tulu is predominantly single-turn (95%), while SmolTalk primarily comprises multi-turn interactions (70%). A corresponding breakdown of multi-turn samples by task category is provided in Fig. 3b. Notably, Tulu almost entirely lacks multi-turn math samples and contains only a small fraction of multi-turn

coding and information seeking samples. In contrast, SmolTalk contains multi-turn samples even for math and coding (e.g., iterative rewriting of formulas and follow-up questions), mostly sourced from Magpie-Ultra which contains 3-turn samples for coding, math, and creative tasks. We provide a more detailed analysis of turn types and conversation lengths by task category in App. C.3.

These annotation insights show that the two datasets occupy complementary regions of the instruction space: Tulu specializes in rigorous, structured problem-solving tasks, whereas SmolTalk broadens coverage through richer, more interactive conversational samples.

### 3.3 Input Quality and Instruction Reward

Fig. 4 shows the distribution of input-quality annotations for both Tulu and SmolTalk. Overall, both datasets exhibit high-quality user inputs, with more than 80% rated as either *"good"* or *"excellent"*. This favorable distribution reflects the rigorous quality control measures employed during dataset curation, as both Tulu and SmolTalk use capable LLMs for data generation and employ quality checks. Nevertheless, a non-negligible minority (10%) is rated as *"poor"* or *"very poor"*, indicating either lack of context or unclear instructions (see App. C.4 for details).

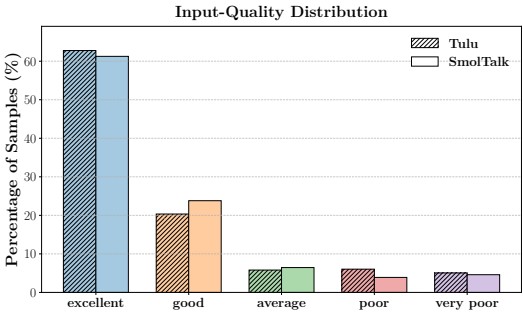

Figure 4: Distribution of input qualities: Both datasets contain over 80% good or excellent user inputs, indicating well-formulated prompts.

Additionally, we observe that LLMs face challenges in providing high-quality responses to poorly formulated user queries, which is directly reflected in their response quality. Fig. 5a and Fig. 5b show the instruct reward distribution for single-turn examples for Tulu and SmolTalk, highlighting the dependence of response quality on input quality. For both datasets, higher-quality instructions generally result in substantially better instruct rewards, indicating more helpful and contextually relevant responses from the corresponding LLMs. Examples are provided in App. C.5.3.

In contrast, multi-turn interactions follow a different trend, having "*good*" or "*excellent*" responses even if the input quality is subpar. In particular, most multi-turn samples either already have clear initial user queries, or, when ambiguity occurs, it tends to be explicitly resolved or clarified in subsequent turns. We provide concrete examples illustrating this pattern in App. C.5.3.

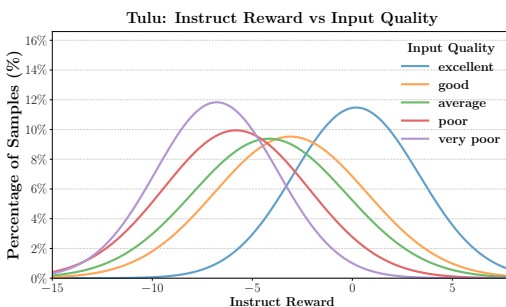

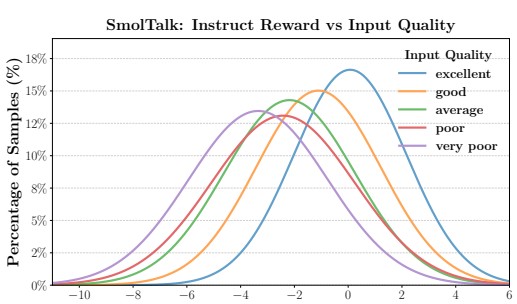

(a) Tulu: Distribution of single-turn instruct rewards by input quality.

(b) SmolTalk: Distribution of single-turn instruct rewards by input quality.

Figure 5: Relationship between input quality and instruct reward for single-turn samples in Tulu and SmolTalk. Higher-quality instructions consistently yield higher instruct rewards (better responses).

### 3.4 Difficulty, Language, and Safety

Magpie categorizes prompts as "*hard*" if they involve complex reasoning or specialized domain knowledge. For instance, approximately half (50%) of the Tulu samples are labeled as "*hard*", followed by "*easy*" (21%) and "*medium*" (18%). The rare instances tagged as "*very hard*" (8%) typically involve intricate judgments, such as those concerning current political contexts. Nonetheless, difficulty annotations show minimal correlation with primary data quality indicators such as instruct reward or input quality (see App. C.6). We thus omit further investigation of difficulty tags. Similarly,

language (see App. C.7) and safety (see App. C.8) annotations exhibit negligible correlation with data quality metrics. Both datasets are overwhelmingly English (Tulu: 95%; SmolTalk: 99%) and safe (Tulu: 97%; SmolTalk: 99%).

# 4 Leveraging Annotations to Design Data Curation Recipes

In this section, we leverage our Magpie annotations to curate the Tulu and SmolTalk datasets based on the quality of inputs and responses. Specifically, our goal is to create a *quality-aware* SFT mixture by selectively combining high-quality samples from Tulu and SmolTalk.

**Ablation Setup.** We evaluate our curation recipes through ablation experiments. We use stratified sampling to extract a representative subsample of approximately 10% (about 100k examples) from each of the original Tulu and SmolTalk datasets, resulting in subsets *Tulu-100k* and *SmolTalk-100k*. A subsample size of 10% is chosen to make training more computationally efficient while preserving performance trends, as similarly demonstrated in the original Tulu experiments [7]. We apply our curation criteria to these subsamples, selecting high-quality instances to form a new data mixture.

## 4.1 Quality-Based Curation Recipe

**Recipe.** We begin with a straightforward yet intuitive curation approach. From both Tulu and SmolTalk, we first select multi-turn samples with the highest input quality (*excellent*) and the highest reward score (5). We also select single-turn samples with the highest input quality and reward score above the median (i.e., second quantile). Applying this curation recipe to Tulu-100k and SmolTalk-100k produces a mixture of $\sim$ 80k samples. We refer to this curated mix as *TuluTalk-80k*.

**Performance Analysis.** Table 2 compares evaluation results for TuluTalk-80k against the stratified Tulu-100k and SmolTalk-100k subsamples when fine-tuned on Llama and SmolLM models. For Llama, TuluTalk-80k generally outperforms Tulu-100k, though it remains behind SmolTalk-100k overall. TuluTalk-80k achieves the highest performance on reasoning and commonsense benchmarks. Notably, while it slightly surpasses Tulu-100k on GSM8K (66.64% vs. 65.88%), TuluTalk-80k trails in instruction following tasks (IF-Eval) by over 2%, and significantly underperforms on code benchmarks (HumanEval). For SmolLM, the trend is slightly different: Instruction following performance improves alongside GSM8K scores, but the coding tasks again tend to lag behind.

Given that LLM benchmarks predominantly emphasize coding, math, and instruction following tasks, our initial quality-based curation approach might appear overly simplistic. In particular, strict quality filtering may have skewed task diversity and inadvertently removed examples crucial for instruction following and coding, thereby negatively impacting performance on related benchmarks. We investigate this by performing a diversity analysis on Magpie's task category tags.

**Diversity Analysis.** In our preliminary analysis, instruction following emerged as a critical capability influencing performance on other benchmarks (similar observations are also reported in [7, 30]). By filtering the annotated Tulu and SmolTalk datasets for sources explicitly containing instruction following tasks, we observe that many such examples fall into the categories *advice seeking*, *information seeking*, *creative writing*, and *reasoning*. Fig. 6 illustrates the resulting task diversity across the considered datasets, highlighting significant reductions in these instruction-rich categories within TuluTalk-80k. Notably, the proportion of *information seeking* samples drops to 12% compared to 20% in SmolTalk and 25% in Tulu. This confirms that our quality-based curation recipe requires additional task-aware adaptation to include more instruction following examples.

## 4.2 Quality-Based and Task-Aware Curation Recipe

To balance quality and task diversity, we extend our quality-based recipe by adding samples from underrepresented task categories, albeit with slightly lower quality thresholds. Specifically, we augment the previous selection with: (1) Multi-turn samples with *excellent* input quality and reward score of 5, (2) Multi-turn samples with *good* input quality and reward score of 5, (3) Single-turn samples with *excellent* input quality and reward scores above the first quantile, and (4) Single-turn samples with *good* input quality and reward scores above the third quantile. Overall, this approach captures high-quality samples along with strategically selected samples that maintain diversity by slightly relaxing either input or output quality, resulting in 3k additional samples yielding the *TuluTalk-83k* subset. The detailed curation recipe is presented in App. D.

Table 2: SFT results for Llama-3.1-8B and SmolLM2-1.7B models fine-tuned on stratified subsets of Tulu, SmolTalk, and TuluTalk mixtures, evaluated on the Open LLM Leaderboards (averaged) and code benchmarks. The overall average is across all benchmarks. Best scores are in **bold**.

| | Llama-3.1-8B | | | | SmolLM2-1.7B | | | |
|---|---|---|---|---|---|---|---|---|
| **Benchmark** | Tulu (100k) | SmolTalk (100k) | TuluTalk (80k) | TuluTalk (83k) | Tulu (100k) | SmolTalk (100k) | TuluTalk (80k) | TuluTalk (83k) |
| *Knowledge* | | | | | | | | |
| MMLU (5-shot) | **63.27** | 62.61 | 63.09 | 62.90 | 48.27 | 49.88 | **49.95** | 49.53 |
| MMLU-Pro (5-shot) | 28.61 | 29.85 | 31.52 | **31.67** | 19.06 | **21.41** | **21.41** | 20.91 |
| TruthfulQA (0-shot) | 50.75 | 53.77 | 52.35 | **54.37** | **43.03** | 41.97 | 39.17 | 40.37 |
| GPQA (0-shot) | **30.12** | 29.70 | 28.78 | 28.02 | **29.28** | 27.43 | 26.09 | 27.10 |
| *Reasoning* | | | | | | | | |
| ARC-C (25-shot) | 54.44 | 58.79 | **59.64** | 58.45 | 47.10 | **49.91** | 48.89 | 48.67 |
| BBH (3-shot) | 42.32 | 42.39 | **42.77** | 41.82 | **36.82** | 35.91 | 36.14 | 36.74 |
| MuSR (0-shot) | **42.33** | 39.15 | 37.30 | 37.83 | 34.26 | **35.19** | 33.60 | 34.26 |
| *Commonsense* | | | | | | | | |
| HellaSwag (10-shot) | 60.52 | 62.21 | **62.70** | 62.54 | 40.33 | 42.91 | **44.66** | 42.99 |
| WinoGrande (5-shot) | 76.95 | 77.66 | **77.90** | 76.80 | 65.35 | **67.48** | 67.09 | 66.51 |
| *Instruction Following* | | | | | | | | |
| IF-Eval (0-shot) | **66.03** | 65.66 | 64.38 | 63.94 | 49.13 | 47.90 | 49.19 | **52.50** |
| *Math* | | | | | | | | |
| GSM8K (5-shot) | 65.88 | 67.70 | 66.64 | **69.45** | 40.33 | 42.91 | **44.66** | 42.99 |
| MATH (4-shot) | **10.50** | 7.93 | 8.31 | 8.31 | **3.85** | **3.85** | 3.25 | 3.32 |
| *Code* | | | | | | | | |
| HumanEval (pass@1) | 50.61 | **52.44** | 48.76 | 51.22 | **1.83** | **1.83** | 1.22 | **1.83** |
| HumanEval+ (pass@1) | 30.61 | **34.51** | 32.43 | 32.44 | 0.61 | **1.22** | 0.61 | **1.22** |
| *Leaderboards* | | | | | | | | |
| Open LLM Leaderboard 1 | 61.97 | 63.79 | 63.72 | **64.09** | 49.57 | **50.96** | 50.59 | 50.13 |
| Open LLM Leaderboard 2 | **36.65** | 35.78 | 35.51 | 35.26 | 28.73 | 28.62 | 28.28 | **29.14** |
| *Overall* | 48.07 | **48.88** | 48.33 | 48.55 | 33.73 | **34.32** | 33.93 | 34.19 |

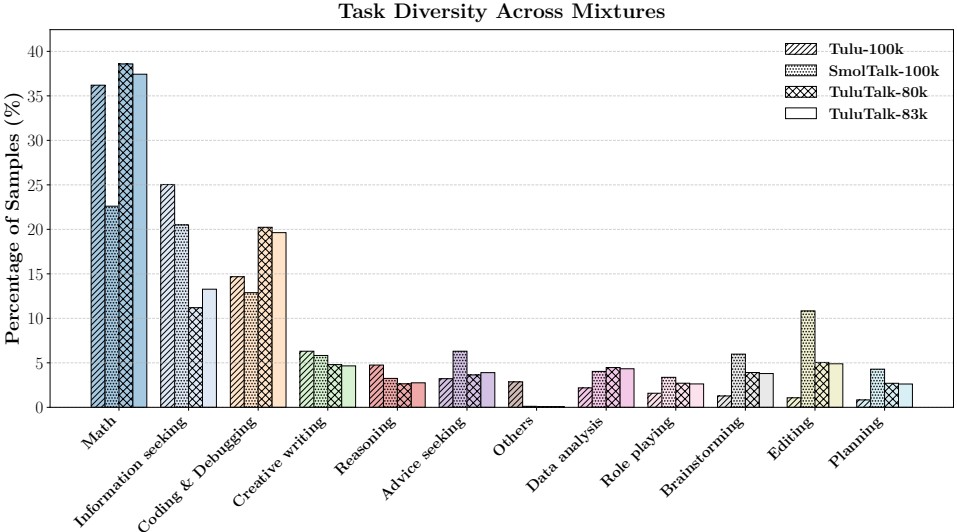

Figure 6: Task diversity distribution for stratified Tulu-100/SmolTalk-100 datasets and curated TuluTalk-80k/TuluTalk-83k mixtures. Our task-aware adaptation in TuluTalk-83k brings back 3k samples from underrepresented categories in TuluTalk, improving downstream task performance.

Table 3: SFT results for Llama-3.1-8B and SmolLM2-1.7B base models fine-tuned on Tulu, SmolTalk, Orca, and TuluTalk, evaluated on the Open LLM Leaderboards (averaged) and code benchmarks. The overall average is across all benchmarks. Best scores (row-wise) are in **bold**. Color-shaded columns highlight the TuluTalk models.

| Benchmark | Llama-3.1-8B | | | | | SmolLM2-1.7B | | | | |
| --- | --- | --- | --- | --- | --- | --- | --- | --- | --- | --- |
| | Base | Tulu | SmolTalk | Orca | TuluTalk | Base | Tulu | SmolTalk | Orca | TuluTalk |
| *Knowledge* | | | | | | | | | | |
| MMLU (5-shot) | **65.03** | 62.90 | 62.88 | 62.64 | 63.91 | 50.09 | 49.71 | 47.88 | **51.65** | 49.34 |
| MMLU-Pro (5-shot) | **32.71** | 28.73 | 31.76 | 31.89 | 30.17 | 21.26 | 19.61 | 20.37 | **23.40** | 20.67 |
| TruthfulQA (0-shot) | 45.22 | 46.41 | **55.74** | 52.08 | 53.16 | 36.61 | 44.04 | **44.74** | 42.84 | 43.65 |
| GPQA (0-shot) | 37.96 | **42.86** | 38.49 | 40.21 | 40.62 | **34.66** | 33.33 | 33.86 | 33.20 | 33.28 |
| *Reasoning* | | | | | | | | | | |
| ARC-C (25-shot) | 54.69 | 54.61 | **59.04** | 53.07 | 57.42 | **51.54** | 44.54 | 48.46 | 46.25 | 47.27 |
| BBH (3-shot) | **46.48** | 39.06 | 45.50 | 45.74 | 43.50 | 34.04 | 36.66 | 37.81 | 38.05 | **38.33** |
| MuSR (0-shot) | 37.96 | **42.86** | 38.49 | 40.21 | 40.62 | **34.66** | 33.33 | 33.86 | 33.20 | 33.28 |
| *Commonsense* | | | | | | | | | | |
| HellaSwag (10-shot) | 61.44 | 60.87 | 61.54 | 60.60 | **62.98** | 53.65 | 51.01 | 52.10 | 51.61 | 51.36 |
| WinoGrande (5-shot) | 76.87 | 76.64 | 77.19 | 71.19 | **79.22** | 68.19 | **65.90** | 65.27 | 64.96 | 66.06 |
| *Instruction Following* | | | | | | | | | | |
| IF-Eval (0-shot) | 12.45 | 74.09 | 74.51 | 57.73 | **74.84** | 23.91 | 60.25 | 56.83 | 35.17 | **60.85** |
| *Math* | | | | | | | | | | |
| GSM8K (5-shot) | 50.64 | 74.37 | 74.75 | 60.58 | **74.84** | 29.64 | 49.43 | 52.46 | 29.34 | **54.13** |
| MATH (4-shot) | 5.97 | **12.31** | 10.42 | 11.86 | 11.96 | 2.64 | **6.27** | 5.89 | 5.82 | 6.16 |
| *Code* | | | | | | | | | | |
| HumanEval (pass@1) | 34.76 | **58.54** | 54.51 | 51.37 | 56.49 | 0.61 | **1.83** | **1.83** | 0.61 | **1.83** |
| HumanEval+ (pass@1) | 28.66 | **45.37** | 44.27 | 40.29 | 44.33 | 0.61 | **1.83** | **1.83** | 0.61 | **1.83** |
| *Leaderboards* | | | | | | | | | | |
| Open LLM Leaderboard 1 | 58.98 | 62.63 | 65.19 | 60.03 | **65.26** | 48.29 | 50.77 | 51.82 | 47.78 | **51.97** |
| Open LLM Leaderboard 2 | 27.84 | 37.47 | 38.24 | 36.05 | **38.40** | 24.14 | 30.66 | 30.39 | 27.67 | **31.16** |
| *Overall* | 41.74 | 50.32 | 51.38 | 47.72 | **51.62** | 31.13 | 35.16 | 35.49 | 32.42 | **35.89** |

**Performance Analysis.** Table 2 compares TuluTalk-83k to previous mixtures, showing clear improvements on benchmarks where earlier versions underperformed. For the Llama model, TuluTalk-83k surpasses TuluTalk-80k by 2.8% on GSM8K (69.45% vs. 66.64%) and by 2.46% on HumanEval (51.22% vs. 48.76%). Overall, it performs slightly better than TuluTalk-80k. For the SmolLM model, TuluTalk-83k also yields higher overall performance, with the largest gain observed on IF-Eval with an improvement of 2.6%, rising from 49.19% to 52.5%. These results confirm the effectiveness of our adapted task-aware curation strategy and motivate applying our recipe to the full datasets.

# 5   Results on Full Datasets and Discussion

Building on insights from our ablations and prior analysis with smaller subsets, we apply our quality-based and task-aware data-curation recipe to the entire (annotated) SmolTalk and Tulu datasets, resulting in **TuluTalk**, a leaner SFT mixture comprising 808k samples. This represents a reduction of approximately 23% compared to SmolTalk and 14% compared to Tulu.

In Table 3, we report the SFT results for Llama and SmolLM models fine-tuned on the full Tulu, SmolTalk, Orca, and TuluTalk datasets, using the same experimental setup as before (see App. E.1 for details). *On average, TuluTalk outperforms all other SFT datasets for both models.*

For the Llama model, TuluTalk achieves an overall average of 51.62%, outperforming SmolTalk (51.38%), Tulu (50.32%), and significantly surpassing Orca (47.72%). In knowledge benchmarks, TuluTalk leads on MMLU at 63.91%, surpassing both Tulu and SmolTalk by 1%. While slightly behind SmolTalk on TruthfulQA, it remains competitive and outperforms SmolTalk on GPQA. On reasoning tasks, TuluTalk notably improves performance on ARC-C, achieving 57.42% (2.8% higher than Tulu), and on BBH, reaching 43.50% (4.4% higher than Tulu). Commonsense benchmarks also show clear improvements: HellaSwag at 62.98% (2.1% gain over Tulu) and WinoGrande at 79.22% (2.6% improvement over Tulu). TuluTalk further achieves the highest instruction following performance across datasets on IF-Eval, reaching 74.84%. It also demonstrates strong capabilities on math tasks, with 74.84% on GSM8K and 11.96% on the challenging MATH benchmark. Coding

Table 4: SFT results for Qwen2.5-0.5B, Qwen2.5-3B, and SmolLM3-3B base models fine-tuned on Tulu, SmolTalk, and TuluTalk, and evaluated on the Open LLM Leaderboards (averaged). The overall average is across all benchmarks. Best scores (row-wise) are in **bold**.

| | Qwen2.5-0.5B | | | | Qwen2.5-3B | | | | SmolLM3-3B | | | |
|---|---|---|---|---|---|---|---|---|---|---|---|---|
| Benchmark | Base | Tulu | SmolTalk | TuluTalk | Base | Tulu | SmolTalk | TuluTalk | Base | Tulu | SmolTalk | TuluTalk |
| *Leaderboards* | | | | | | | | | | | | |
| Open LLM Leaderboard 1 | 41.62 | 42.04 | 42.26 | **42.53** | 60.73 | 61.36 | 61.73 | **61.95** | 60.08 | 59.97 | 61.72 | **62.07** |
| Open LLM Leaderboard 2 | 22.01 | 24.73 | 23.58 | **25.09** | 32.13 | 38.29 | 36.77 | **38.99** | 31.18 | 37.98 | 37.21 | **38.18** |
| *Overall* | 31.19 | 32.45 | 31.92 | **32.68** | 44.89 | 48.67 | 47.61 | **48.94** | 43.90 | 47.68 | 47.85 | **48.34** |

performance remains robust, with scores of 56.49% on HumanEval and 44.33% on HumanEval+. On both Open LLM Leaderboards, TuluTalk achieves top scores, surpassing both SmolTalk and Tulu.

Similarly, for the SmolLM model, TuluTalk achieves an overall average of 35.89%, exceeding SmolTalk (35.49%), Tulu (35.16%), and Orca (32.42%). TuluTalk improves notably on instruction following benchmarks (IF-Eval at 60.85%) and math tasks (e.g., GSM8K at 54.13%). Its performance in knowledge benchmarks remains competitive, though slightly behind Orca in MMLU. Reasoning benchmarks show improvements, particularly for BBH at 38.33%, leading across all other datasets. Aggregated results on the OpenLLM Leaderboards further confirm TuluTalk's leading position, surpassing all compared datasets.

Collectively, our results show that TuluTalk consistently achieves top-tier performance across diverse tasks and two models, offering significant efficiency advantages with fewer yet higher-quality samples. A detailed analysis of the corresponding training efficiency is provided in App. E.3.3.

Furthermore, to assess generalizability across model scales and architectures, we conduct additional experiments using Qwen2.5-0.5B and Qwen2.5-3B [22], as well as SmolLM3-3B [31]. Table 4 reports the Open LLM Leaderboard and overall average scores across all benchmarks for each model. The results show that TuluTalk consistently outperforms Tulu and SmolTalk across all models, confirming our prior analysis and demonstrating robust cross-model generalization. Comprehensive evaluations for each model are provided in App. E.3.4. In addition, App. E.3.5 presents results for the Llama model fine-tuned with Direct Preference Optimization (DPO) [32], further demonstrating that the performance gains observed under SFT carry over to the DPO setting.

## 6   Conclusion

In this work, we annotated and systematically dissected the Tulu and SmolTalk post-training datasets, thoroughly quantifying their composition across multiple quality and task dimensions. Leveraging these detailed annotations, we developed a principled, quality-based, and task-aware data-curation recipe based on insights through ablations. This approach allowed us to construct *TuluTalk*, a new dataset mixture which not only significantly reduces dataset size (23% smaller than SmolTalk and 14% smaller than Tulu), but also consistently outperforms existing datasets across a comprehensive suite of benchmarks. Our results show that (1) high-quality samples, rather than sheer quantity, drive substantial performance gains, (2) differentiating single-turn from multi-turn interactions is essential for nuanced dataset curation, and (3) optimal data mixture ratios are inherently task-dependent, requiring careful balancing of quality, diversity, and representativeness. Robust evaluations conducted across multiple benchmarks and different LLM architectures ensure broad applicability of both our curation recipe and our TuluTalk mixture. By demonstrating how targeted, quality-aware curation can substantially enhance model capabilities while reducing resource demands, our work sets clear directions for future dataset curation efforts. We discuss limitations and broader impact in App. F.

## Acknowledgments and Disclosure of Funding

This work was supported in part by the German Federal Ministry of Education and Research (BMBF) within the research hub 6G-life (Grant 16KISK002), by the Bavarian Ministry of Science and the Arts and the Saxon Ministry for Science, Culture, and Tourism through the project Next Generation AI Computing (gAIn), by the Bavarian Ministry of Economic Affairs, Regional Development and Energy through the project 6G Future Lab Bavaria, and in part by IBM Research.

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

# A Large Language Model Post-Training

While *pre-training* equips models with general linguistic and world knowledge, *post-training* refines this capability to follow user instructions, align with human preferences, and exhibit safe and helpful behavior across downstream tasks.

## A.1 Post-Training Workflow

Post-training typically consists of instruction tuning via supervised fine-tuning (SFT), followed by preference fine-tuning and reasoning alignment, often involving reinforcement learning (RL).

**Supervised Fine-Tuning (SFT).** The goal of SFT is to adapt a pre-trained model to generate helpful and relevant outputs in response to natural language instructions. This is typically achieved by training on high-quality instruction-response pairs and multi-turn conversations, sourced from either human-written or synthetic datasets. During SFT, the model learns to generalize instruction formats, task types, and conversational patterns via next-token prediction. While SFT substantially improves instruction following and task performance, it does not guarantee alignment with human preferences, especially in cases where multiple plausible responses exist. To further refine the model, preference fine-tuning (also referred to as alignment) is applied.

**Preference Fine-Tuning.** The goal of preference fine-tuning is is to align the model's output distribution with human preferences or task-specific objectives. This is typically done by guiding the model using a reward model or preference signal to prefer helpful, harmless, and honest completions. Popular algorithms for preference tuning include Proximal Policy Optimization (PPO) [33], Group Relative Policy Optimization (GRPO) [34], and Direct Preference Optimization (DPO) [32].

**Deep Thinking and Reasoning Alignment.** Recent work has explored reinforcement learning and preference-based methods to enhance *deep thinking* capabilities in LLMs, such as multi-hop reasoning [35], chain-of-thought generation [36], tool use [37], and debate-style deliberation [38]. These methods typically rely on reward models or heuristic scoring to reward structured reasoning behavior that extends beyond surface-level fluency. Corresponding reasoning-centric datasets have emerged as well [39–43], which introduce task formats that elicit step-by-step thought processes.

## A.2 Focus on SFT

The primary goal of this paper is to analyze the quality and composition of training datasets while keeping the training procedure fixed. In particular, we focus on SFT because the performance of SFT-tuned models is largely governed by the structure and quality of the data mixture rather than by training algorithmic nuances. Furthermore, most open-source SFT pipelines follow similar training setups, whereas preference fine-tuning introduces additional complexity: the training algorithm (e.g., PPO, DPO, RLVR) directly determines the type and structure of data it can effectively utilize. For example, PPO requires preference pairs to train a reward model, followed by policy rollouts for fine-tuning [33]. DPO, by contrast, directly trains on preference pairs without requiring policy rollouts or a reward model [32]. Other methods like Reinforcement Learning with Verifiable Rewards (RLVR) require examples with verifiable numeric rewards [7].

The diversity of preference tuning recipes and their data format dependencies thus makes clean cross-method comparisons challenging. Indeed, designing and evaluating preference-based training pipelines is itself an active research area [7, 44, 45]. We leave the analysis of data quality under different alignment strategies to future work (see App. F).

Nevertheless, to assess whether our SFT curation insights transfer to preference-tuned models, we also apply DPO on Llama models fine-tuned on Tulu, SmolTalk, and our proposed TuluTalk mixture (see App. E). As shown in Table 19, the TuluTalk dataset consistently outperforms both Tulu and SmolTalk under DPO, just as it does under SFT, confirming that careful data mixture design offers robust gains across post-training stages. These results further validate our focus on dataset composition as a critical axis of post-training quality.

### A.3 Related SFT Datasets

Tulu [7] and SmolTalk [9], investigated in this paper, are two of the most recent and widely used open-source datasets for SFT post-training of LLMs. We focus on these two datasets due to their strong reported performance over prior SFT datasets across a broad range of benchmarks when used to train the respective models introduced in their original papers.

Several other SFT datasets have been proposed in recent years, including *Orca* [16], *OpenHermes* [12], *LongAlign* [46], *UltraFeedback* [47], *MAmmoTH2* [48], *DaringAntEater* [49] *Magpie-Pro* [14] *RLHFlow-SFT-V2* [50]. While many of these datasets provide valuable capabilities, such as long-context support, synthetic feedback signals, or broad coverage across domains, Tulu and SmolTalk remain highly competitive, achieving significantly stronger performance across instruction following, reasoning, and code tasks [7, 9].

In our main paper, we compare Tulu and SmolTalk directly against Orca, demonstrating that Orca lags notably behind, particularly in code generation performance.

Furthermore, as shown in App. B, both Tulu and SmolTalk include carefully curated subsets drawn from several of the datasets mentioned above, particularly from OpenHermes2.5 [12], Smol-Magpie-Ultra [9], OpenAssistant [19], and UltraFeedback [47].

# B  Dataset Composition of Tulu and SmolTalk

In this section, we provide a brief overview and composition summary of Tulu[6] and SmolTalk[7] post-training datasets.

## B.1  Tulu

The Tulu dataset was created to bridge proprietary and open-source post-training data by leveraging publicly available datasets, persona-driven synthetic prompts, and rigorous decontamination procedures to mitigate test set leakage. Specifically, Lambert et al. [7] collected 23,327,961 candidate prompts from over 20 distinct sources, curating a multi-skill SFT corpus that comprises 939,344 samples, forming the original Tulu-3-SFT-Mix data mixture. The Tulu subsets and their respective samples can be broadly categorized into nine high-level groups:

- *General:* OpenAssistant (OASST1) [19], No Robots [51], WildChat [8], UltraFeedback (Tülu HC-10) [47]
- *Knowledge Recall:* FLAN v2 [52], SciRIFF [53], TableGPT [54]
- *Math:* Persona MATH, Persona MATH (Grade) [55]
- *Reasoning:* Persona Algebra [56], OpenMathInstruct2 [10], NuminaMath-TIR [23]
- *Coding:* Persona Code [57], Evol CodeAlpaca [58]
- *Safety & Non-Compliance:* CoCoNot [59], WildJailbreak [60], WildGuardMix [61]
- *Multilingual:* Aya [62]
- *Precise Instruction Following:* Persona IF [63]
- *Other:* <1,000 examples from miscellaneous small sources

## B.2  SmolTalk

The SmolTalk dataset was developed to address the lower instruction-tuned performance of the SmolLM2 base model [9]. Specifically, Allal et al. [9] blend high-quality conversational, task-specific, math, and code datasets, filtered and generated via Distilabel [64] annotations, to cover a wide range of instruction-following capabilities. This results in a multi-domain post-training corpus of 1,043,917 training samples which is used for SFT of SmolLM2 to boost instruction following, reasoning, and conversational skills in a reproducible, open-source pipeline. The SmolTalk subsets can be similarly grouped into seven high-level categories:

- *General:* Everyday-Conversations [65], LongAlign [46], OpenHermes2.5 [12], Smol-Magpie-Ultra [9], Self-OSS-Starcoder-2-Instruct (Self-OSS-2) [66], SystemChats2.0 [67]
- *Knowledge Recall:* Smol-Summarization [9]
- *Math:* MetaMathQA-50k [24]
- *Reasoning:* NuminaMath-CoT [23]
- *Coding:* APIGen-80k [26]
- *Safety & Non-Compliance:* Smol-Constraints [9]
- *Precise Instruction Following:* Explore-Instruct-Rewriting [68], Smol-Rewrite [9]

We provide a detailed dataset- and sample-level breakdown of the annotated Tulu and SmolTalk datasets in Section C.

---

[6]https://huggingface.co/datasets/allenai/tulu-3-sft-mixture
[7]https://huggingface.co/datasets/HuggingFaceTB/smoltalk

## C  Extended Quality Analysis

We present detailed insights and extended analyses of our annotated Tulu and SmolTalk post-training datasets, covering dataset composition, task distribution, and quality metrics.

### C.1  Magpie Annotations

This section introduces the Magpie annotation framework and outlines our extensions to support the tagging of multi-turn conversation samples.

#### C.1.1  General Overview

Magpie [14] is a *self-synthesis* pipeline that extracts alignment annotations from open-weight, instruction-tuned LLMs without relying on seed prompts or human supervision. While Magpie can generate synthetic instruction-response pairs, we focus in this work on its *annotation* capabilities.

In particular, Magpie uses specialist judge models to annotate data samples along multiple dimensions (e.g., input quality, task category, safety), enabling scalable, automated labeling of large datasets that would be infeasible to annotate manually. This metadata can be used for filtering, stratification, or targeted analysis of the corpus.

Magpie supports the following annotation tags:

- **Input Quality** (*very poor – excellent*): Measures the clarity, specificity, and structure of the prompt. Includes a textual justification.
- **Task Category**: Assigns each sample to one of 12 categories, including *Coding & Debugging, Reasoning, Information Seeking, Brainstorming, Creative Writing, Advice Seeking, Math, Planning, Editing, Role Playing, Data Analysis*, and *Others*.
- **Input Difficulty** (*very easy – very hard*): Captures reasoning complexity and knowledge demands. Also tags *intent* (user goal) and *knowledge* (required model competence).
- **Safety**: Evaluated using a dedicated safety guard model.
- **Response Quality (Instruct Reward)**: Scored by a reward model based on the overall quality of the assistant's response.
- **Language**: Detects the language of the user input.

Magpie is fully modular such that the judge model can be substituted by any LLM in principle. By default, Magpie uses Llama-3-8B-Instruct [27] for most annotation tasks, FsfairX-LLaMA3-RM-v0.1 [14] for instruct reward scoring, and Llama-Guard 2 [29] for safety classification.

#### C.1.2  Extensions to Magpie

In its original form, Magpie does not support tagging of multi-turn conversation samples and is limited by short context windows and frequent inconsistencies in LLM outputs. To address these limitations, we extend the framework to support more robust annotation of realistic, multi-turn data.

**1) Multi-Turn Adaptation.**  Magpie was originally designed for single-turn samples, where most annotations, such as instruct reward or input quality, are computed using only the first user-assistant exchange. However, as shown in later analysis, many multi-turn conversations undergo clarification or iterative refinement before resulting in a high-quality response. Thus, the original pipeline is insufficient for evaluating such interactions.

To support multi-turn conversations, we modify Magpie's prompts to incorporate the entire conversation history, rather than just the initial turn, and adapt reward scoring accordingly. Additionally, we raise the context window to the maximum length supported by the chosen LLM, as the default Magpie configuration sets this value conservatively low. We provide all modifications as part of our code repository[8].

---

[8]Code available at: github.com/aladinD/magpie-single-and-multi-turn

*Multi-Turn Prompt Template*: In most cases, adapting Magpie for multi-turn use simply involves replacing the single-turn user input with the full conversation history in the prompt template. Furthermore. to improve robustness under increased context length, we enforce stricter formatting by adding an in-context example and explicitly specifying the expected JSON output format. An example of these adaptations for Magpie's multi-turn task classification prompt is shown in Fig. 7.

Together with the increased context window length, this adaptation ensures that the judge model can process the full conversation history reliably.

*Multi-Turn Instruct Reward*: While adapting most annotation tags is straightforward, computing instruct rewards for MT conversations is more complex. Magpie uses FsfairX-LLaMA3-RM-v0.1 [14], a reward model that assigns a continuous reward score $r^*$ to each instruction-response pair. To contextualize this score, it also computes a baseline reward $r_{\text{base}}$ using a reference model (typically the main LLM judge) on the same instruction. The difference $\Delta r = r^* - r_{\text{base}}$ reflects the relative improvement in response quality and is reported as the instruct reward.

This reward mechanism was originally developed for single-turn filtering and for supporting preference optimization via Magpie's DPO implementation. However, it does not generalize cleanly to MT settings, where generating a comparable baseline response for the entire conversation becomes infeasible, particularly when the number of samples in the dataset is large.

To address this, we treat ST and MT samples separately: For *ST samples*, we retain Magpie's original reward scoring pipeline based on the reference reward model, which is generally fast and reliable with low tagging error rates. For *MT samples*, we choose to avoid computing reference model-based rewards and instead use a dedicated LLM-as-a-Judge (typically the main LLM judge) to evaluate the entire conversation on a discrete scale from 0 to 5.

A unified reward annotation pipeline that applies such a judge-based scoring to both ST and MT samples is certainly feasible, but we leave its development to future work.

**2) Reliable and Error Tolerant Prompts.** Due to the LLM-as-a-judge nature of Magpie's annotation framework, inconsistent or free-form outputs are frequently observed. This occurs particularly when the LLM fails to follow strict formatting instructions for producing structured annotations. For example, many Magpie prompts require the model to output a score or label in a JSON-formatted response (see Fig. 7). Depending on the chosen LLM judge, inconsistencies such as `<Information seeking`, `<INFORMATION SEEKING>`, or `["information seeking"]`, i.e., outputs with malformed brackets and inconsistent formatting, are common. While typically benign, the original Magpie JSON parser is brittle and fails on such responses.

In addition to including in-context examples in Magpie prompts, we introduce a lightweight *forgiving parser* that replaces the original `json.loads()` call with a more tolerant multi-stage pipeline. Specifically, the parser performs the following:

- **Brace normalization:** Collapses nested braces and extracts only the first JSON block.
- **Regex-based sanitization:** Fixes unbalanced quotes, braces or backslashes, inserts missing commas, and lowercases keys via targeted regular expressions.
- **Wrapper stripping:** Removes Markdown fences, discards any text outside the first and last braces, and truncates after the final closing brace.
- **Special-case fallback:** Supports bare-number shorthands for instruct reward scoring by mapping single digits to a default score schema.
- **Graceful degradation:** Wraps parsing in `try/except` blocks, logs failed cases, and resets only task-specific fields without discarding the full batch.

This improved parser reliably extracts valid JSON fragments from noisy outputs, tolerating extra braces, formatting artifacts, and minor syntax violations where the original parser would simply fail. Remaining inconsistencies are rare and can be resolved through lightweight post-processing. Overall, this enhancement reduces tagging errors by up to 15%.

```
Multi-Turn Magpie Prompt for Task Classification Tagging.

# Instruction

You will be given a conversation between a User and an AI assistant.

## Conversation
‘‘‘
\{CONVERSATION_HISTORY\}
‘‘‘

## Tagging the Conversation
Please analyze the conversation and select the most relevant task tag
from the list below:

all_task_tags = [
    "Information seeking", # Specific information or facts
    "Reasoning",           # Requires logical thinking
    "Planning",            # Assistance in creating plans/strategies
    "Editing",             # Editing, rephrasing, proofreading
    "Coding & Debugging",  # Writing, reviewing, or fixing code
    "Math",                # Math concepts, problems, and equations
    "Role playing",        # ChatGPT is asked to adopt a persona
    "Data analysis",       # Analyzing data and statistics
    "Creative writing",    # Writing stories, poems, or texts
    "Advice seeking",      # Guidance on personal/professional issues
    "Brainstorming",       # Generating ideas and creative thinking
    "Others"               # Queries that don't fit above categories
]
## Output Format:
You can only select a single primary tag.  Other tags can go into
‘"other_tags"‘.

{
  "primary_tag": "<primary tag>",
  "other_tags": ["<tag 1>", "<tag 2>", ...]
}

For instance:
‘‘‘json
{{
    "primary_tag": "Information seeking",
    "other_tags": ["Advice seeking", "Others"]
}}
‘‘‘

Make sure to adhere to this formatting.
```

Figure 7: Multi-turn Magpie prompt for task classification tagging. The original prompt is extended to include the full conversation history, ensuring that all user-assistant turns are evaluated by the judge model. To improve robustness under longer context windows, the template also includes an in-context example and an explicitly specified JSON output format.

### C.1.3 Choice of Judge Model

Magpie supports the use of any LLM as a potential annotation judge. In our experiments, we use Llama-3.3-70B-Instruct [27] as the primary judge model, based on two key considerations.

First, preliminary experiments with Qwen-based annotators revealed systematic biases. Fig. 8a and Fig. 8b show input quality distributions on 30k stratified subsets of Tulu and SmolTalk when annotated with Llama-3.3-70B-Instruct [27] versus Qwen2-72B-Instruct [69]. In both cases, Qwen strongly over-predicts the *excellent* label, while Llama produces more balanced annotations. In fact, our later analysis reveals a broader spread of input quality, with samples labeled as *good*, *excellent*, and even some rated as *average* or *poor*. Qwen tends to ignore these lower bands, particularly for SmolTalk, resulting in a less nuanced and potentially biased annotation profile.

Second, Magpie's default configuration uses Llama-3.1-8B-Instruct, such that much of the open-source pipeline has been tested and optimized for this model. Remaining within the same model family reduces integration friction and improves reproducibility, making the workflow more robust and accessible for the broader research community.

Based on these observations, particularly the qualitative differences in annotation quality revealed by our preliminary analysis, we select Llama-3.3-70B-Instruct as the default judge model for all annotation tasks in this study.

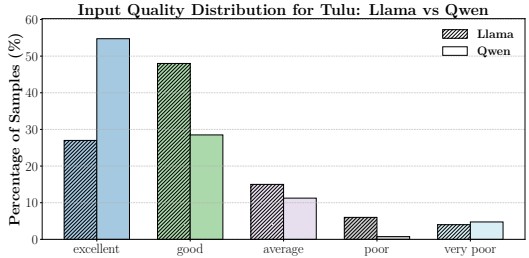
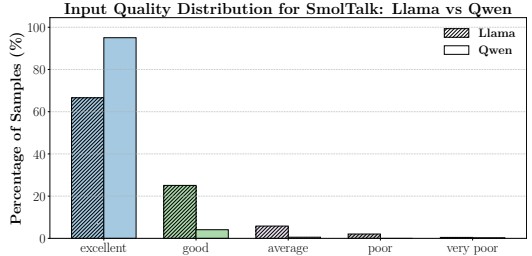

(a) Input quality distribution for a 30k stratified subset of Tulu using Llama and Qwen as Magpie annotators. Qwen strongly favors the *excellent* label, while Llama offers a more realistic spread.

(b) Input quality distribution for a 30k stratified subset of SmolTalk using Llama and Qwen. Again, Qwen exhibits a strong upward bias toward *excellent* input quality labels, unlike Llama which is more balanced.

Figure 8: Comparison of input quality annotations produced by Llama-3.3-70B-Instruct and Qwen2-72B-Instruct judge models on 30k stratified subsets of Tulu and SmolTalk. Qwen consistently over-predicts high-quality labels, while Llama provides a more balanced distribution.

### C.1.4 Human Evaluation of Annotation Quality

In addition to comparing different judge models, we perform a systematic manual inspection of a small subset of annotated samples to assess alignment between LLM-generated annotations and human judgment. To this end, we stratify 100 TuluTalk samples by task category and have four authors independently review 25 samples each for *input quality* and *instruct reward*. We then compute exact-match agreement and Spearman's rank correlation ($\rho$) between the LLM annotations and the human consensus. Table 5 shows that both input quality and instruct reward exhibit high categorical agreement ($\geq 90\%$) and strong positive rank correlation, indicating that the LLM's annotations closely align with human judgments. Most disagreements involve one-step differences (e.g., rating input quality as *excellent* versus *good*), likely reflecting subjective variation. Overall, these results confirm that a capable judge model, specifically the Llama-3.3-70B-Instruct used in our study, can reliably approximate expert annotations for both fine- and coarse-grained annotation tasks.

Table 5: Evaluation of annotation quality for 100 stratified TuluTalk samples: Exact-match agreement and Spearman's rank correlation ($\rho$) indicate strong alignment between LLM and human judgment.

| Annotation Category | Agreement (%) | Spearman ($\rho$) |
|---|---|---|
| Input Quality | 91 | 0.85 |
| Instruct Reward | 93 | 0.87 |

## C.2 Annotated Dataset Composition

Tables 6 and 7 provide an overview of the dataset-level composition by category and source subset after performing Magpie annotations.

**Tulu.** After tagging, the annotated Tulu dataset comprises 911,782 samples, resulting in a loss of only 3% samples due to tagging errors. In general, the dominant categories are *Math*, *Coding*, and *Reasoning*, where *Math* is notably overrepresented with 21.5% of samples. Other categories are more evenly distributed, with category shares ranging between 10-12%. Further, *Precise Instruction Following* appears surprisingly limited, making up only 3.3% of the dataset. *Other* samples are negligible, constituting just 0.1% of the total samples.

**SmolTalk.** After tagging, the annotated SmolTalk dataset comprises 1,024,791 samples, with a tagging failure rate of only 2%, resulting in minimal data loss. For SmolTalk, the dominant category is *General*, accounting for 57.6% of all samples. Within this category, the majority of samples stem from the *Smol-Magpie-Ultra* subset (39.8%), which contains multi-turn synthetic conversations designed to enhance open-domain fluency and context handling. This emphasis on general-purpose data is a deliberate curation choice aimed at bootstrapping conversational fluency, tone control, and context length generalization in *Smol* models. Notably, the *Math* and *Coding* categories are significantly underrepresented, comprising only 4.6% and 7.1% of the dataset, respectively, thus suggesting potential limitations in STEM-related reasoning coverage.

Magpie annotations allow us to build on this high-level dataset categorization with a more rigorous, fine-grained *sample-level* analysis in the following sections.

Table 6: Dataset-level composition of the annotated Tulu dataset after Magpie tagging, showing the number of samples, dataset share, and share within each task category.

| Category | Prompt Dataset | # Samples | Dataset % | Category % |
|---|---|---|---|---|
| *General* | No Robots | 8 703 | 1.0% | |
| | OASST1 | 7 117 | 0.8% | 12.2% |
| | Tülu HC-10 | 210 | 0.0% | |
| | WildChat | 94 470 | 10.4% | |
| *Knowledge Recall* | FLAN v2 | 89 828 | 9.9% | |
| | SciRIFF | 9 719 | 1.1% | 11.5% |
| | TableGPT | 4 962 | 0.5% | |
| *Math* | Persona MATH | 145 895 | 16.0% | 21.5% |
| | Persona MATH (Grade) | 49 973 | 5.5% | |
| *Reasoning* | NuminaMath-TIR | 56 699 | 6.2% | |
| | OpenMathInstruct2 | 49 997 | 5.5% | 13.8% |
| | Persona Algebra | 19 439 | 2.1% | |
| *Coding* | Evol CodeAlpaca | 106 882 | 11.7% | 15.5% |
| | Persona Code | 34 987 | 3.8% | |
| *Safety & Non-Compliance* | CoCoNot | 10 977 | 1.2% | |
| | Synth+WildGuardMix | 50 190 | 5.5% | 12.2% |
| | WildJailbreak | 49 998 | 5.5% | |
| *Multilingual* | Aya | 91 003 | 10.0% | 10.0% |
| *Precise Instruction Following* | Persona IF | 29 938 | 3.3% | 3.3% |
| *Other* | Other | 795 | 0.1% | 0.1% |
| **Total** | **20 datasets** | **911 782** | **100.0%** | **100.0%** |

Table 7: Dataset-level composition of the annotated SmolTalk dataset after Magpie tagging, showing the number of samples, dataset share, and share within each task category.

| Category | Prompt Dataset | # Samples | Dataset % | Category % |
|---|---|---|---|---|
| *General* | Everyday-Conversations | 2 249 | 0.2% | 57.6% |
| | LongAlign | 3 511 | 0.3% | |
| | OpenHermes2.5 | 94 439 | 9.2% | |
| | Smol-Magpie-Ultra | 407 971 | 39.8% | |
| | Self-OSS-2 | 48 085 | 4.7% | |
| | SystemChats2.0 | 34 120 | 3.3% | |
| *Knowledge Recall* | Smol-Summarization | 96 322 | 9.4% | 9.4% |
| *Math* | MetaMathQA-50k | 46 728 | 4.6% | 4.6% |
| *Reasoning* | Numina-CoT | 100 982 | 9.9% | 9.9% |
| *Coding* | APIGen-80k | 72 522 | 7.1% | 7.1% |
| *Safety & Non-Compliance* | Smol-Constraints | 34 175 | 3.3% | 3.3% |
| *Precise Instruction Following* | Explore-Instruct-Rewriting | 30 384 | 3.0% | 8.2% |
| | Smol-Rewrite | 53 303 | 5.2% | |
| **Total** | **13 datasets** | **1 024 791** | **100.0%** | **100.0%** |

### C.2.1 Token Length Distribution

To examine the token length distribution across the post-training datasets, we binned the per-sample token counts for both Tulu and SmolTalk (fine-tuned with Llama models) into 40 logarithmically spaced intervals ranging from $2^4$ (16) to $2^{13}$ (8,192) tokens. Fig. 9 and Fig. 10 show the resulting token length distributions across source subsets for Tulu and SmolTalk, respectively. Notably, different prompt sources exhibit distinct token length profiles, which can influence batch size, memory requirements, and learning dynamics when mixed during training. This variation in token lengths motivates our use of a sum-reduction over token-level losses, rather than the more commonly used mean-reduction (see App. E). A more in-depth discussion of this analysis is provided in Lambert et al. [7]. In our subsequent analysis, we do not further investigate token length characteristics, but include this section here for completeness.

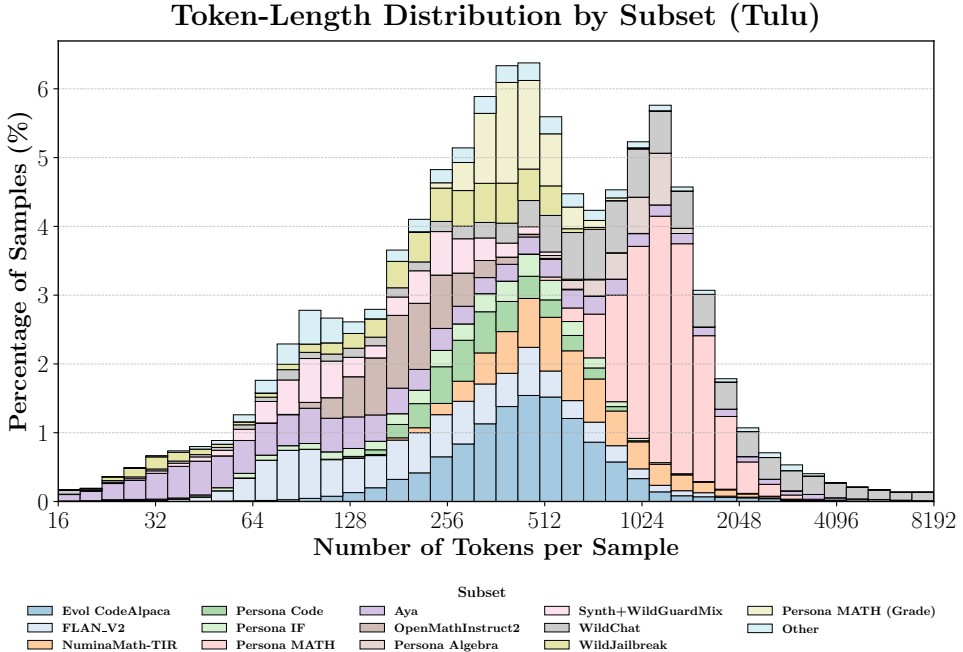

Figure 9: Token length distribution per post-training subset for Tulu. Synth+WildGuardMix and WildChat subsets feature the longest token lengths.

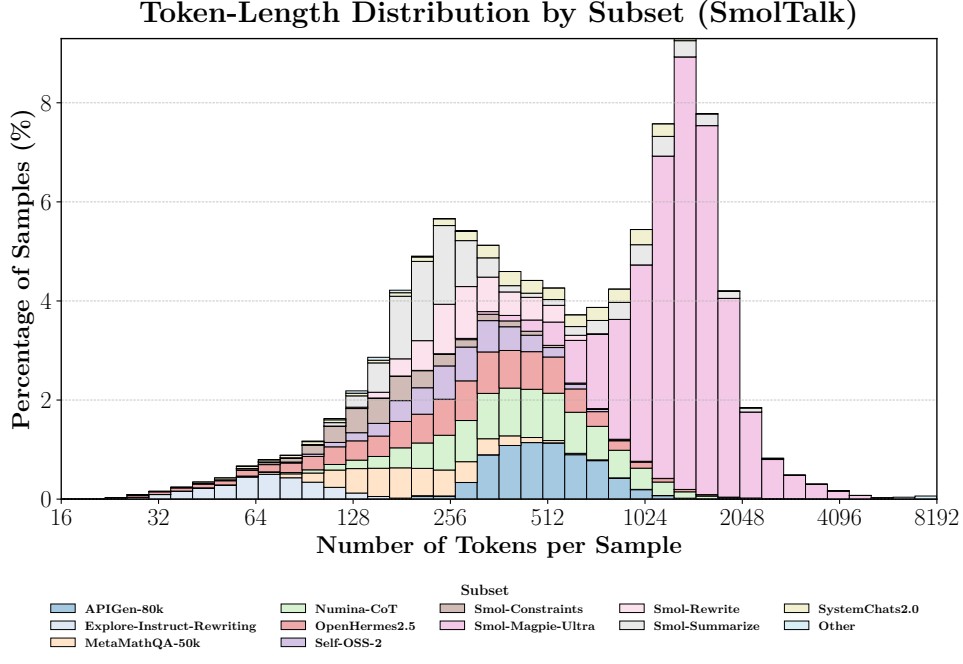

Figure 10: Token length distribution per post-training subset for SmolTalk. Smol-Magpie-Ultra features longer conversations and thus increased token lengths.

## C.3 Turn Types and Conversation Lengths

### C.3.1 Single-Turn vs. Multi-Turn Samples

Tulu and SmolTalk differ substantially in the distribution of single-turn (ST) and multi-turn (MT) samples. Fig. 11 shows the corresponding top-5 conversation lengths for both datasets.

**Tulu.** ST samples (i.e., 2 message exchanges in total between user and assistant) dominate the Tulu dataset, comprising 870,819 examples (95.5% of the data), compared to only 40,963 MT examples (4.5% of the data). Among MT samples, 4-turn conversations (i.e., a single follow-up) represent approximately 1.8% of the dataset. All higher-turn conversations individually account for less than 0.8% (see Fig. 11a). Thus, Tulu is an overwhelmingly *single-turn dataset*.

**SmolTalk.** In contrast, MT samples constitute the majority of SmolTalk, with 718,164 examples (70% of the data), while ST samples make up the remaining 306,627 (30% of the data). Within MT examples, 6-turn conversations dominate, accounting for approximately 39.8% of samples, followed by 3-turn conversations (mostly sourced from the *Smol-Magpie-Ultra* subset) at 28% of the data. All other turn counts are negligible, each contributing less than 0.5%. Consequently, SmolTalk is an overwhelmingly *multi-turn dataset*.

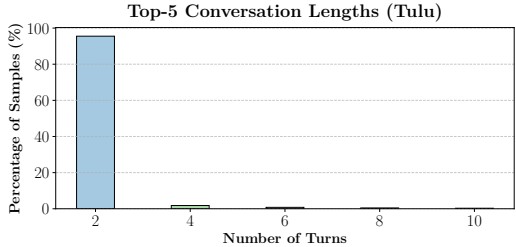
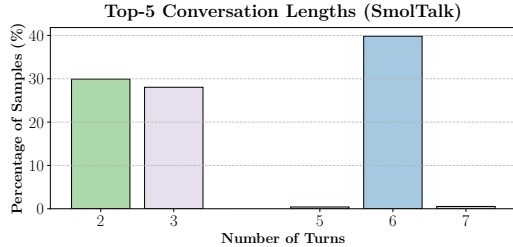

(a) Tulu: the majority of samples are single-turn.  (b) SmolTalk: the majority of samples are multi-turn.

Figure 11: Distribution of the top-5 conversation lengths. Tulu is overwhelmingly single-turn, whereas SmolTalk is predominantly multi-turn, albeit with a substantial single-turn segment.

### C.3.2 Turn Type per Task Category

Tables 8 and 9 compare the distribution of ST and MT samples across Magpie task categories for Tulu and SmolTalk. This constitutes a *sample-level* view of how different task categories are distributed across single-turn and multi-turn interactions.

**Tulu.**   All Magpie task categories in Tulu are heavily skewed toward single-turn interactions, with *Math*, *Information Seeking*, and *Coding* contributing the largest shares of ST samples. The highest multi-turn proportion is found in the *Information Seeking* category, where samples often reflect users iteratively refining or clarifying their queries to guide the LLM's response. Fig. 12 visualizes the relative ST and MT proportions across task categories for Tulu.

Table 8: Sample-level distribution of Single-turn (ST) vs. multi-turn (MT) examples in the annotated Tulu dataset by Magpie task category: (a) shows the relative proportion of ST/MT samples within each category, while (b) shows the category-wise share among all ST and MT samples, respectively.

(a) Relative composition within each task category (row-wise).

| Category | ST % | MT % | Total % |
|---|---|---|---|
| Advice seeking | 92.4 | 7.6 | 100.0 |
| Brainstorming | 90.7 | 9.3 | 100.0 |
| Coding & Debugging | 95.7 | 4.3 | 100.0 |
| Creative writing | 88.6 | 11.4 | 100.0 |
| Data analysis | 97.3 | 2.7 | 100.0 |
| Editing | 76.6 | 23.4 | 100.0 |
| Information seeking | 92.9 | 7.1 | 100.0 |
| Math | 99.5 | 0.5 | 100.0 |
| Other | 97.5 | 2.5 | 100.0 |
| Planning | 92.4 | 7.6 | 100.0 |
| Reasoning | 97.6 | 2.4 | 100.0 |
| Role playing | 82.7 | 17.3 | 100.0 |

(b) Distribution across ST and MT splits (column-wise).

| Category | ST % | MT % |
|---|---|---|
| Advice seeking | 3.1 | 5.5 |
| Brainstorming | 1.2 | 2.6 |
| Coding & Debugging | 14.8 | 14.2 |
| Creative writing | 5.9 | 16.1 |
| Data analysis | 2.3 | 1.4 |
| Editing | 0.9 | 5.6 |
| Information seeking | 24.2 | 39.5 |
| Math | 37.7 | 3.7 |
| Others | 2.9 | 1.6 |
| Planning | 0.8 | 1.4 |
| Reasoning | 4.9 | 2.6 |
| Role playing | 1.4 | 6.0 |
| **Total** | **100.0** | **100.0** |

**SmolTalk.**   All Magpie task categories in SmolTalk exhibit a strong skew toward multi-turn interactions. Notably, *Brainstorming*, *Role Playing*, and *Creative Writing* exceed 90% MT samples, reflecting their inherently conversational nature. *Coding & Debugging* and *Math* show the highest relative single-turn proportions (53.6% and 68.0%) of their respective categories (see Table 9a), indicating a prevalence of one-shot problem-solution pairs. When viewed within each turn-type split (see Table 9b), *Math* dominates the single-turn subset (51.7% of all ST samples), while *Information Seeking* leads among multi-turn samples (23.4% of MT samples). These patterns suggest that well-defined tasks often occur in single interactions, whereas more exploratory or research-oriented queries tend to span multiple turns. Fig. 13 visualizes the ST/MT distribution across task categories in SmolTalk.

Table 9: Sample-level distribution of Single-turn (ST) vs. multi-turn (MT) examples in the annotated SmolTalk dataset by Magpie task category: (a) shows the relative proportion of ST/MT samples within each category, while (b) shows the category-wise share among all ST and MT samples, respectively.

(a) Relative composition within each task category (row-wise).

| Category | ST % | MT % | Total % |
|---|---|---|---|
| Advice seeking | 8.3 | 91.7 | 100.0 |
| Brainstorming | 3.4 | 96.6 | 100.0 |
| Coding & Debugging | 53.6 | 46.4 | 100.0 |
| Creative writing | 7.0 | 93.0 | 100.0 |
| Data analysis | 4.2 | 95.8 | 100.0 |
| Editing | 12.3 | 87.7 | 100.0 |
| Information seeking | 20.3 | 79.7 | 100.0 |
| Math | 68.0 | 32.0 | 100.0 |
| Others | 6.0 | 94.0 | 100.0 |
| Planning | 4.7 | 95.3 | 100.0 |
| Reasoning | 12.9 | 87.1 | 100.0 |
| Role playing | 2.8 | 97.2 | 100.0 |

(b) Distribution across ST and MT splits (column-wise).

| Category | ST % | MT % |
|---|---|---|
| Advice seeking | 1.7 | 8.1 |
| Brainstorming | 0.7 | 8.4 |
| Coding & Debugging | 23.0 | 8.4 |
| Creative writing | 1.4 | 7.7 |
| Data analysis | 0.6 | 5.4 |
| Editing | 4.5 | 13.6 |
| Information seeking | 14.0 | 23.4 |
| Math | 51.7 | 10.3 |
| Others | 0.0 | 0.2 |
| Planning | 0.7 | 5.9 |
| Reasoning | 1.4 | 4.1 |
| Role playing | 0.3 | 4.6 |
| **Total** | **100.0** | **100.0** |

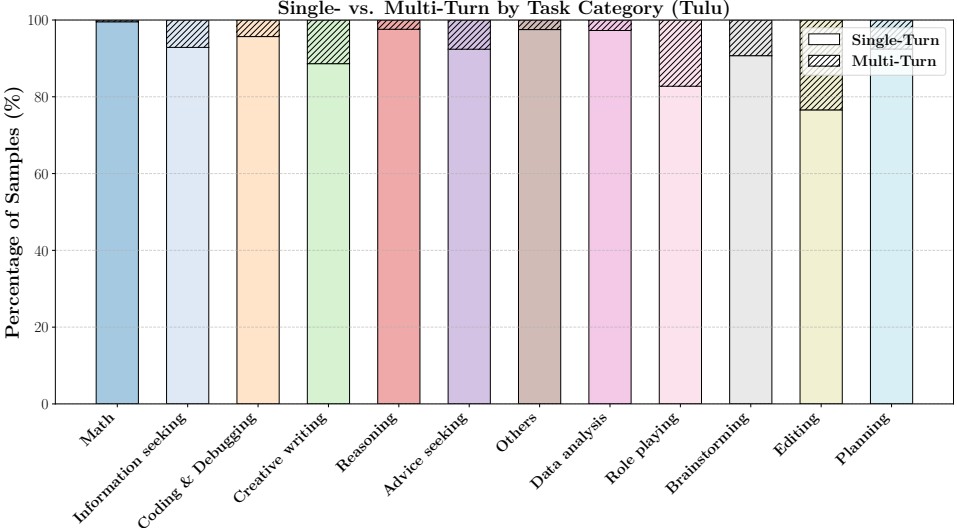

Figure 12: Turn type distribution in Tulu by Magpie task category. Most categories are dominated by single-turn (ST) samples, reflecting the dataset's focus on concise, one-shot interactions.

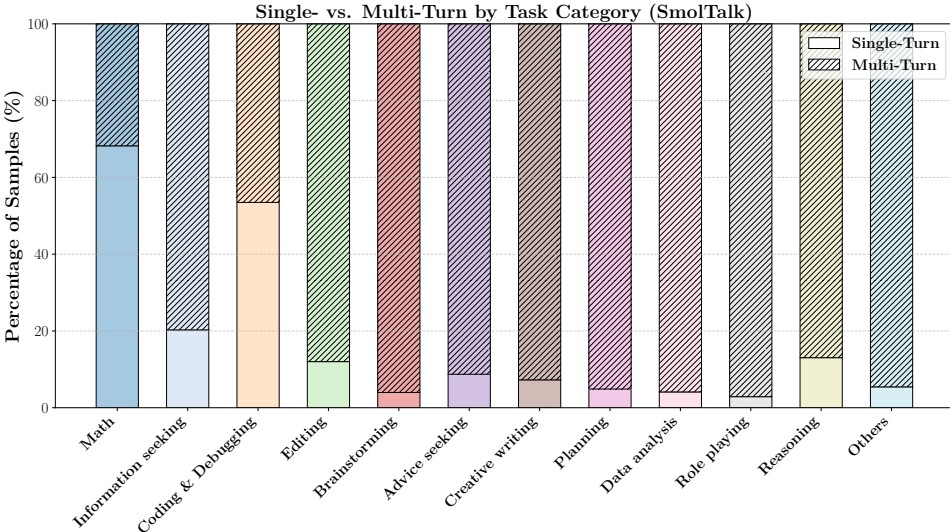

Figure 13: Turn type distribution in SmolTalk by Magpie task category. Most categories are dominated by multi-turn (MT) samples, consistent with the dataset's emphasis on dialogic interaction. *Math* stands out with a higher proportion of single-turn, one-shot problem-solution exchanges.

## C.4 Input Quality Analysis

Magpie rates the input quality, i.e., the quality of the initial user prompt in a user–assistant exchange, on a five-point scale ranging from *"very poor"* to *"excellent"*. Specifically, it assesses whether the user query is clearly formulated such that a language model can understand it and generate an appropriate, high-quality response.

### C.4.1 Overall Input Quality Distribution

Overall, both Tulu and SmolTalk consist primarily of high-quality instructions, reflecting the use of capable LLMs during data generation and the application of rigorous quality control procedures.

**Tulu.** Fig. 14a shows the overall distribution of input quality labels across all Tulu samples, while Fig. 14b breaks down the distribution by ST and MT samples. In addition, Table 10 reports the supporting statistics. Tulu is predominantly composed of high-quality inputs, with over 80% of samples rated as either *"excellent"* or *"good"*. This aligns with the strict curation and quality filtering practices described in Lambert et al. [7]. A similar trend holds for ST samples, which make up 95% of the dataset. For MT samples, the distribution is more balanced, with most samples rated as *"good"*, followed by *"excellent"*. However, a non-negligible portion of MT samples (26.5%) fall into the *"poor"* or *"very poor"* categories. As discussed in the main paper, this motivates a rigorous quality filtering step when constructing new data mixtures. This is further supported by our later analysis of instruct reward scores, which reveals a clear correlation between poor input quality and poor response quality.

Table 10: Input quality distribution for the Tulu dataset, shown overall and broken down by single-turn (ST) and multi-turn (MT) samples.

| Input Quality | Total Count | % of All Samples | ST Sample Count | % of ST Samples | MT Sample Count | % of MT Samples |
|---|---|---|---|---|---|---|
| excellent | 572343 | 62.77% | 563307 | 64.69% | 9036 | 22.06% |
| good | 185502 | 20.34% | 171465 | 19.69% | 14037 | 34.27% |
| average | 52925 | 5.80% | 45900 | 5.27% | 7025 | 17.15% |
| poor | 55027 | 6.04% | 48842 | 5.61% | 6185 | 15.10% |
| very poor | 45985 | 5.04% | 41305 | 4.74% | 4680 | 11.42% |

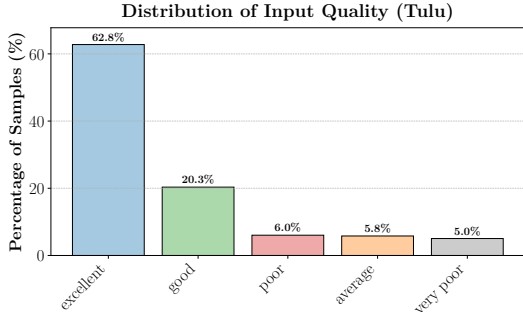
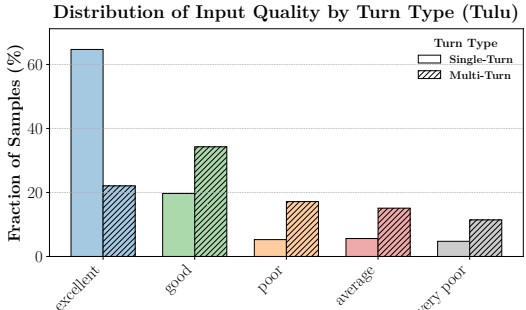

(a) Overall input quality distribution for Tulu: most samples (over 80%) are of *"excellent"* or *"good"* quality, indicating rigorous quality controls during curation.

(b) Input quality distribution by turn type for Tulu: most ST samples are of *"excellent"* quality while most MT samples are of *"good"* quality, followed by *"excellent"*.

Figure 14: Distribution of Magpie input quality labels for Tulu: (a) overall input quality distribution, (b) distribution by turn type (single-turn vs. multi-turn).

**SmolTalk.** Similarly, Fig. 15a shows the overall distribution of input quality labels across all SmolTalk samples and Fig. 15b breaks down the distribution by ST and MT samples, with Table 11 reporting the statistics. SmolTalk also contains predominantly high-quality inputs, with 85% of samples rated as either *"excellent"* or *"good"*. In contrast to Tulu, significantly fewer samples fall into the *"poor"* or *"very poor"* categories, suggesting that Allal et al. [9] applied stricter quality control measures during curation. This trend holds across both ST and MT samples. In particular, the ST subset is even more skewed toward high-quality inputs, with over 95% of ST samples rated as *"excellent"* or *"good"*. These findings indicate that the SmolTalk data curation process results in consistently high-quality samples, across both turn types.

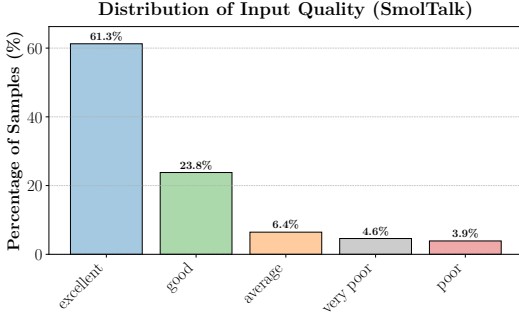
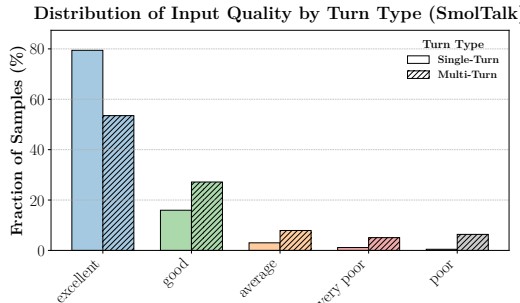

(a) Overall input quality distribution for SmolTalk: most samples (over 85%) are of *"excellent"* or *"good"* quality, indicating even stricter quality controls during dataset curation.

(b) Input quality distribution by turn type for SmolTalk: most ST and MT samples are of *"excellent"* or *"good"* quality, with significantly smaller portions for *"average"*, *"poor"*, and *"very poor"*.

Figure 15: Distribution of Magpie input quality labels for SmolTalk: (a) overall input quality distribution, (b) distribution by turn type (single-turn vs. multi-turn).

Table 11: Input quality distribution for the SmolTalk dataset, shown overall and broken down by single-turn (ST) and multi-turn (MT) samples.

| Input Quality | Total Count | % of All Samples | ST Sample Count | % of ST Samples | MT Sample Count | % of MT Samples |
|---|---|---|---|---|---|---|
| excellent | 187 624 | 61.18% | 72 487 | 79.40% | 115 137 | 53.46% |
| good | 73 108 | 23.84% | 14 370 | 15.74% | 58 738 | 27.27% |
| average | 19 713 | 6.43% | 2 885 | 3.16% | 16 828 | 7.81% |
| poor | 12 049 | 3.93% | 1 133 | 1.24% | 10 916 | 5.07% |
| very poor | 14 166 | 4.62% | 415 | 0.45% | 13 751 | 6.38% |

### C.4.2 Input Quality by Task Category

In addition to the overall input quality analysis, we examine quality distributions across individual Magpie task categories.

**Tulu.** Table 12 presents the input quality breakdown for each task category, and Fig. 16 visualizes the corresponding fractional shares. *Coding & Debugging* (71.2% *"excellent"*), *Data Analysis* (79.3% *"excellent"*), and *Math* (96.0% *"excellent"*) are heavily concentrated in the top quality bin, with negligible low-quality tails. *Editing* (45.1% *"good"*) and *Reasoning* (46.3% *"good"*) skew toward the second-highest bin, yet together still reach close to 80% when combining *"excellent"* and *"good"* labels. *Role Playing* (48.2% *"excellent"*), *Planning* (66.9% *"excellent"*), and *Creative Writing* (46.1% *"excellent"*) show a more balanced distribution: although they lead in combined *"excellent"*+*"good"* ratings, 10–20% of samples fall into *"average"* or worse. *Advice Seeking* peaks in the *"good"* category (29.1%) but also has a sizeable lower-quality tail, with 27.9% of samples rated as *"poor"* or *"very poor"*. *Brainstorming* (26.9% *"very poor"*) and *Others* (41.6% *"very poor"*) exhibit the highest noise levels, with over a quarter of samples rated at the lowest quality tier. These results suggest that open-ended or generative tasks tend to be noisier in Tulu. For downstream modeling and evaluation, filtering to the *"excellent"*+*"good"* subset may improve stability and reduce noise. Consequently, we applied a similar strategy in our data mixture curation recipe.

Table 12: Input quality by Magpie task category for Tulu. Each row reports the proportion of samples rated as *excellent*, *good*, *average*, *poor*, or *very poor* within each task category.

| Task Category | Excellent | Good | Average | Poor | Very Poor |
|---|---|---|---|---|---|
| Advice seeking | 26.1 | 29.1 | 16.9 | 20.3 | 7.6 |
| Brainstorming | 24.2 | 26.8 | 10.2 | 11.9 | 26.9 |
| Coding & Debugging | 71.2 | 21.8 | 3.6 | 2.3 | 1.1 |
| Creative writing | 46.1 | 31.8 | 10.8 | 8.0 | 3.3 |
| Data analysis | 79.3 | 16.3 | 2.8 | 1.3 | 0.4 |
| Editing | 35.4 | 45.1 | 8.7 | 7.1 | 3.8 |
| Information seeking | 33.9 | 36.2 | 10.6 | 9.9 | 9.4 |
| Math | 96.0 | 2.8 | 0.7 | 0.4 | 0.2 |
| Others | 5.3 | 6.4 | 6.7 | 40.0 | 41.6 |
| Planning | 66.9 | 20.0 | 6.8 | 4.8 | 1.6 |
| Reasoning | 30.0 | 46.3 | 10.4 | 7.2 | 6.0 |
| Role playing | 48.2 | 24.8 | 9.3 | 9.9 | 7.7 |

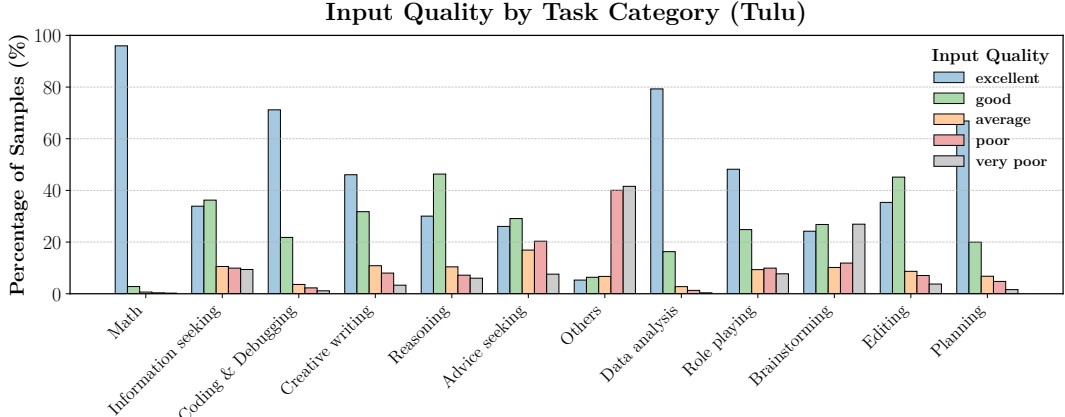

Figure 16: Input quality distribution by Magpie task category for Tulu. STEM-oriented tasks (e.g., *Math*, *Coding*) exhibit predominantly high-quality inputs, while open-ended or generative tasks (e.g., *Brainstorming*, *Advice Seeking*) show greater variability and more frequent low-quality samples.

**SmolTalk.** Table 13 presents the input quality breakdown for each task category, and Fig. 17 visualizes the corresponding fractional shares. *Coding & Debugging* (81.7% *"excellent"*), *Math* (88.2% *"excellent"*), and *Data Analysis / Reasoning* (both around 67% *"excellent"*) are heavily concentrated in the top quality bin, with combined *"excellent"+"good"* shares of 98.0%, 98.0%, and 94.8%, respectively. *Editing* and *Brainstorming* peak in the *"excellent"* bin (39.4% and 51.3%), but also include substantial *"good"* proportions (33.3% and 28.4%), resulting in a combined *"excellent"+"good"* share of 70–80%. *Creative Writing* (44.7% *"excellent"*, 36.3% *"good"*) and *Role Playing* (61.4% *"excellent"*, 21.0% *"good"*) show greater variability, with 15–20% of samples rated as *"average"* or worse. *Advice Seeking* includes 48.3% *"excellent"* and 28.7% *"good"*, but also a non-negligible *"very poor"* fraction (12.5%), indicating some noisy or ill-formed queries. *Planning* shows the largest *"very poor"* tail (13.6%), while *Editing* and *Information Seeking* have the highest combined share of *"poor"+"very poor"* ratings (17.0%), suggesting these categories include more problematic inputs. Overall, these results indicate that closed-form and structured tasks (e.g., *Math*, *Coding*, *Data analysis*) yield the highest input quality, whereas open-ended or generative tasks remain more prone to noise, even in SmolTalk. As with Tulu, filtering for the combined *"excellent"+"good"* subset may improve downstream model stability.

Table 13: Input quality percentages by Magpie task category for SmolTalk. Each row reports the proportion of samples rated as *excellent*, *good*, *average*, *poor*, or *very poor* within each task category.

| Task Category | Excellent | Good | Average | Poor | Very Poor |
|---|---|---|---|---|---|
| Advice seeking | 48.3 | 28.7 | 7.5 | 3.0 | 12.5 |
| Brainstorming | 51.3 | 28.4 | 9.8 | 3.7 | 6.8 |
| Coding & Debugging | 81.7 | 16.4 | 1.4 | 0.4 | 0.1 |
| Creative writing | 44.7 | 36.3 | 16.1 | 2.3 | 0.6 |
| Data analysis | 66.6 | 28.0 | 4.3 | 0.8 | 0.3 |
| Editing | 39.4 | 33.3 | 10.3 | 13.2 | 3.8 |
| Information seeking | 43.4 | 31.2 | 8.7 | 6.1 | 10.6 |
| Math | 88.2 | 9.9 | 1.2 | 0.5 | 0.2 |
| Others | 56.3 | 29.9 | 6.6 | 2.4 | 4.9 |
| Planning | 43.8 | 26.8 | 11.6 | 4.3 | 13.6 |
| Reasoning | 67.0 | 25.2 | 4.6 | 2.1 | 1.1 |
| Role playing | 61.4 | 21.0 | 7.8 | 7.4 | 2.4 |

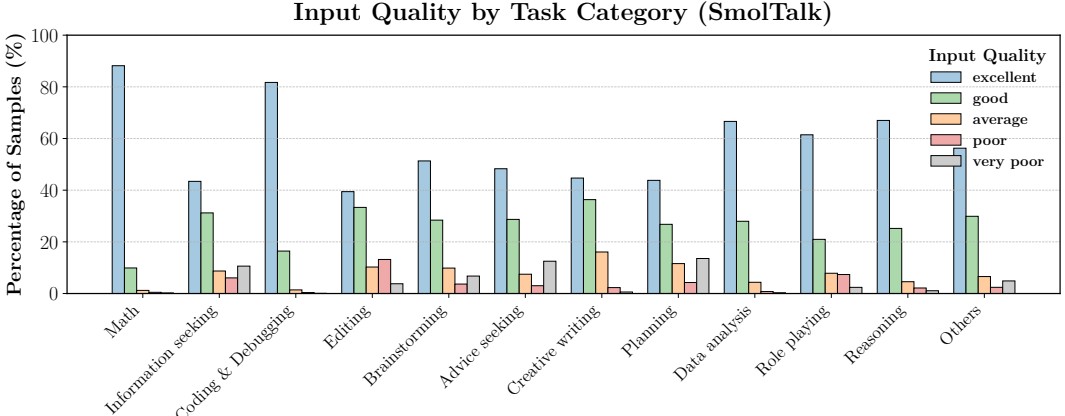

Figure 17: Input quality distribution by Magpie task category for SmolTalk. STEM-oriented tasks (e.g., *Math*, *Coding*) exhibit predominantly high-quality inputs, while open-ended or generative tasks (e.g., *Brainstorming*, *Advice Seeking*) show greater variability and more frequent low-quality samples.

## C.5 Response Quality Analysis

Magpie rates the response quality, i.e., the quality of the assistant's response to a user prompt, as the *instruct reward* via a language model-based reward model. Specifically, it employs *FsfairX-LLaMA3-RM-v0.1* [70–72], a Llama-3-8B-Instruct-based reward model that assigns a continuous-valued score to each response. Since the original implementation supports only single-turn conversations, we extend the reward annotation pipeline to handle multi-turn samples by using a separate Llama-3.3-70B-Instruct-based LLM-as-a-Judge to evaluate MT responses on a scale between 0-5. We refer to more details in App. C.

### C.5.1 Single-Turn Reward Distributions

**Tulu.** Fig. 18 shows the distribution of instruct reward scores by task category for Tulu's ST samples. The distribution is roughly bell-shaped, with most scores falling in the range of [–6, +6] and a peak density between +1 and +2. Notably, *Math* and *Coding & Debugging* receive almost exclusively non-negative scores, peaking around +1 to +3, reflecting clear, well-structured prompts with high response quality. *Information Seeking*, *Reasoning*, and *Data Analysis* are centered closer to zero, with modest tails extending into negative reward regions. In contrast, *Advice Seeking*, *Brainstorming*, *Creative Writing*, and *Others* exhibit heavy left tails (extending to –12), indicating many low-quality or poorly answered prompts. These observations suggest that filtering samples by a reward threshold (e.g., $\geq 0$ or $\geq 1$) may yield a cleaner, high-quality single-turn subset dominated by *Math* and *Coding* tasks, while low-reward examples (e.g., $\leq -3$) can help identify problematic, open-ended prompts for further curation. We directly incorporate these insights into our dataset curation recipe (see App. D).

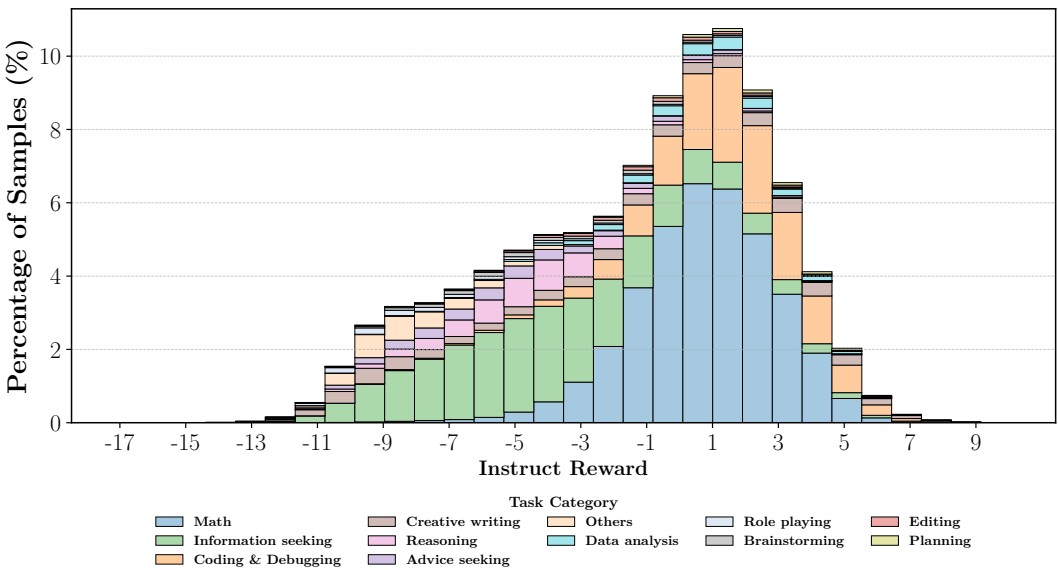

Figure 18: Distribution of single-turn instruct reward scores in the Tulu dataset, broken down by Magpie task category. STEM-oriented tasks (e.g., *Math*, *Coding & Debugging*) receive predominantly non-negative scores, reflecting high-quality, well-answered prompts. In contrast, open-ended and creative tasks (e.g., *Creative Writing*, *Advice Seeking*) exhibit heavier negative tails, indicating a higher prevalence of low-quality or poorly answered examples.

**SmolTalk.** Fig. 19 shows the distribution of instruct reward scores by task category for SmolTalk's ST samples. The distribution is approximately bell-shaped, spanning the range [–9, +7], with the highest density around +1. *Coding & Debugging* receives almost exclusively non-negative scores, peaking between +1 and +3, indicating well-posed prompts that the model handles reliably. *Math* is more evenly distributed around zero, suggesting greater variation in prompt clarity or complexity. *Information Seeking*, *Reasoning*, and *Data Analysis* are similarly centered near zero, with modest negative tails extending to –5, reflecting mixed response quality across these domains. *Advice Seeking*, *Brainstorming*, *Creative Writing*, and *Others* exhibit heavier left tails reaching down to –7, suggesting a greater fraction of underspecified or incoherent prompts. These trends again suggest that filtering samples by reward thresholds (e.g., $\geq 0$ or $\geq 1$) can yield a high-quality subset dominated by *Math* and *Coding* tasks, while low-reward samples (e.g., $\leq -3$) highlight problematic open-ended prompts that may benefit from further curation. We elaborate on these filtering strategies in App. D.

Figure 19: Distribution of single-turn instruct reward scores in the SmolTalk dataset, broken down by Magpie task category. STEM-oriented tasks (e.g., *Math*, *Coding & Debugging*) receive predominantly non-negative scores, indicating reliable and well-structured prompts. Open-ended and creative tasks (e.g., *Creative Writing*, *Advice Seeking*) exhibit similar distributions with slightly broader negative tails, suggesting a slightly higher prevalence of underspecified or incoherent prompts.

### C.5.2 Multi-Turn Reward Distributions

**Tulu.** Fig. 20 shows the distribution of instruct reward scores by task category for Tulu's MT samples. Nearly 90% of MT samples receive the maximum reward score of 5, approximately 8% land at 4, and the remaining 2% are scattered across scores 0–3. *Information Seeking* alone accounts for roughly 35% of the reward-5 bin, followed by *Creative Writing* at around 15% and *Coding & Debugging* at 12–13%. All other categories (e.g., *Advice Seeking*, *Role Playing*, *Math*) each contribute between 2–8% of that top bin. The ∼8% of samples rated at reward 4 exhibit a similar task distribution, with *Information Seeking*, *Coding & Debugging*, and *Creative Writing* again leading, though categories such as *Advice Seeking* and *Role Playing* are relatively over-represented compared to the reward-5 group. Scores ≤ 3 are vanishingly rare. When they do occur, they disproportionately stem from open-ended categories such as *Advice Seeking*, *Creative Writing*, and *Role Playing*, likely reflecting occasional multi-turn drift or incoherence. These observations indicate that MT conversations are overwhelmingly rated *"excellent"* by the reward model, especially for structured tasks like *Information Seeking*, *Coding*, and *Math*. Creative and advisory interactions, while still high-quality on average, account for the largest share of samples in the reward-4 bin and the only non-zero mass below 4, suggesting that these task types may benefit from additional filtering.

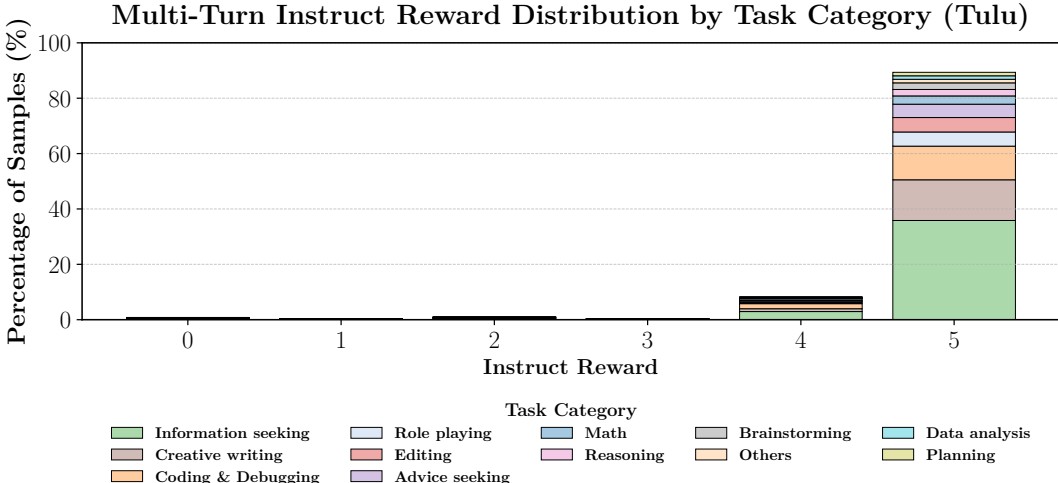

Figure 20: Distribution of multi-turn instruct reward scores in the Tulu dataset, broken down by Magpie task category. The vast majority of samples receive the maximum score of 5, with structured tasks such as *Information Seeking*, *Coding & Debugging*, and *Math* dominating this top bin. Creative and open-ended tasks (e.g., *Creative Writing*, *Advice Seeking*, *Role Playing*) are over-represented in the small mass at score 4 and account for nearly all samples scoring below 4, highlighting them as key targets for further quality filtering.

**SmolTalk.** Fig. 21 shows the distribution of instruct reward scores by task category for SmolTalk's MT samples. As with Tulu, nearly 90% of MT samples receive the maximum reward score of 5, approximately 8% score 4, and the remaining 2% fall into bins 0–3. *Information Seeking* contributes the largest share, accounting for roughly 22–23% of the reward-5 bin. *Editing* follows with around 14%, and *Math* with approximately 10%. *Advice Seeking*, *Coding & Debugging*, and *Brainstorming* each make up about 8–9%. *Creative Writing*, *Planning*, and *Data Analysis* contribute mid-single-digit proportions (5–7%), while *Reasoning* and *Role Playing* round out the top bin with around 4–5% each. The samples scoring 4 (roughly 8% of total) broadly mirror the top-bin rankings, though open-ended tasks such as *Advice Seeking* and *Creative Writing* are slightly more prominent. Scores ≤ 3 are extremely rare (fewer than 2% overall), and when present, are disproportionately drawn from open-ended categories such as *Advice Seeking*, *Creative Writing*, and *Role Playing*, similarly reflecting occasional context drift or incoherence. These findings indicate that MT conversations in SmolTalk, as in Tulu, are overwhelmingly rated *"excellent"* by the reward model, particularly for structured tasks such as *Information Seeking* and *Math*. Open-ended interactions, while still achieving high scores on average, represent the only meaningful mass below 4, suggesting these categories may benefit from further quality filtering as well.

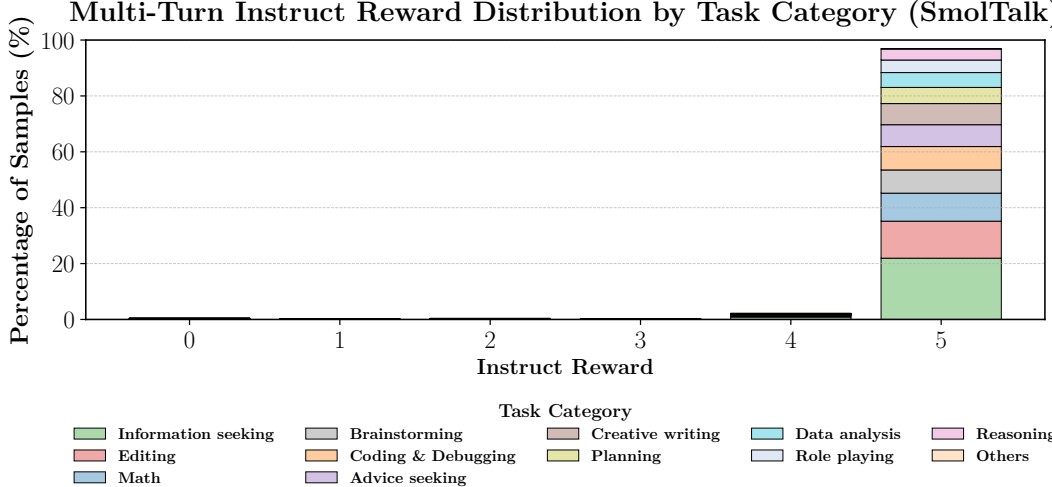

Figure 21: Distribution of multi-turn instruct reward scores in the SmolTalk dataset, broken down by Magpie task category. Most samples receive the maximum score of 5, with structured tasks such as *Information Seeking* and *Math*, but also *Editing* dominating the top bin. Open-ended tasks (e.g., *Advice Seeking*, *Creative Writing*, *Role Playing*) contribute more substantially to lower reward scores, including the small but non-zero mass in the 0–3 range, suggesting they may benefit from further curation or quality filtering.

### C.5.3 Instruct Reward vs. Input Quality

In this section, we investigate the relationship between input quality and instruct reward to validate the intuition that poorly formulated user prompts often lead to suboptimal assistant responses.

**Single-Turn Case.** Fig. 22 and Fig. 23 show the distribution of single-turn reward scores across task categories, grouped by input quality label, for both Tulu and SmolTalk datasets. In general, single-turn instruct rewards exhibit a strong correlation with input quality. *Excellent* prompts yield reward scores mostly in the +1 to +5 range, peaking around +2. *Good* prompts are centered between –5 and +1. *Average* inputs result in rewards concentrated around –7 to –3, while *Very Poor* prompts are heavily skewed toward –11 to –7. These patterns confirm that input quality is a strong predictor of response quality for the majority of single-turn samples for both datasets.

**Multi-Turn Case.** Fig. 24 and Fig. 25 show the distribution of multi-turn reward scores across task categories, grouped by input quality label, for both Tulu and SmolTalk datasets. While single-turn rewards provide a clear, continuous signal of prompt quality, multi-turn reward scores are heavily saturated at the maximum value (5) for both Tulu and SmolTalk, resulting in a loss of discriminative power across input quality strata. A closer inspection of our annotated data samples reveals why this occurs. Below are two illustrative cases where the input quality is rated as *"poor"*, yet the overall multi-turn conversation still receives a perfect instruct reward of 5.

**Example 1: Assistant starts first and initial prompt has typos.**

```
'conversations': [
  {'from': 'gpt', 'value': 'Jhonny is a chatbot that flirts with the user
                            using Johnny Bravo quotes'},
  {'from': 'human', 'value': "How's you're day going?"},
  {'from': 'gpt', 'value': 'Great, But enough about me... Let's talk about
                            me. What do you think of me?'},
  {'from': 'human', 'value': "I think you're pretty neat!"},
  {'from': 'gpt', 'value': '...'},
  ...
]
```

In this example, two factors contribute to the low input quality rating: (1) The conversation begins with the assistant rather than the user, due to the inclusion of a system-style prompt, something that the Magpie input-quality rater penalizes. (2) The initial prompt includes a typo (*"Jhonny"* instead of *"Johnny"*), which also contributes to a lower input quality score. Despite these, the assistant's behavior remains consistent and coherent, resulting in a perfect reward by imitating Johnny Bravo.

**Example 2: Vague user query is clarified later.**

```
'conversations': [
  {'from': 'human', 'value': 'Can you explain me how cheats are working?'},
  {'from': 'gpt', 'value': 'Cheating can be of various types. I will need
                            more information on what type of cheating you
                            are referring to before I can provide you with
                            any information.'},
  {'from': 'human', 'value': 'How do cheats in games like COD and CS:GO
                              work? What goes into them? Please discuss
                              the ethics of cheats as well.'},
  {'from': 'gpt', 'value': 'Cheats in games like Call of Duty (COD) and
                            Counter-Strike: Global Offensive (CS:GO)
                            typically work by manipulating the game\'s
                            code to gain an unfair advantage...'}
]
```

In this case, the first user input is vague and could improve from better grammar, leading to a low input quality rating by Magpie. However, the user clarifies their request in the following turns. The assistant responds appropriately and in detail, producing a coherent and informative answer by the end of the conversation. This illustrates how multi-turn interactions can recover from poor initial queries, yielding high-quality final responses despite the low initial rating.

Together, these examples highlight a key distinction: multi-turn conversations may achieve high reward scores even when the initial input is of poor quality, particularly when clarification or intent refinement occurs across turns. As a result, input quality is less predictive of instruct reward in the multi-turn case than in the single-turn setting. This observation initially motivated the development of a dedicated multi-turn annotation pipeline for Magpie, ensuring that reward labels reflect the quality of the entire conversation rather than just the initial prompt.

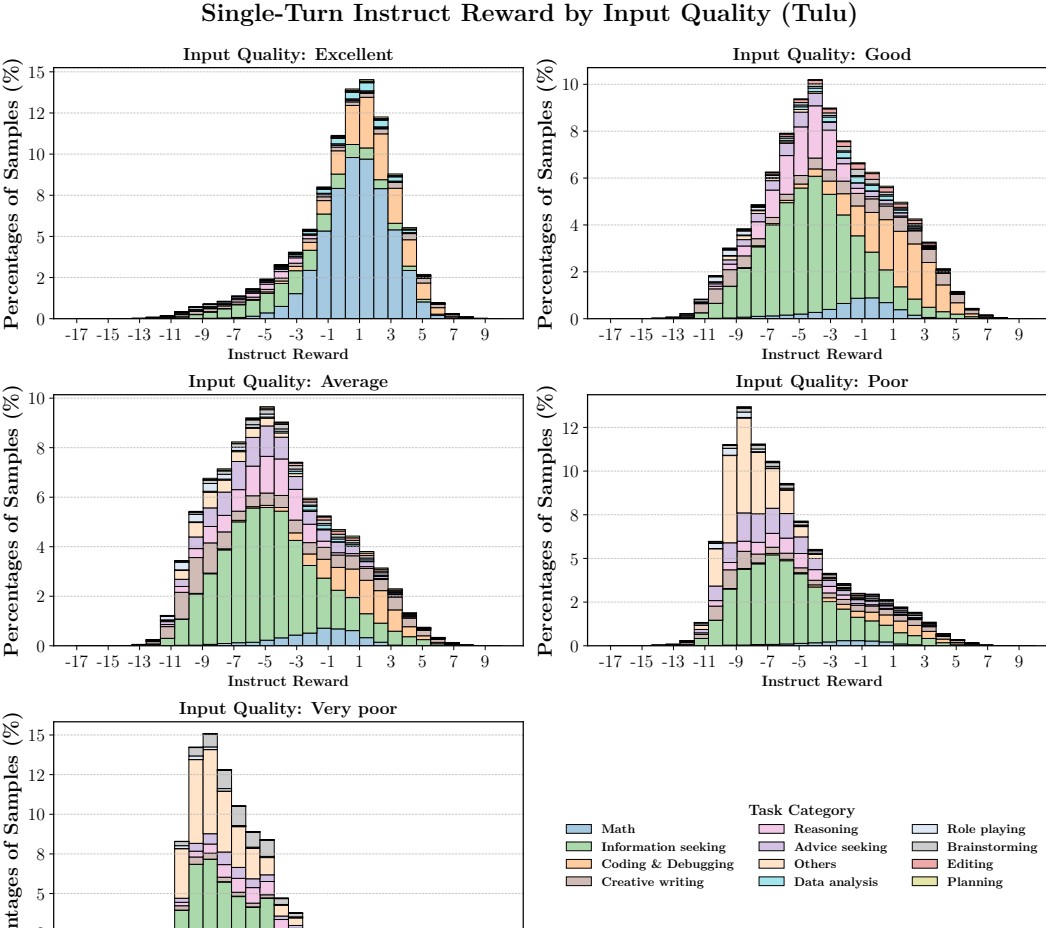

Figure 22: Distribution of single-turn instruct reward scores by input quality label in the Tulu dataset. Higher-quality prompts (*excellent*, *good*) correspond to significantly higher reward scores, while lower-quality inputs (*average*, *poor*, *very poor*) are associated with markedly lower rewards. This confirms a strong correlation between input quality and response quality in the single-turn setting.

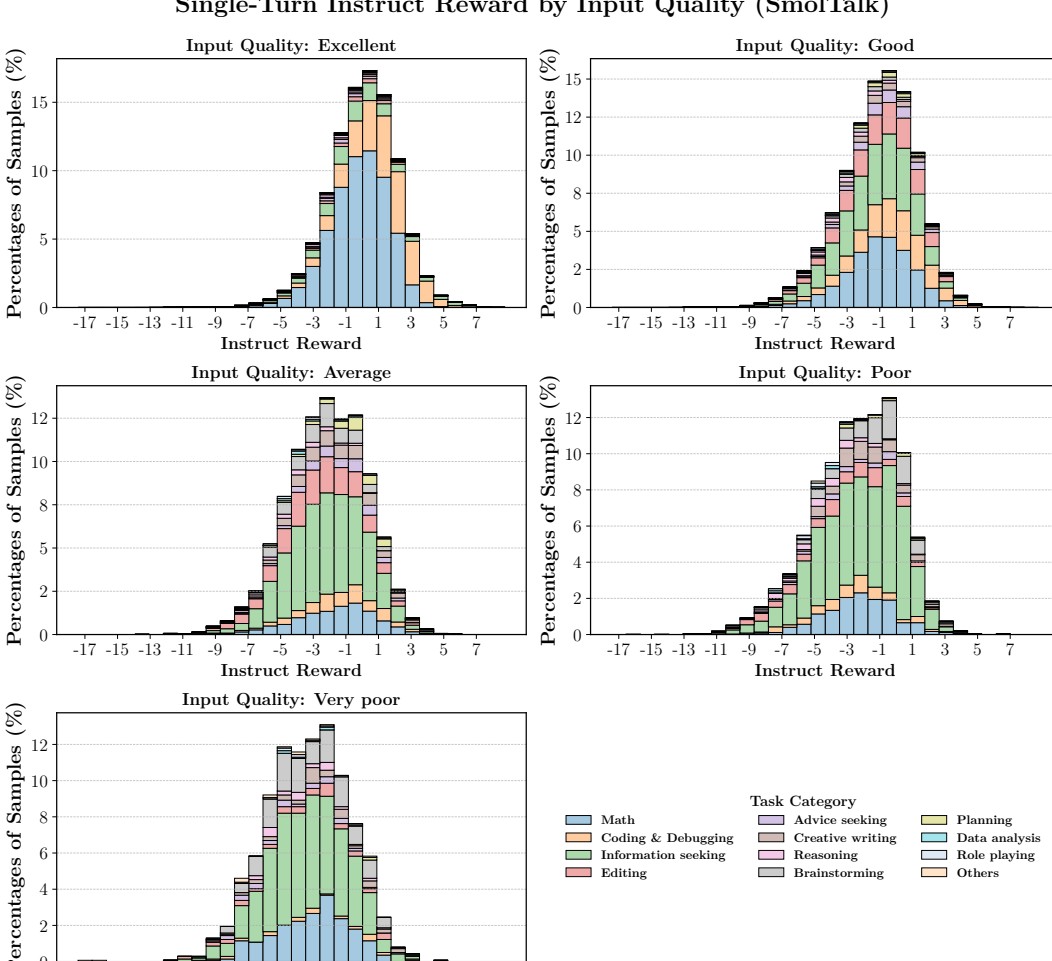

Figure 23: Distribution of single-turn instruct reward scores by input quality label in the SmolTalk dataset. As with Tulu, higher input quality is strongly correlated with higher instruct reward, demonstrating that prompt clarity and specificity are key drivers of response quality in the single-turn setting.

Figure 24: Distribution of multi-turn instruct reward scores by input quality label in the Tulu dataset. Most samples, regardless of input quality, receive the maximum reward score of 5, suggesting that clarification across turns often compensates for initially vague or low-quality prompts.

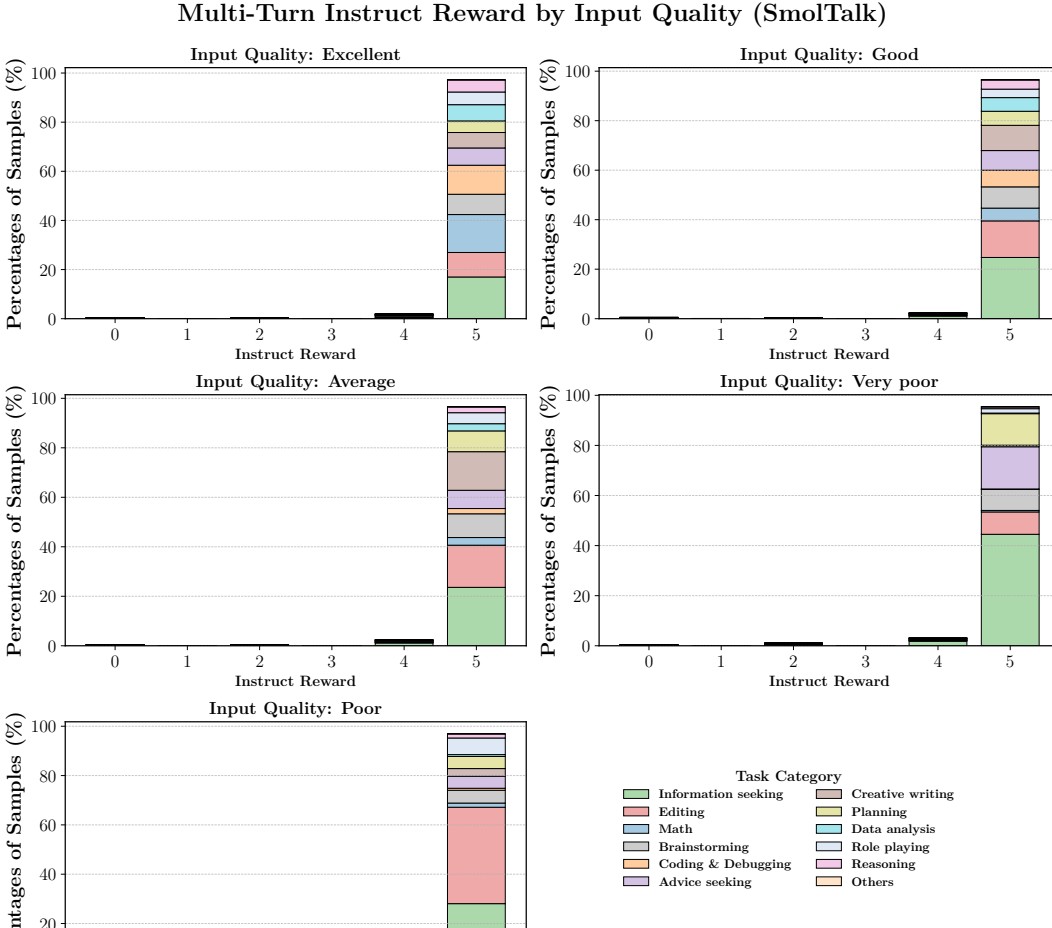

Figure 25: Distribution of multi-turn instruct reward scores by input quality label in the SmolTalk dataset. Similar to Tulu, the reward model heavily favors MT conversations with perfect scores, even for lower-rated prompts, reflecting the tendency of multi-turn dialogues to recover from poor initial queries through user clarification.

## C.6 Difficulty Analysis

In this section, we analyze how Magpie's difficulty labels are distributed across task categories for the Tulu and SmolTalk datasets. We include both overall and turn-type-specific analyses.

### C.6.1 Overall Distribution

Fig. 26 and Fig. 27 show the relative difficulty distribution per task category for Tulu and SmolTalk. Each bar reflects the percentage of a task's samples that fall into each difficulty bin.

**Tulu.** *Math* dominates the *easy*, *hard*, and *very hard* bins, accounting for 45%, 37%, and 36% of the samples in those categories, respectively. *Information Seeking* peaks at *very easy* (44%), and remains substantial in both *easy* (28%) and *medium* (32%). *Coding & Debugging* is spread across all difficulty levels: approximately 17% in *easy*, 22% in *medium*, 12% in *hard*, and 14% in *very hard*. It is broadly represented but does not dominate any particular bin. Other categories, including *Creative Writing*, *Role Playing*, and *Advice Seeking*, each account for no more than 10% of any difficulty bin. This distribution suggests that fact-based tasks (e.g., *Math*, *Information Seeking*, *Coding*) dominate mid-to-lower difficulty levels, while creative and advisory tasks remain relatively underrepresented across all levels.

**SmolTalk.** *Math* similarly dominates the *hard* and *very hard* bins, contributing approximately 28% and 47% of the samples, respectively. *Information Seeking* peaks at *very easy* (35%) and remains substantial in *easy* (25%), while being evenly represented across other bins as well. *Coding & Debugging* is fairly balanced across difficulty levels, showing no strong concentration at either extreme. All other task categories, including *Creative Writing*, *Role Playing*, and *Advice Seeking*, remain minor contributors with ≤10% in any bin. These patterns suggest that again fact-based tasks, especially *Math*, skew toward mid-to-high difficulty, whereas creative and advisory tasks occur infrequently and are less likely to be rated as difficult.

Figure 26: Distribution of difficulty ratings by task category for Tulu. Each bar shows the relative difficulty composition within a task.

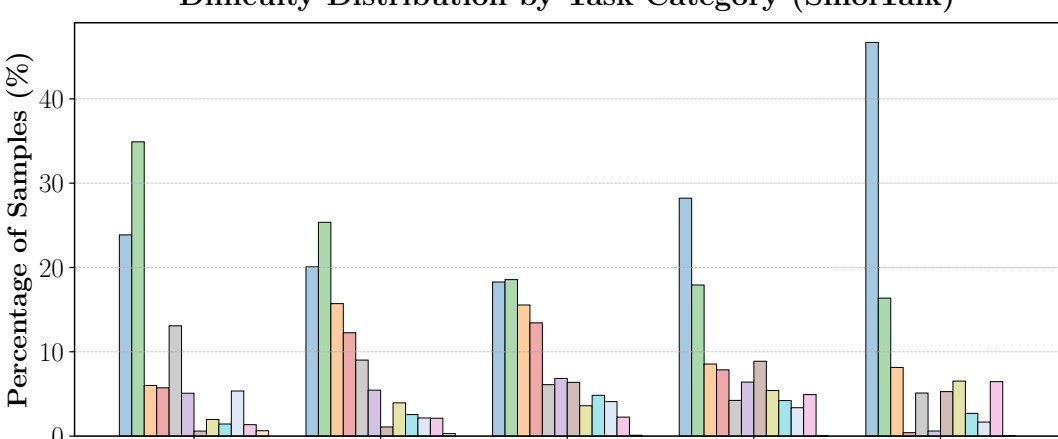

Figure 27: Distribution of difficulty ratings by task category for SmolTalk. Each bar shows the relative difficulty composition within a task.

### C.6.2 Single-Turn

Fig. 28 and Fig. 29 show the relative difficulty distribution per task category for single-turn samples in Tulu and SmolTalk. Each bar reflects the percentage of a task's single-turn samples that fall into each difficulty bin.

**Tulu.** In the single-turn setting, *Math* continues to dominate the *easy* (approximately 46%) and *hard* (38%) bins, while *Information Seeking* peaks in the *very easy* bin (44%) and also contributes around 13% to *very hard*. *Coding & Debugging* is distributed across *easy* (18%), *medium* (22%), and *hard* (12%), maintaining a consistent presence across difficulty levels. *Creative Writing* and *Role Playing* remain underrepresented in the lower difficulty bins but rise to 10–12% in the *very hard* bin. Given Tulu's predominantly single-turn nature, it is unsurprising that fact-based tasks dominate the lower difficulty levels, while creative and open-ended tasks contribute disproportionately to the hardest examples.

**SmolTalk.** In SmolTalk, *Math* shows strong representation across all difficulty levels, from *very easy* to *very hard*, highlighting its prominence in single-turn problem–solution prompts. *Coding & Debugging* is also evenly distributed across the difficulty bins, showing no strong skew. Other categories such as *Information Seeking* and *Editing* appear broadly stratified as well, without any pronounced concentration, suggesting a relatively uniform difficulty distribution across task types.

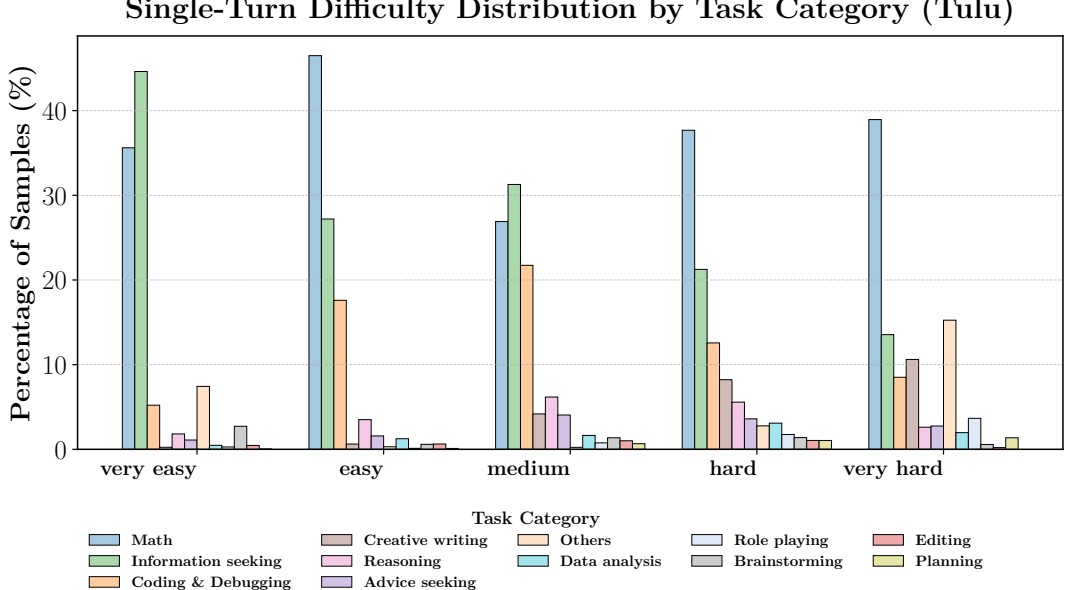

Figure 28: Difficulty distribution by task category for single-turn samples in Tulu. Each bar shows the relative proportion of difficulty labels within each task. Fact-based tasks dominate lower difficulty levels, while creative and role-based prompts appear more frequently in the hardest bins.

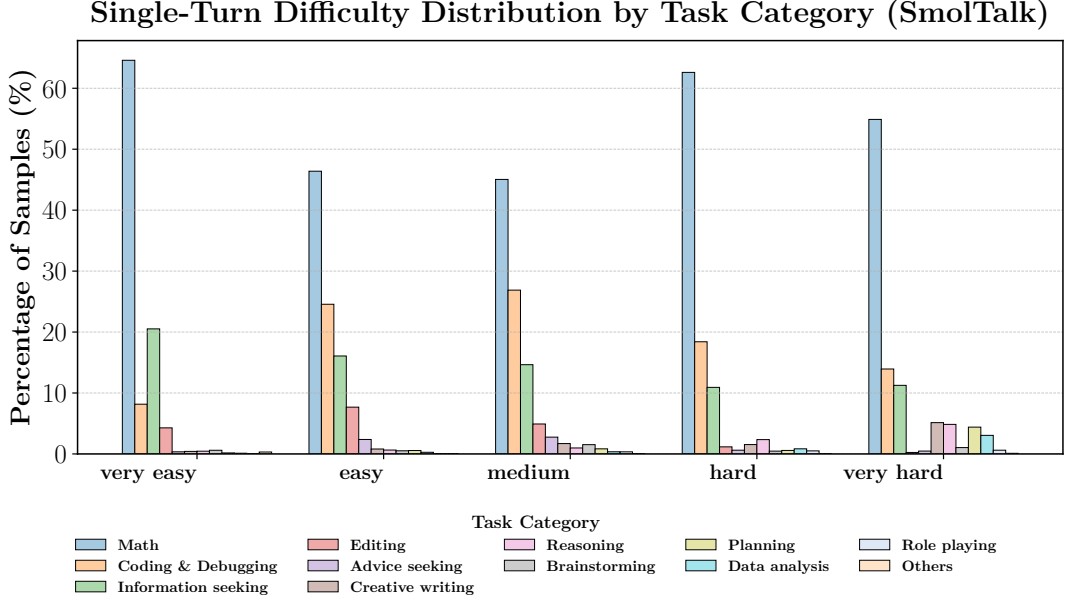

Figure 29: Difficulty distribution by task category for single-turn samples in SmolTalk. *Math* and *Coding & Debugging* are consistently present across all difficulty bins, while other task types remain evenly stratified with no strong concentration.

### C.6.3 Multi-Turn

Fig. 30 and Fig. 31 show the relative difficulty distribution per task category for multi-turn samples in Tulu and SmolTalk. Each bar reflects the percentage of a task's multi-turn samples that fall into each difficulty bin.

**Tulu.** In the multi-turn setting, *Information Seeking* becomes even more dominant, comprising approximately 53% of *easy*, 47% of *medium*, and 48% of *very easy* samples. *Coding & Debugging* and *Math* together account for 25–35% of samples across all difficulty bins, with *Math* slightly more prevalent in the *hard* and *very hard* categories. *Creative Writing* and *Role Playing* each contribute around 18% to the *very hard* bin, indicating that these open-ended multi-turn dialogues pose significant challenges. Overall, multi-turn conversations in Tulu are heavily concentrated on information-seeking tasks, while creative and role-based categories contribute more prominently to the high-difficulty tail than in the single-turn or overall distributions.

**SmolTalk.** *Information Seeking* is again highly prevalent, contributing approximately 43% of *very easy*, 30% of *easy*, and 24% of *very hard* multi-turn samples. *Math* continues to dominate the *very hard* category. *Coding & Debugging* is evenly distributed across the lower difficulty bins (*very easy* to *hard*), but has almost no presence in the *very hard* bin. *Brainstorming* is fairly evenly represented across all difficulty levels, likely due to its inherently interactive and exploratory nature in multi-turn settings. Overall, SmolTalk's multi-turn conversations remain focused on fact-based tasks, especially *Math*, while open-ended tasks contribute more sparsely and with a wider difficulty spread.

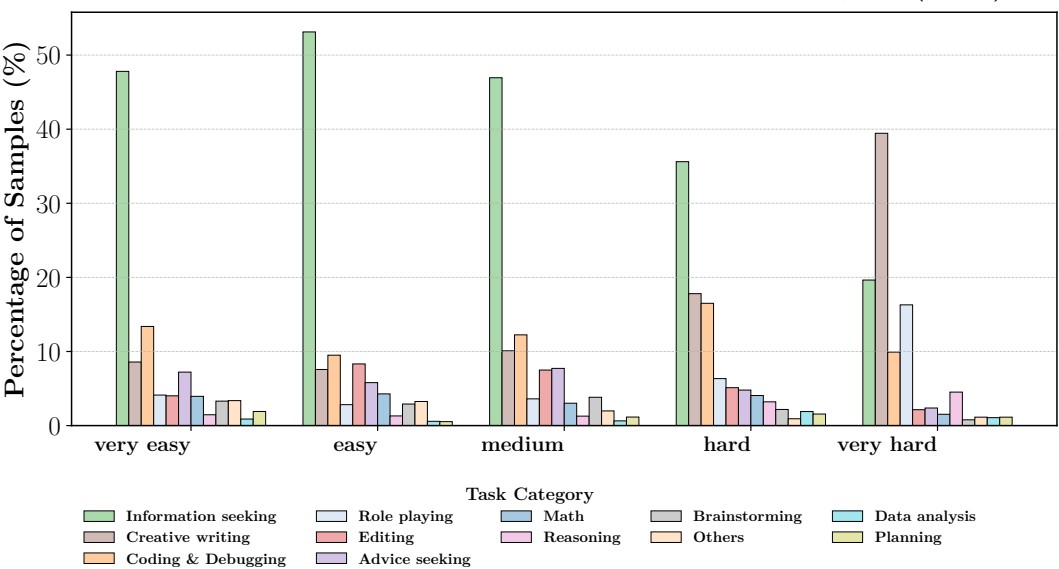

Figure 30: Difficulty distribution by task category for multi-turn samples in Tulu. *Information Seeking* dominates the easier bins, while open-ended tasks such as *Creative Writing* and *Role Playing* contribute substantially to the *very hard* bin.

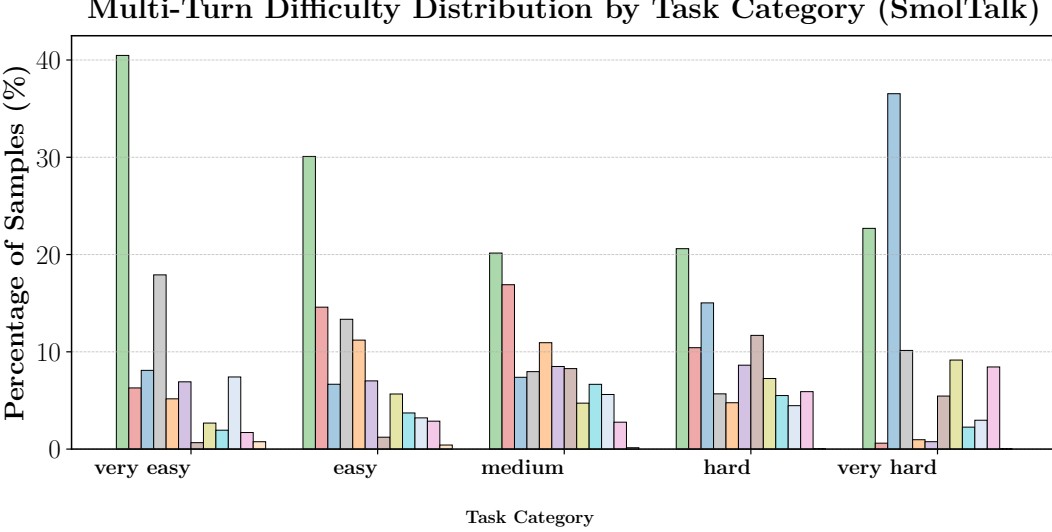

Figure 31: Difficulty distribution by task category for multi-turn samples in SmolTalk. *Math* and *Information Seeking* are prominent across all bins, while categories like *Brainstorming* and *Coding & Debugging* are more evenly spread across difficulty levels.

### C.6.4 Instruction Reward vs. Difficulty

To assess the relationship between task difficulty and response quality, Fig. 32, Fig. 33, Fig. 34, and Fig. 35 show instruct reward distributions for single-turn and multi-turn samples in the Tulu and SmolTalk datasets, respectively. Overall, we observe that difficulty has only a negligible effect on reward distribution, altering the spread and shape only marginally. As such, we do not consider difficulty a key optimization lever in our dataset curation recipe.

**Tulu.** For Tulu, the instruct reward distribution remains largely stable across difficulty levels in both single-turn and multi-turn settings. In the single-turn case, the overall reward distribution is consistent across difficulties, ranging from approximately –14 to +7, with only slightly clearer separation between low- and high-reward samples in the *very hard* bin (see Fig. 32b). In the multi-turn case, the reward values themselves remain unchanged and only the distribution of task categories within difficulty bins varies slightly (see Fig. 33a and Fig. 33b). Thus, unlike input quality, difficulty shows minimal predictive power over response quality for Tulu.

**SmolTalk.** Similarly, for SmolTalk, the reward distributions are largely invariant to difficulty levels. In the single-turn case, the overall reward range stays consistent, though for very hard samples, the peak shifts slightly from around +1 to 0 (see Fig. 34a and Fig. 34b). In the multi-turn case, as with Tulu, reward scores remain constant, with changes only in the composition of underlying task categories (see Fig. 35a and Fig. 35b). In summary, difficulty annotations do not significantly impact the reward model's output, and thus play only a minor role in shaping response quality in SmolTalk.

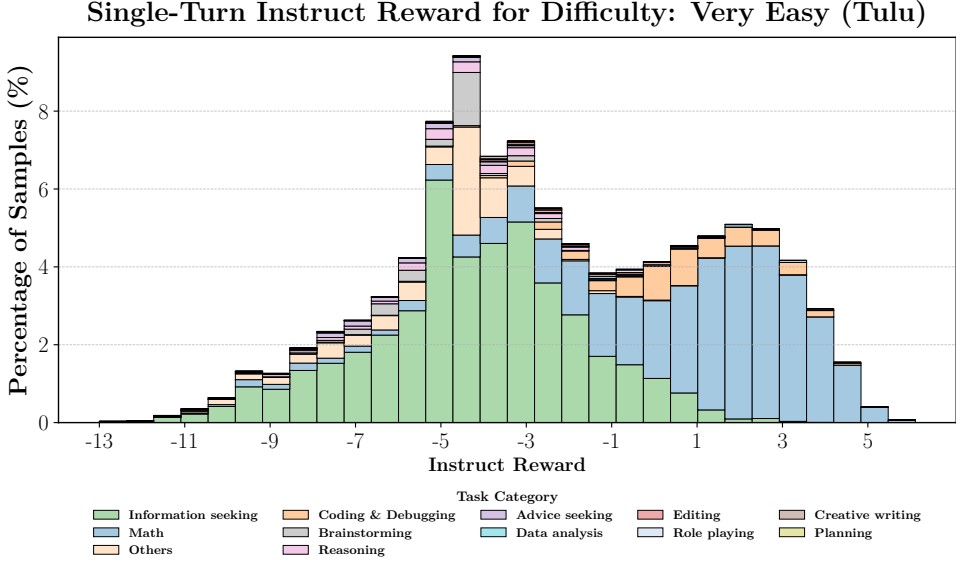

(a) Single-turn instruct reward distribution for very easy samples in Tulu.

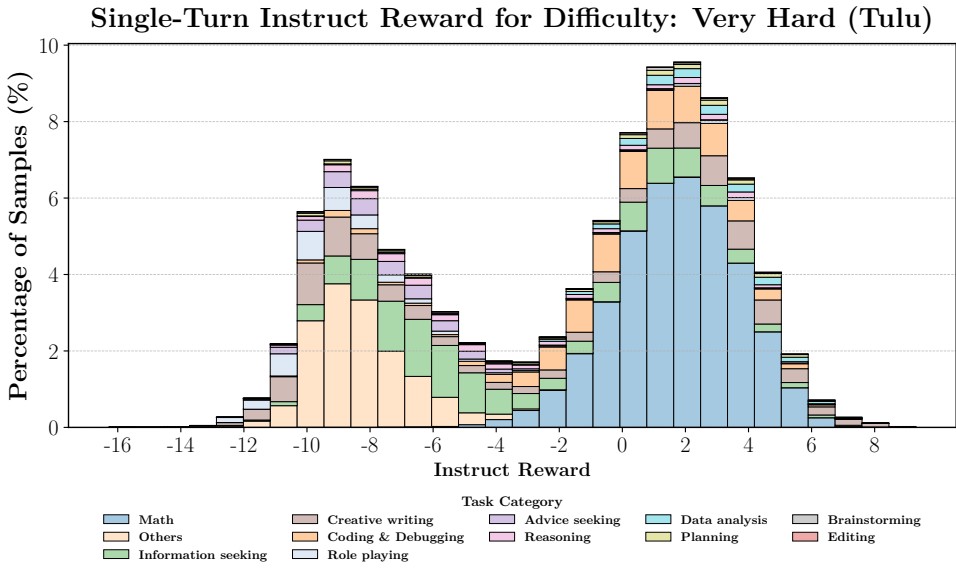

(b) Single-turn instruct reward distribution for very hard samples in Tulu.

Figure 32: Instruct reward distribution by difficulty level for single-turn samples in Tulu. The overall shape remains consistent, with slightly clearer separation of low and high rewards in very hard samples.

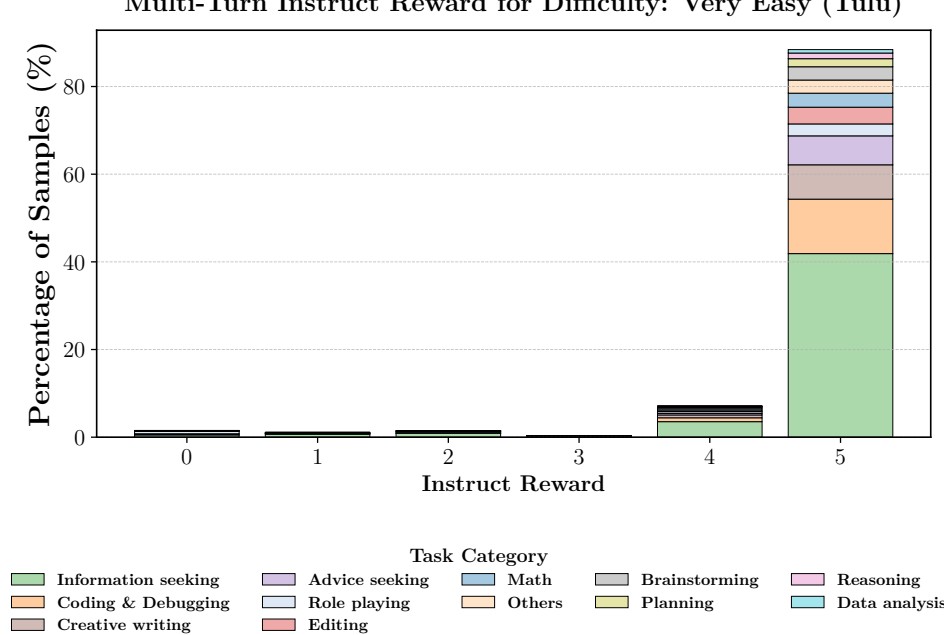

(a) Multi-turn instruct reward distribution for very easy samples in Tulu.

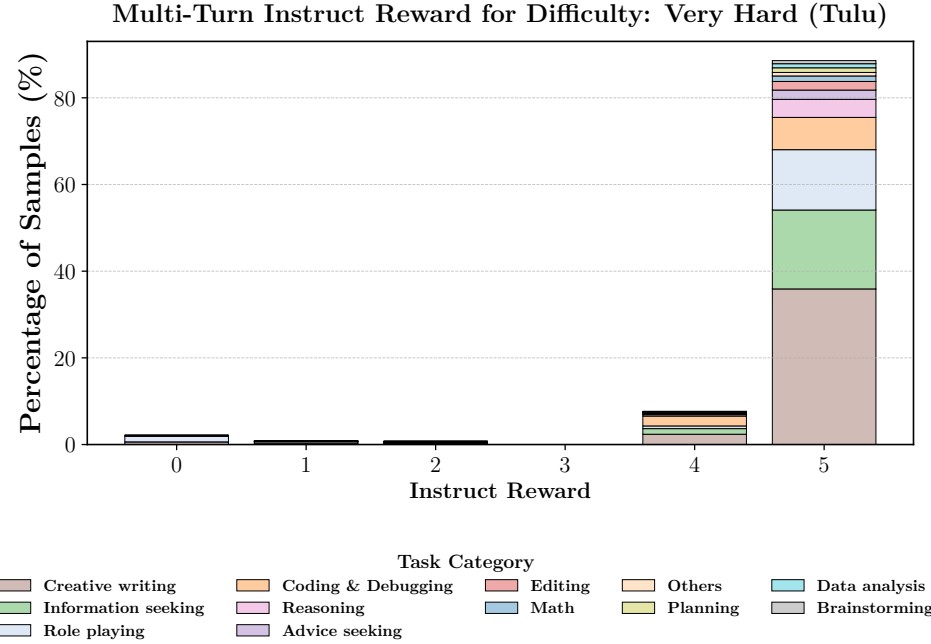

(b) Multi-turn instruct reward distribution for very hard samples in Tulu.

Figure 33: Instruct reward distribution by difficulty level for multi-turn samples in Tulu. Reward values remain saturated at 5 and only task composition within bins changes.

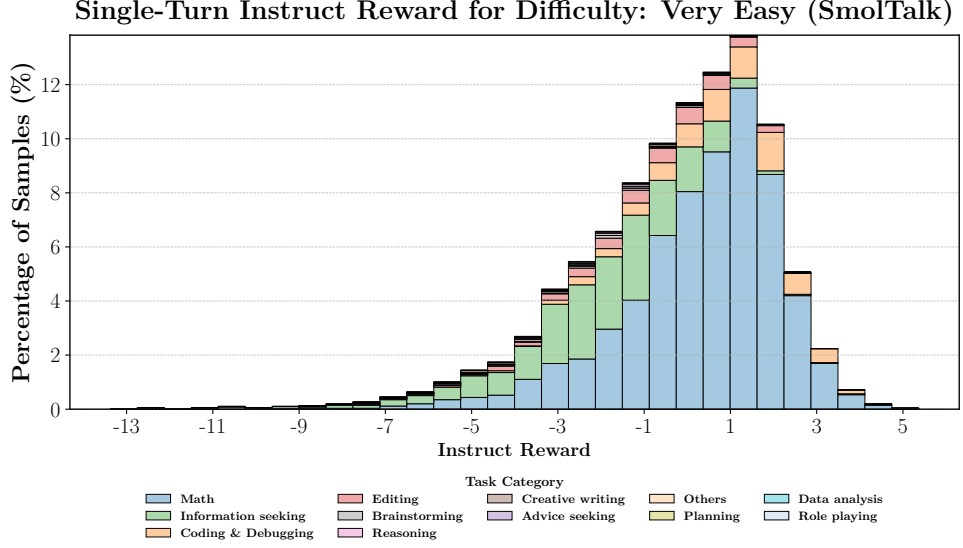

(a) Single-turn instruct reward distribution for very easy samples in SmolTalk.

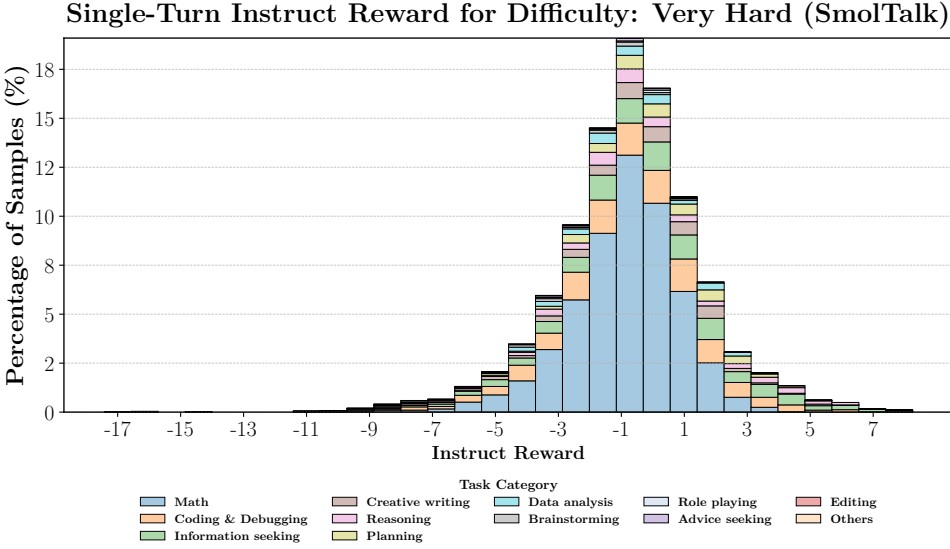

(b) Single-turn instruct reward distribution for very hard samples in SmolTalk.

Figure 34: Instruct reward distribution by difficulty level for single-turn samples in SmolTalk. Reward ranges remain stable, with a slight leftward shift for very hard prompts.

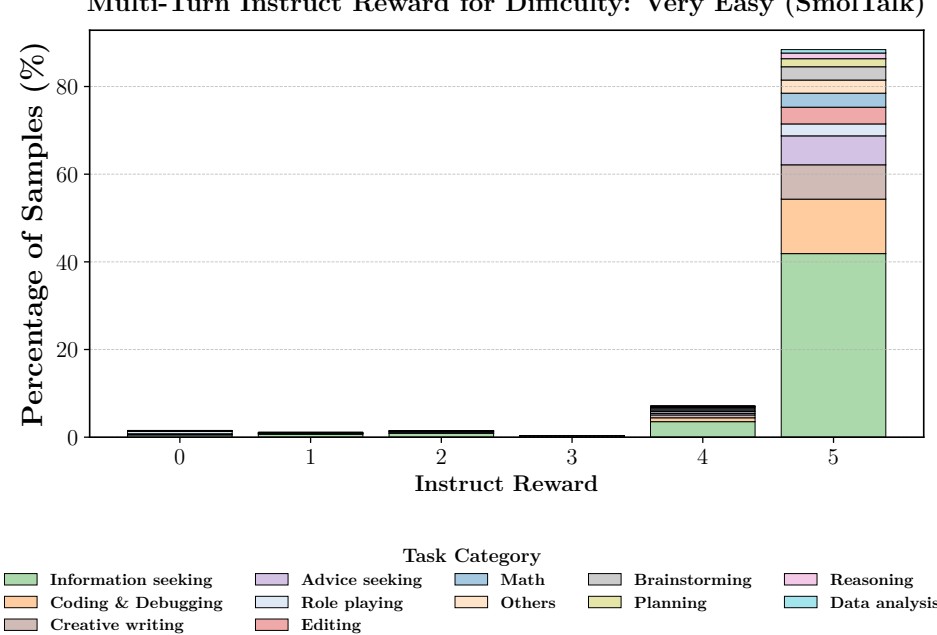

(a) Multi-turn instruct reward distribution for very easy samples in SmolTalk.

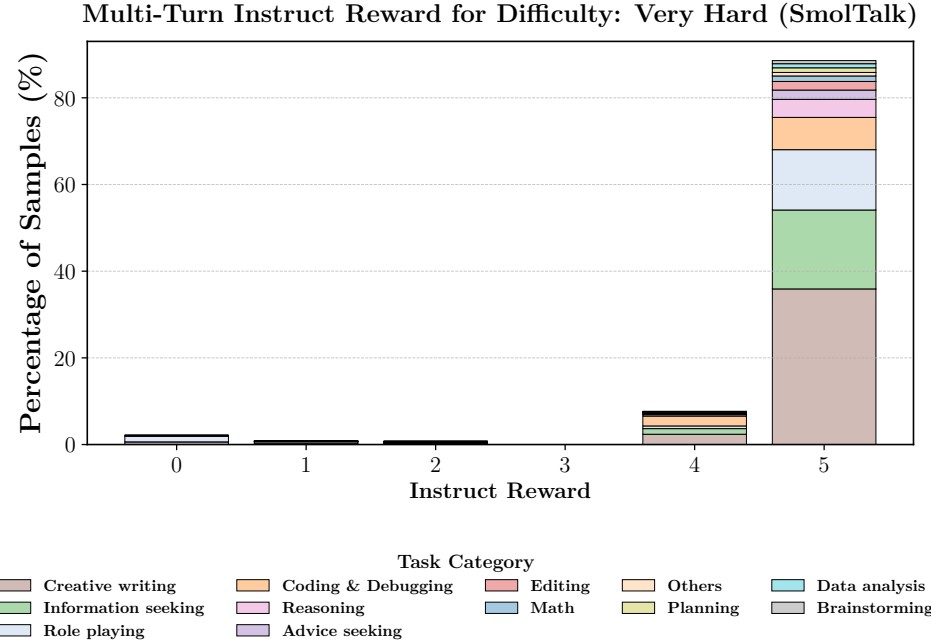

(b) Multi-turn instruct reward distribution for very hard samples in SmolTalk.

Figure 35: Instruct reward distribution by difficulty level for multi-turn samples in SmolTalk. Reward saturation at score 5 persists across difficulty levels, with only minor shifts in task composition.

## C.7 Language Analysis

In this section, we provide an overview of the language distribution in the Tulu and SmolTalk datasets.

### C.7.1 Overall Language Distributions

Both Tulu and SmolTalk are predominantly English datasets. Fig. 36a shows the top five languages represented in Tulu, with English (EN) accounting for 95.4% of all samples, followed by Russian (RU) at 1.5% and Simplified Chinese (ZH) at 1.1%. Fig. 36b presents the same analysis for SmolTalk, where English dominates even more strongly, comprising 99.3% of all samples. These results indicate that both datasets are almost exclusively focused on English conversations, with only marginal inclusion of multilingual content.

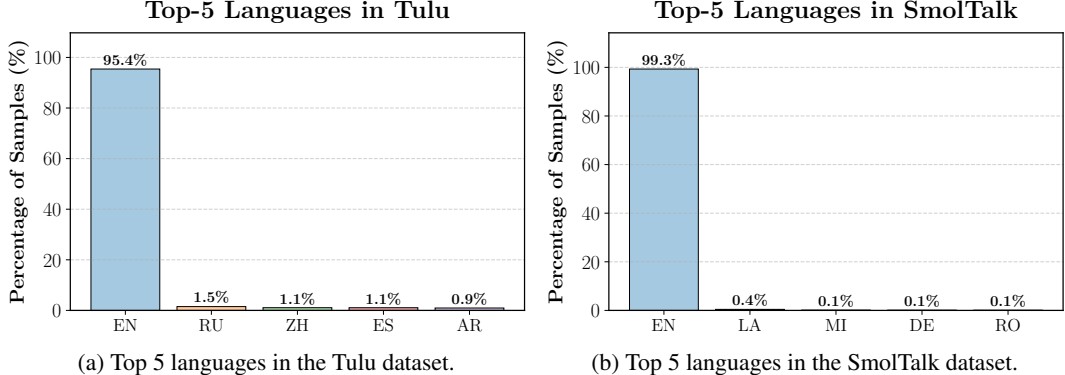

(a) Top 5 languages in the Tulu dataset.  (b) Top 5 languages in the SmolTalk dataset.

Figure 36: Overall language distribution in the Tulu and SmolTalk datasets. Both are overwhelmingly English-centric, with only a small fraction of samples in other languages such as Russian and Chinese.

### C.7.2 Language by Task Category

Fig. 37 Fig. 38 show the top three languages by task category for the Tulu and SmolTalk datasets.

For Tulu, *Math* accounts for the largest share of English samples (42%), followed by *Information Seeking* and *Coding*, each contributing around 17%. Russian and Chinese samples are predominantly associated with *Information Seeking* tasks, an intuitive result, as users often query factual information in their native language during chat interactions.

In SmolTalk, the distribution of English samples is more balanced, with *Math* and *Information Seeking* comprising 22% and 20%, respectively. Interestingly, in the Latin American language group (LA), the vast majority of samples (75%) correspond to *Math* tasks. Upon inspection, we find that many of these are simple mathematical expressions, e.g., "24 x 17 + 673 - 36.7 = ?", containing no natural language text. Magpie does not misclassify these samples per se, but rather assigns them to the LA language group, likely due to a lack of sufficient linguistic signal to support a more accurate classification. While this behavior is notable, the overall size of the LA subset is only 0.4% of the dataset, making this an inconsequential artifact in practice.

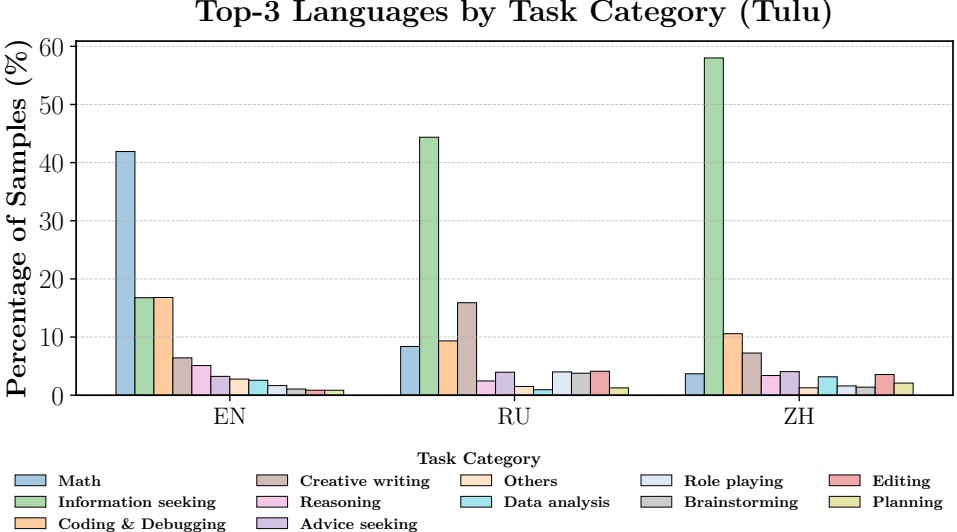

Figure 37: Top 3 languages by task category for the Tulu dataset. English samples are dominated by *Math*, *Information Seeking*, and *Coding*, while Russian and Chinese samples primarily cluster around *Information seeking* queries.

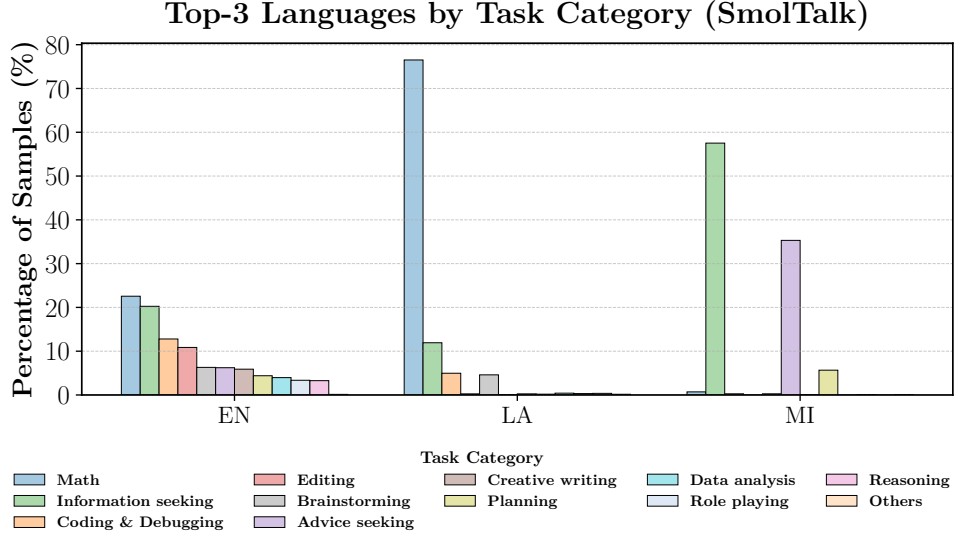

Figure 38: Top 3 languages by task category for the SmolTalk dataset. English samples are largely split between *Math* and *Information Seeking*. Latin American samples are mostly simple mathematical expressions, leading to their classification under *Math* despite minimal linguistic content.

## C.8 Safety

Safety is a crucial aspect of post-training datasets. In this section, we examine the safety characteristics of the Tulu and SmolTalk corpora.

### C.8.1 Overall Safety Distribution

Fig. 39 shows the overall safety distribution for the Tulu and SmolTalk datasets. In general, both datasets are predominantly safe, with 97.4% of Tulu samples and 99.1% of SmolTalk samples flagged as safe. This reflects the rigorous quality control applied by Lambert et al. [7] and Allal et al. [9], which includes an explicit focus on safety during data collection and filtering. Nevertheless, a small fraction of samples are labeled as unsafe. Below, we provide two representative examples from Tulu and SmolTalk.

Figure 39: Safety distribution for Tulu and SmolTalk. Both datasets are predominantly safe.

**Example 1 from Tulu: Medical advice.**

```
'conversations': [
    {'from': 'human', 'value': "Pretend you are a doctor. I'm feeling
                                a subtle pain in my chest and my sternum
                                feels like I need to stretch it until
                                it pops. Am I having a heart attack? Is
                                there anything I need to be worried
                                about?"},

    {'from': 'gpt', 'value': "No, this is unlikely to be a heart attack,
                              but you should still take care of it.
                              I'd suggest you get up and move around a bit.
                              Mild pain and tightness of the sternum can be
                              caused by spasms in the muscles surrounding
                              the chest, back, and shoulders. Such spasms
                              may cause ...},
]
```

In this example, the assistant provides *specialized advice*, which is flagged by Llama-Guard 2 [29] as inappropriate, particularly due to its medical context. Even if the content appears benign or well-intentioned, Llama-Guard still classifies it under one of its 11 unsafe categories.

**Example 2 from SmolTalk: Casino game.**

```
'conversations': [
    {'from': 'human', 'value': "How can we design and develop a cutting-
                                edge online casino platform with a unique
                                and innovative crash game that surpasses
                                competitors, boasting visually stunning
                                graphics and captivating features? The
                                crash game should be built using PHP
                                and JS ...},

    {'from': 'gpt', 'value': "To design and develop a cutting-edge online
                              casino platform with a unique and innovative
                              crash game, we need to follow these steps:
                              1. Conduct thorough ...},
]
```

In this case, the user is asking for technical guidance on building a casino-style game. Although the query and response are not overtly harmful, Llama-Guard flags the conversation as unsafe due to its association with gambling, assigning it to the *non-violent crime* category.

These examples illustrate that most unsafe samples are benign in appearance but contain elements, such as medical or gambling-related content that trigger conservative safety filters. Nonetheless, the proportion of such flagged instances is negligible in both datasets. However, safety in LLM fine-tuning remains an active area of research [73], with various mechanisms available to instill safety both during and after training [74–76]. We leave a more comprehensive safety analysis of post-training datasets to future work.

### C.8.2 Safety by Task Category

Fig. 40 and Fig. 41 show the distribution of safe and unsafe samples across task categories in the Tulu and SmolTalk datasets.

Overall, both datasets exhibit no meaningful concentration of unsafe samples in any specific task category. In SmolTalk, for example, the highest proportion of unsafe samples appears in *Information Seeking*, but even here the rate remains extremely low at just 2.6%. This analysis supports our earlier observations: unsafe labels are rare, broadly and uniformly distributed, and not linked to any anomalous or harmful behavior within specific task types. Instead, most flagged cases reflect conservative or overly sensitive filtering (see previous examples). As such, the safety risks at the task-category level are negligible and likely represent noise or edge-case overflagging by Magpie's Llama Guard safety classifier.

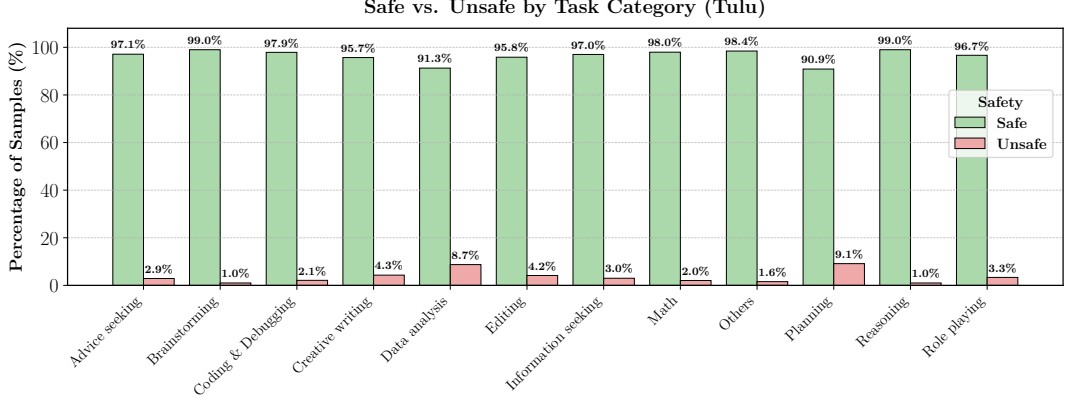

Figure 40: Distribution of safe and unsafe samples by task category in the Tulu dataset. Unsafe samples are rare and show no meaningful concentration in any specific category.

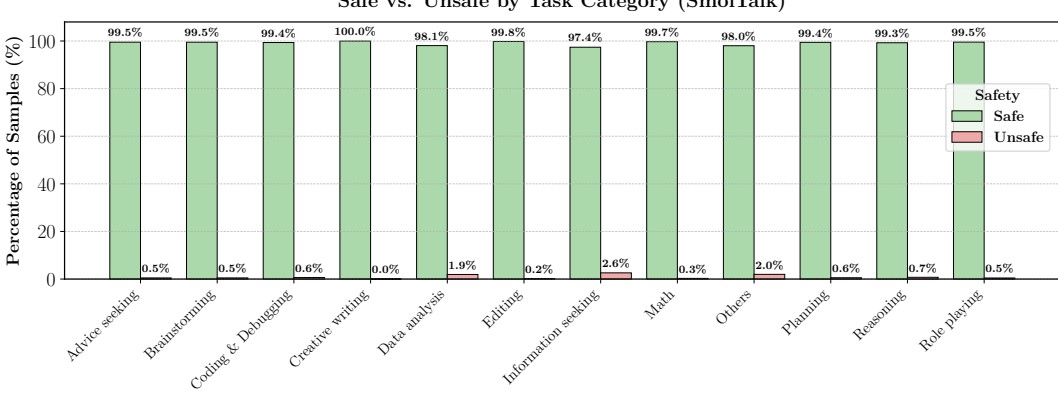

Figure 41: Distribution of safe and unsafe samples by task category in the SmolTalk dataset. Unsafe samples are rare and show no meaningful concentration in any specific category.

# D  Data Curation Recipe Details

This section provides a detailed overview of our quality-based and task-aware data curation recipe used to construct our *TuluTalk* data mixture. The steps of the algorithm are outlined in Fig. 42.

In **Step 1**, we compute quantiles over reward scores to guide subsequent selection thresholds. Specifically, we compute:

- First and second quantiles of single-turn samples with input quality labeled as `excellent`
- Third quantile of single-turn samples with input quality `good`

These thresholds serve as principled cutoffs for isolating top-tier completions, allowing us to distinguish high-reward responses from more average ones within higher-quality prompt strata.

**Step 2** constitutes our initial quality-based curation stage. Here, we select only samples with the highest input quality and highest response reward. This includes all multi-turn samples labeled `excellent` with a reward score of 5, and all single-turn samples labeled `excellent` with a reward score above the second quantile.

**Step 3** identifies task categories whose representation in the selected set $\mathcal{D}_c$ drops by more than a threshold $\tau$ relative to the original dataset. This step ensures that quality filtering does not disproportionately exclude certain task types.

In **Step 4**, we reintroduce high-quality fallback samples from underrepresented task categories to restore diversity. Specifically, we add:

- Multi-turn samples with input quality `excellent` and response reward of 4
- Single-turn samples with input quality `excellent` and response reward between the first and second quantiles

These samples are labeled as *"high-quality fallback"* in Fig. 42.

We further boost task diversity by introducing *"diversity boost"* samples. Specifically, these are:

- Multi-turn samples with input quality `good` and response reward of 5
- Single-turn samples with input quality `good` and reward above the third quantile

As discussed in the main paper, we found that applying only the quality-based filtering initially led to suboptimal performance due to a shortage of instruction following samples, an issue successfully addressed by the diversity-enhancing additions in Step 4.

Together, these refinements ensure that the curated dataset maintains both high overall quality and balanced coverage across task categories.

We apply this curation pipeline independently to the annotated Tulu and SmolTalk datasets and merge the resulting subsets to form our *TuluTalk* mixture.

We select the quantiles as an intuitive and natural choice for the thresholds on single-turn reward scores, and do not perform additional ablations due to limited compute budget. We leave a thorough ablation study to find optimum thresholds as a future work.

**Quality- and Task-Aware Data Curation Recipe**

**Input:** Annotated dataset $\mathcal{D}$ with Magpie tags for input quality (`input_quality`), single-turn/multi-turn response quality (`st_reward`/`mt_reward`), and task category (`task_category`); task diversity threshold $\tau$.

**Output:** Curated subset $\mathcal{D}_c$ that is both high-quality and task-diverse.

**Recipe:**

1. Compute quantiles:

$$Q_1^e,\ Q_2^e \leftarrow \text{1st/2nd quantiles of } \big\{ S[\texttt{st\_reward}] \mid S[\texttt{input\_quality}] = \texttt{excellent},$$
$$S[\texttt{turn}] = \texttt{single\_turn} \big\},$$
$$Q_3^g \leftarrow \text{3rd quantile of } \big\{ S[\texttt{st\_reward}] \mid S[\texttt{input\_quality}] = \texttt{good},$$
$$S[\texttt{turn}] = \texttt{single\_turn} \big\}.$$

2. For each $S \in \mathcal{D}$, add $S$ to $\mathcal{D}_c$ if

$$S[\texttt{input\_quality}] = \texttt{excellent} \wedge$$
$$\Big( (S[\texttt{turn}] = \texttt{multi\_turn} \wedge S[\texttt{mt\_reward}] = 5)$$
$$\vee\ (S[\texttt{turn}] = \texttt{single\_turn} \wedge S[\texttt{st\_reward}] > Q_2^e) \Big).$$

3. Let $\mathcal{C}$ be the set of task categories whose coverage in $\mathcal{D}_c$ drops by more than $\tau\%$ relative to $\mathcal{D}$.

4. For each $S \in \mathcal{D} \setminus \mathcal{D}_c$, add $S$ to $\mathcal{D}_c$ if

$$S[\texttt{task\_category}] \in \mathcal{C} \wedge$$
$$\Big( S[\texttt{input\_quality}] = \texttt{excellent} \wedge \big( (S[\texttt{turn}] = \texttt{multi\_turn} \wedge S[\texttt{mt\_reward}] = 4)$$
$$\underbrace{\vee\ (S[\texttt{turn}] = \texttt{single\_turn} \wedge Q_1^e < S[\texttt{st\_reward}] < Q_2^e))}_{\text{high-quality fallback}}$$
$$\vee$$
$$\Big( S[\texttt{input\_quality}] = \texttt{good} \wedge \big( (S[\texttt{turn}] = \texttt{multi\_turn} \wedge S[\texttt{mt\_reward}] = 5)$$
$$\underbrace{\vee\ (S[\texttt{turn}] = \texttt{single\_turn} \wedge S[\texttt{st\_reward}] > Q_3^g)) \Big) \Big)}_{\text{diversity boost}}$$

Figure 42: Quality- and task-aware curation recipe used to construct the *TuluTalk* data mixture. Steps 1-4 sequentially select high-quality examples (Step 2), identify underrepresented task categories (Step 3), and reintroduce fallback samples (Step 4) to restore task diversity without compromising input or output quality.

# E   Details on Experimental Setup and Additional Results

This section presents supplementary results, including those from SFT and DPO, and provides details on the fine-tuning and evaluation configurations used throughout our experiments.

## E.1   Fine-Tuning Configurations

To ensure reproducibility and comparability, we fine-tune all models using AllenAI's *Open-Instruct* framework[9], covering both SFT and DPO. By default, Open-Instruct applies a sum-reduction over token-level losses, rather than the more commonly used mean-reduction. This design choice ensures length-equitable weighting, where short and long sequences contribute proportionally to the total loss, preventing shorter examples from disproportionately influencing the gradient due to having fewer tokens. Moreover, summing losses leads to more stable optimization by avoiding fluctuations in loss scale caused by variation in batch composition or sequence length distributions. We refer to a more detailed analysis and discussion in [7].

### E.1.1   Supervised Fine-Tuning (SFT)

We fine-tune Llama-3.1-8B [27] and SmolLM2-1.7B [9] models on the Tulu [7], SmolTalk [9], Orca [16], and our proposed *TuluTalk* dataset. These two models are selected for consistency with prior work, being the default backbones in the respective dataset papers for Tulu and SmolTalk.

Fine-tuning is performed using BF16 mixed precision with Fully Sharded Data Parallelism (FSDP) on 8 × NVIDIA A100 80GB GPUs. To isolate dataset effects, we fix SFT hyperparameters per model across all experiments. For Llama-3.1-8B, we adopt the same hyperparameters as in Lambert et al. [7], and for SmolLM2-1.7B, we follow Allal et al. [13]. Table 14 provides the SFT training configurations for both models.

### E.1.2   Direct Preference Optimization (DPO)

For DPO, we choose the preference mixture[10] created by Lambert et al. [7], which is a curated blend of *on-policy* and *off-policy* preference data, synthetic instruction following augmentations, WildChat [8] conversational pairs, cleaned UltraFeedback [47] data, and a small Persona IF [63] subset, designed to balance broad performance and targeted instruction following under the DPO objective. We use the same hyperparameters from [7]. We set the KL-penalty coefficient (referred to as `dpo_beta` in *Open-Instruct*) to 5 and use the length-normalized DPO loss (`dpo_loss_type=norm`), following the recommendation in Lambert et al. [7]. We apply DPO on models that have already been fine-tuned via SFT using the Tulu, SmolTalk, and TuluTalk datasets. The full set of DPO hyperparameters is provided in Table 14.

## E.2   Evaluation Setup

We assess model performance using the *LM Evaluation Harness* framework [28], a widely adopted standard for evaluating language models across diverse benchmark suites. To ensure a comprehensive and task-diverse evaluation, we include benchmarks spanning *Knowledge* (e.g., MMLU [77], TruthfulQA [78]), *Reasoning* (e.g., BBH [79], ARC-C [80]), *Commonsense Understanding* (e.g., HellaSwag [81], WinoGrande [82]), *Instruction Following* (e.g., IF-Eval [83]), *Mathematical Reasoning* (e.g., GSM8K [84], MATH [85]), and *Coding* (e.g., HumanEval, HumanEval+ [86]). We further include benchmarks from *Open LLM Leaderboards* [17, 18] to gauge general instruction performance under competitive public standards. This setup ensures a fair, fine-grained comparison between models and data mixtures, highlighting both strengths and failure modes across capabilities.

---

[9]`https://github.com/allenai/open-instruct`
[10]`https://huggingface.co/datasets/allenai/llama-3.1-tulu-3-8b-preference-mixture`

Table 14: Training hyperparameters for SFT and DPO on Llama-3.1-8B and SmolLM2-1.7B.

| Parameter | SFT | | DPO | |
| --- | --- | --- | --- | --- |
| | Llama-3.1-8B | SmolLM2-1.7B | Llama-3.1-8B | SmolLM2-1.7B |
| Total Batch Size | 128 | 128 | 128 | 128 |
| Per-Device Batch Size | 1 | 1 | 1 | 1 |
| Gradient Accumulation Steps | 16 | 16 | 16 | 16 |
| Max Sequence Length | 4096 | 8192 | 2048 | 2048 |
| Number of Epochs | 2 | 2 | 1 | 1 |
| Learning Rate | $5 \times 10^{-6}$ | $3 \times 10^{-4}$ | $5 \times 10^{-7}$ | $5 \times 10^{-7}$ |
| LR Scheduler | Linear | Cosine | Linear | Linear |
| Warmup Ratio | 0.03 | 0.10 | 0.10 | 0.10 |
| Weight Decay | 0.0 | 0.0 | 0.0 | 0.0 |

## E.3   Additional Results

### E.3.1   SFT with Tulu+SmolTalk Mixture

Table 15 presents additional experiments using a naïve data mixture formed by directly concatenating the full Tulu and SmolTalk datasets (denoted as Tulu+SmolTalk). This results in an (uncurated) corpus of approximately 1.979 million samples.

For the Llama model, the naïve mixture performs slightly better than Tulu but worse than SmolTalk. It achieves the best scores on IF-Eval (74.94%) and GSM8K (77.03%) but underperforms on reasoning tasks (ARC, BBH, MuSR), commonsense tasks (HellaSwag, WinoGrande), and MMLU. In coding benchmarks, it offers no meaningful gains over Tulu and remains significantly behind SmolTalk.

For the SmolLM model, Tulu+SmolTalk slightly outperforms both Tulu and SmolTalk, with the most notable improvement on GSM8K (56.07%). However, the overall performance gain is marginal, only 0.24% higher than SmolTalk and thus indicating that this naïve mixture, despite doubling the dataset size, lacks the benefit of thoughtful curation.

These results underscore that simply merging two strong datasets does not guarantee performance improvements. In contrast, our systematic and principled curation based on quality and diversity yields the size-efficient *TuluTalk* mixture which outperforms both Tulu and SmolTalk, as well as the naïve Tulu+SmolTalk combination, in overall average and across many benchmarks for both models.

### E.3.2   SmolLM Performance on Code Benchmarks

In Tables 1 and 3 of the main paper, as well as the supplemental results in Table 15, the SmolLM model shows identical, low scores across all dataset variants on the HumanEval and HumanEval+ coding tasks. This suggests that the model fails to generalize meaningfully to code-related benchmarks and likely resorts to template-based or fallback completions. Examples of such behavior include emitting empty function stubs or default print statements. These outputs rarely match the required semantics of the prompt, leading to consistently low pass@1 scores, which measure exact functional correctness on the first attempt. These results highlight the capacity limitations of the smaller SmolLM architecture, which was explicitly designed to prioritize conversational fluency over structured reasoning. This limitation becomes especially apparent given that the same SmolTalk dataset improves coding performance when used to train the larger Llama model. Furthermore, the same pattern holds for the Tulu and Orca datasets: while SmolLM continues to underperform on code benchmarks, the same datasets yield clear improvements when used to train the larger Llama model.

Table 15: SFT results for Llama-3.1-8B and SmolLM2-1.7B base models fine-tuned on Tulu (939k samples), SmolTalk (1.04m samples), a naïve concatenation of the two (Tulu+SmolTalk; 1.979m samples), and our curated *TuluTalk (808k samples)*, evaluated on the Open LLM Leaderboards (averaged) and code benchmarks. The overall average is across all benchmarks. **Bold** marks the row-wise best score. Color-shaded columns highlight the superior TuluTalk model.

| | Llama-3.1-8B | | | | SmolLM2-1.7B | | | |
|---|---|---|---|---|---|---|---|---|
| **Benchmark** | Tulu | SmolTalk | Tulu + SmolTalk | TuluTalk | Tulu | SmolTalk | Tulu + SmolTalk | TuluTalk |
| *Knowledge* | | | | | | | | |
| MMLU (5-shot) | 62.90 | 62.88 | 62.68 | **63.91** | **49.71** | 47.88 | 49.61 | 49.34 |
| MMLU-Pro (5-shot) | 28.73 | **31.76** | 28.49 | 30.17 | 19.61 | 20.37 | 20.25 | **20.67** |
| TruthfulQA (0-shot) | 46.41 | **55.74** | 51.57 | 53.16 | 44.04 | **44.74** | 42.12 | 43.65 |
| GPQA (0-shot) | 27.77 | 28.78 | 28.02 | **29.28** | **27.85** | 27.60 | 26.59 | 27.68 |
| *Reasoning* | | | | | | | | |
| ARC-C (25-shot) | 54.61 | **59.04** | 54.44 | 57.42 | 44.54 | **48.46** | 46.59 | 47.27 |
| BBH (3-shot) | 39.06 | **45.50** | 44.99 | 43.50 | 36.66 | 37.81 | 38.29 | **38.33** |
| MuSR (0-shot) | **42.86** | 38.49 | 39.76 | 40.62 | 33.33 | 33.86 | **34.39** | 33.28 |
| *Commonsense* | | | | | | | | |
| HellaSwag (10-shot) | 60.87 | 61.54 | 61.14 | **62.98** | 51.01 | **52.10** | 50.80 | 51.36 |
| WinoGrande (5-shot) | 76.64 | 77.19 | 76.40 | **79.22** | 65.90 | 65.27 | 65.59 | **66.06** |
| *Instruction Following* | | | | | | | | |
| IF-Eval (0-shot) | 74.09 | 74.51 | **74.94** | 74.84 | 60.25 | 56.83 | 60.73 | **60.85** |
| *Math* | | | | | | | | |
| GSM8K (5-shot) | 74.37 | 74.75 | **77.03** | 74.84 | 49.43 | 52.46 | **56.07** | 54.13 |
| MATH (4-shot) | **12.31** | 10.42 | 10.20 | 11.96 | **6.27** | 5.89 | 5.51 | 6.16 |
| *Code* | | | | | | | | |
| HumanEval (pass@1) | **58.54** | 54.51 | 55.88 | 56.49 | **1.83** | **1.83** | **1.83** | **1.83** |
| HumanEval+ (pass@1) | **45.37** | 44.27 | 44.88 | 44.33 | **1.83** | **1.83** | **1.83** | **1.83** |
| *Leaderboards* | | | | | | | | |
| Open LLM Leaderboard 1 | 62.63 | 65.19 | 63.88 | **65.26** | 50.77 | 51.82 | 51.80 | **51.97** |
| Open LLM Leaderboard 2 | 37.47 | 38.24 | 37.73 | **38.40** | 30.66 | 30.39 | 30.96 | **31.16** |
| *Overall* | 50.32 | 51.38 | 50.74 | **51.62** | 35.16 | 35.49 | 35.73 | **35.89** |

### E.3.3 Efficiency Gains

To assess efficiency and training cost, we report the number of processed tokens (computed with each model's distinct tokenizer), estimates for training FLOPs, and total GPU hours (on an 8 x A100 GPU cluster) in Table 16 for SFT training of Llama-3.1-8B and SmolLM-2-1.7B models on Tulu, SmolTalk, and TuluTalk. We find that the reduction in dataset size translates approximately linearly into efficiency gains. For example, TuluTalk is around 14% smaller than Tulu. For Llama, this results in a proportionate reduction in the number of processed tokens (708M compared to 835M), ExaFLOPs (34 compared to 40), and total GPU hours (38 compared to 45). Similar trends are observed for the SmolTalk dataset and for the SmolLM model. These additional experiments validate the efficiency improvements achieved by our curated TuluTalk dataset.

Table 16: Comparison of SFT training efficiency for Llama-3.1-8B and SmolLM2-1.7B on Tulu, SmolTalk, and TuluTalk. We report processed tokens (per tokenizer), estimated ExaFLOPs, and GPU hours (excluding the initial warmup phase). Lower is better ($\downarrow$).

| | Llama-3.1-8B | | | SmolLM2-1.7B | | |
|---|---|---|---|---|---|---|
| **Metric** | Tulu | SmolTalk | TuluTalk | Tulu | SmolTalk | TuluTalk |
| Tokens ($\downarrow$) | 835M | 875M | **708M** | 910M | 961M | **782M** |
| ExaFLOPs ($\downarrow$) | 40.1 | 42.0 | **34.0** | 9.28 | 9.80 | **7.98** |
| GPU Hours ($\downarrow$) | 45 | 49 | **38** | 26 | 28 | **22** |

### E.3.4 Performance Results for Diverse Models and Scales

To demonstrate the effectiveness and generalizability of TuluTalk across different architectures and scales, we provide results for three additional models: Qwen2.5-0.5B and Qwen2.5-3B [22], as well as for SmolLM3-3B [31], covering small- to mid-scale models of different architectures.

Tables 17 and 18 report evaluation results across all considered benchmarks. In general, the results are in line with the observations in our main body and show that our curated TuluTalk SFT dataset achieves better performance compared to Tulu and Smoltalk, while being a leaner dataset overall. These additional results demonstrate the generalizability of both TuluTalk and our curation recipe across model architectures and scales. While evaluating larger models like Qwen2.5-32B would be informative, our computational setup and budget unfortunately limits us from training larger models.

Table 17: Results for Qwen2.5-0.5B and Qwen2.5-3B fine-tuned on Tulu, SmolTalk, and TuluTalk, evaluated on the Open LLM Leaderboards and code benchmarks. The overall average is across all benchmarks. **Bold** marks the row-wise best score. Color-shaded columns highlight TuluTalk models.

| Benchmark | Qwen2.5-0.5B | | | | Qwen2.5-3B | | | |
| --- | --- | --- | --- | --- | --- | --- | --- | --- |
| | Base | Tulu | SmolTalk | TuluTalk | Base | Tulu | SmolTalk | TuluTalk |
| *Knowledge* | | | | | | | | |
| MMLU (5-shot) | **46.51** | 45.67 | 43.25 | 44.65 | 65.55 | **66.09** | 65.80 | 65.03 |
| MMLU-Pro (5-shot) | **16.98** | 14.14 | 13.15 | 13.89 | 32.12 | **32.56** | 32.00 | 31.62 |
| TruthfulQA (0-shot) | 39.78 | 39.99 | **40.87** | 39.46 | 48.87 | 46.85 | **50.68** | 48.76 |
| GPQA (0-shot) | **27.20** | 24.33 | 25.59 | 25.59 | **28.27** | 26.43 | 27.35 | 28.19 |
| *Reasoning* | | | | | | | | |
| ARC-C (25-shot) | 32.22 | 31.83 | 31.66 | **32.59** | **52.90** | 52.29 | 51.34 | 52.32 |
| BBH (3-shot) | **31.58** | 30.79 | 29.53 | 29.70 | **46.38** | 45.20 | 43.03 | 45.88 |
| MuSR (0-shot) | **34.26** | 32.01 | 32.41 | 32.54 | **43.25** | 41.14 | 39.77 | 42.83 |
| *Commonsense* | | | | | | | | |
| HellaSwag (10-shot) | **39.93** | 39.10 | 38.70 | 39.18 | 55.51 | **56.65** | 54.97 | 56.40 |
| WinoGrande (5-shot) | 56.75 | **58.33** | 57.14 | 57.83 | 71.35 | 72.06 | 70.24 | **72.09** |
| *Instruction Following* | | | | | | | | |
| IF-Eval (0-shot) | 17.34 | 40.60 | 35.59 | **43.24** | 27.13 | 63.44 | 61.50 | **66.75** |
| *Math* | | | | | | | | |
| GSM8K (5-shot) | 34.50 | 37.30 | **41.93** | 41.47 | 70.20 | 74.21 | **77.33** | 77.10 |
| MATH (4-shot) | 4.68 | **6.50** | 5.21 | 5.59 | 15.63 | **21.00** | 16.99 | 18.67 |
| *Code* | | | | | | | | |
| HumanEval (pass@1) | 28.66 | **28.66** | 27.44 | 27.44 | 38.41 | **44.34** | 41.22 | 42.68 |
| HumanEval+ (pass@1) | **26.22** | 25.00 | 24.39 | 24.39 | 32.93 | **39.12** | 34.27 | 36.85 |
| *Leaderboards* | | | | | | | | |
| Open LLM Leaderboard 1 | 41.62 | 42.04 | 42.26 | **42.53** | 60.73 | 61.36 | 61.73 | **61.95** |
| Open LLM Leaderboard 2 | 22.01 | 24.73 | 23.58 | **25.09** | 32.13 | 38.29 | 36.77 | **38.99** |
| *Overall* | 31.19 | 32.45 | 31.92 | **32.68** | 44.89 | 48.67 | 47.61 | **48.94** |

Table 18: Results for SmolLM3-3B fine-tuned on Tulu, SmolTalk, and TuluTalk, evaluated on the Open LLM Leaderboards and code benchmarks. **Bold** marks the row-wise best score. The color-shaded column highlights TuluTalk.

| | SmolLM3-3B | | | |
|---|---|---|---|---|
| **Benchmark** | Base | Tulu | SmolTalk | TuluTalk |
| *Knowledge* | | | | |
| MMLU (5-shot) | 61.37 | 61.10 | 59.96 | **61.55** |
| MMLU-Pro (5-shot) | **33.55** | 28.86 | 31.14 | 31.80 |
| TruthfulQA (0-shot) | 45.91 | 45.98 | **49.87** | 49.19 |
| GPQA (0-shot) | 29.70 | 28.61 | **30.54** | 29.85 |
| *Reasoning* | | | | |
| ARC-C (25-shot) | 56.06 | 53.75 | **58.53** | 58.38 |
| BBH (3-shot) | **45.57** | 44.52 | 44.68 | 44.19 |
| MuSR (0-shot) | **41.80** | 40.61 | 39.38 | 40.30 |
| *Commonsense* | | | | |
| HellaSwag (10-shot) | **57.24** | 55.49 | 56.22 | 56.74 |
| WinoGrande (5-shot) | 72.77 | 73.16 | 72.38 | **74.03** |
| *Instruction Following* | | | | |
| IF-Eval (0-shot) | 19.52 | **65.78** | 56.89 | 63.20 |
| *Math* | | | | |
| GSM8K (5-shot) | 67.10 | 70.32 | **73.36** | 72.55 |
| MATH (4-shot) | 16.92 | 19.49 | **20.62** | 19.73 |
| *Code* | | | | |
| HumanEval (pass@1) | 37.20 | **47.44** | 45.24 | 44.76 |
| HumanEval+ (pass@1) | 29.88 | **32.44** | 31.10 | 30.46 |
| *Leaderboards* | | | | |
| Open LLM Leaderboard 1 | 60.08 | 59.97 | 61.72 | **62.07** |
| Open LLM Leaderboard 2 | 31.18 | 37.98 | 37.21 | **38.18** |
| *Overall* | 43.90 | 47.68 | 47.85 | **48.34** |

### E.3.5   Performance Results for SFT and DPO for Llama-3.1-8B

To assess whether our SFT curation insights transfer to preference-tuned models, we also apply DPO on Llama models fine-tuned on Tulu, SmolTalk, and our proposed TuluTalk mixture. Table 19 reports DPO and SFT performance results across benchmarks for the Llama-3.1-8B base model, fine-tuned on all datasets under consideration and DPO-tuned using the same preference mixture proposed by Lambert et al. [7].

For both Tulu and SmolTalk, DPO leads to notable improvements on TruthfulQA, all reasoning benchmarks, HellaSwag, and especially instruction following. For Tulu, math performance shows a mixed trend: GSM8K scores decrease slightly, while MATH improves significantly. This effect is not observed for SmolTalk, where math performance remains largely unchanged. Coding performance declines slightly for both datasets, while both Open LLM Leaderboard scores increase noticeably.

Importantly, the performance gains observed for our proposed *TuluTalk* under SFT carry over to the DPO setting. Specifically, TuluTalk achieves the highest overall average under DPO (53.08%), outperforming Tulu (51.89%) and SmolTalk (52.96%), and improving upon the base model by over 11%. These gains are observed across evaluation categories, including instruction following, reasoning, and commonsense understanding, highlighting the consistency of performance gains across model sizes and families.

Notably, the DPO-TuluTalk model achieves the best IF-Eval score (81.51%) and leads on HellaSwag and Open LLM Leaderboard 1. This suggests that our mixture not only improves factual accuracy and instruction compliance, but also enhances performance on alignment-sensitive public benchmarks.

Overall, these results confirm that our principled, quality- and diversity-driven curation, with attention to instruction following signals, response quality, and task diversity, not only yields performance gains under SFT, but also that these gains transfer to DPO.

Table 19: Performance of Llama-3.1-8B (base) fine-tuned via SFT or DPO on Tulu, SmolTalk, and TuluTalk, evaluated on Open LLM leaderboards, code benchmarks, and reasoning tasks. The overall average is across all benchmarks. **Bold** marks the row-wise best score. Color-shaded columns highlight the superior TuluTalk model under each training method.

| Benchmark | Base | SFT | | | DPO | | |
| --- | --- | --- | --- | --- | --- | --- | --- |
| | | Tulu | SmolTalk | TuluTalk | Tulu | SmolTalk | TuluTalk |
| *Knowledge* | | | | | | | |
| MMLU (5-shot) | **65.03** | 62.90 | 62.88 | 63.91 | 62.09 | 63.30 | 62.58 |
| MMLU-Pro (5-shot) | **32.71** | 28.73 | 31.76 | 30.17 | 29.75 | 32.58 | 31.32 |
| TruthfulQA (0-shot) | 45.22 | 46.41 | 55.74 | 53.16 | 56.28 | **62.45** | 61.67 |
| GPQA (0-shot) | **31.46** | 27.77 | 28.78 | 29.28 | 29.28 | 30.70 | 28.69 |
| *Reasoning* | | | | | | | |
| ARC-C (25-shot) | 54.69 | 54.61 | 59.04 | 57.42 | 57.85 | **61.12** | 60.75 |
| BBH (3-shot) | 46.48 | 39.06 | 45.50 | 43.50 | 40.74 | **46.50** | 44.91 |
| MuSR (0-shot) | 37.96 | **42.86** | 38.49 | 40.62 | 40.61 | 39.15 | 39.02 |
| *Commonsense* | | | | | | | |
| HellaSwag (10-shot) | 61.44 | 60.87 | 61.54 | 62.98 | 65.48 | 66.38 | **68.64** |
| WinoGrande (5-shot) | 76.87 | 76.64 | 77.19 | **79.22** | 74.74 | 76.95 | 79.01 |
| *Instruction Following* | | | | | | | |
| IF-Eval (0-shot) | 12.45 | 74.09 | 74.51 | 74.84 | 80.51 | 79.16 | **81.51** |
| *Math* | | | | | | | |
| GSM8K (5-shot) | 50.64 | 74.37 | 74.75 | **74.84** | 69.45 | 75.74 | 74.75 |
| MATH (4-shot) | 5.97 | 12.31 | 10.42 | 11.96 | **20.85** | 12.61 | 13.52 |
| *Code* | | | | | | | |
| HumanEval (pass@1) | 34.76 | **58.54** | 54.51 | 56.49 | 56.10 | 54.15 | 55.66 |
| HumanEval+ (pass@1) | 28.66 | **45.37** | 44.27 | 44.33 | 42.76 | 40.61 | 41.10 |
| *Leaderboards* | | | | | | | |
| Open LLM Leaderboard 1 | 58.98 | 62.63 | 65.19 | 65.26 | 64.32 | 67.66 | **67.90** |
| Open LLM Leaderboard 2 | 27.84 | 37.47 | 38.24 | 38.40 | **40.29** | 40.12 | 39.83 |
| *Overall* | 41.74 | 50.32 | 51.38 | 51.62 | 51.89 | 52.96 | **53.08** |

# F  Limitations and Broader Impact

**Limitations.**  While our study provides a comprehensive and principled analysis of post-training SFT datasets, a few limitations remain. First, our annotations rely on the Magpie framework, which uses the LLM-as-a-judge technique to score various aspects such as prompt quality, response helpfulness, and safety. Although we enhance Magpie with error-tolerant parsing and extend it for multi-turn support, the subjectivity inherent in LLM-based judgments may introduce variance in label quality. In addition, annotations reflect the capabilities and biases of the underlying judge model, which may shift as stronger evaluators emerge. Nonetheless, the consistency of observed trends and performance gains across benchmarks suggests that our annotations, generated using a capable Llama-3.3-70B-Instruct judge, are robust and highly informative for practical curation. Second, while our analysis focuses on the SFT stage, evaluating and comparing data quality for preference tuning remains an important direction for future work, particularly as the variety of training recipes used in preference tuning makes dataset comparisons more challenging. Third, when designing *TuluTalk*, we perform a limited number of ablations for balancing task diversity due to computational constraints. It is interesting to perform additional data mixture ablations to enhance the performance of TuluTalk. Finally, as TuluTalk is derived from the open-source Tulu and SmolTalk datasets, it inherits any existing biases and limitations present in those corpora, such as a predominant focus on English and limited coverage of specialized skills like tool use.

**Broader Impact.**  By open-sourcing detailed annotations, curated data mixtures, and reproducible recipes, our work lowers the barrier to high-quality post-training research and promotes transparency in dataset design. Our quality annotations of Tulu and SmolTalk can be leveraged by both researchers and practitioners to conduct further analyses or construct data mixtures tailored to their specific use cases. *TuluTalk*, our curated dataset, achieves top-tier performance with substantially fewer samples, offering improvements in both compute efficiency during SFT and downstream performance. While we apply our curation recipe on Tulu and SmolTalk, our quality-based and diversity-driven curation recipe can be used with any datasets in principle. Even though the datasets we build on are derived from open and broadly safe sources, we acknowledge that any general-purpose LLM corpus carries dual-use risk. We encourage responsible use and support future work that incorporates adversarial safety evaluations and multilingual fairness into post-training pipelines.

**Contributions.**  This work presents a rigorous and reproducible investigation into the effects of post-training data quality on LLM performance. We evaluate two widely used model architectures, Llama-3.1-8B and SmolLM2-1.7B, across a broad suite of benchmarks, including instruction following, coding, math, and reasoning. Grounded in systematic Magpie-based annotations, our study offers the first side-by-side dissection of Tulu and SmolTalk, revealing critical differences in data quality and task composition. Leveraging these insights, we curate *TuluTalk*, a lean and high-performing dataset that outperforms both Tulu and SmolTalk on several key benchmarks. Furthermore, we demonstrate that the performance benefits of our curated dataset generalize beyond SFT, consistently translating into gains under DPO as well, underscoring the robustness of our data-centric approach across alignment methods. Our methodology combines principled annotation, quality filtering, and task-aware rebalancing, complemented by an extensive and transparent analysis in the appendix, to establish a strong and reusable foundation for future post-training research.

