# OpenReview forum: "Fixing It in Post: A Comparative Study of LLM Post-Training Data Quality and Model Performance"
_NeurIPS.cc/2025/Datasets_and_Benchmarks_Track — NeurIPS 2025 Datasets and Benchmarks Track spotlight_

### Official Review · Reviewer_DYpu · 2025-06-23

**Rating:** 5
**Confidence:** 3

**Summary:**

This paper identifies the most useful instances in two population SFT datasets (Tulu-3-SFT-Mix and SmolTalk) and creates a new SFT data set that retains most of the utility from the original two but reduces the need for SFT. The authors release the revised versions of the two data sets for others to use in post-training.

**Dataset Code Accessibility:**

Yes

**Dataset Code Comments:**

I reviewed the 3 cited data sets on huggingface. The files are available and the columns are described rather well.

I don't understand why the language variable is missing in https://huggingface.co/datasets/aladinDJ/tulutalk-annotated. The paper stated that nearly all cases are english (line 202).

**Ethical Considerations:**

No, there are no or only very minor ethics concerns

**Final Justification:**

I appreciate the authors' response. I stand by my score.

**Limitations Weaknesses:**

I would have like to see some quantification of the efficiency gains with the reduced data set sizes. Table 3 shows the performance metrics but not any sense of the savings in terms of training budget or compute requirements.

I found my self wondering what made some instances more useful for SFT than others, which the paper did not address. Perhaps this is a topic to address in future work.

Table 1 --- what are the leaderboard scores averaged over?

Table 2 -- the bolded numbers are not the highest in the row (see first row)

Figure 2 -- (a) seems to show percentages within each data set (shaded bars sum to 1, unshaded bars sum to 1). However, I can't figure out what is going on in (b) -- the bars do not sum to 1 by shading but they also do no sum to 1 within each category (ex: Math)

Typo in section 3 heading: "Annorations"

**Strengths Contributions:**

The paper makes a convincing case that we can get similar performance from models with less fine tuning. I appreciate that the authors blended instances from two different datasets --- one with predonimnately single-turn conversations (Tulu) and the other with multi-turn conversations.

---

> ### Author Rebuttal · Authors · 2025-07-31
>
> We appreciate the reviewer's recognition of our efforts to get similar performance from models with less fine tuning samples! We hope to address your points below.
>
> ## 1. Additional Results on Efficiency Gains
>
> - To assess efficiency and training cost, we report the number of processed tokens (computed with each model's distinct tokenizer), estimates for training FLOPs, and GPU hours (on an 8xA100 cluster) for SFT training of Llama and SmolLM on Tulu, SmolTalk, and TuluTalk.
> - Since the training setup and hyperparameters are fixed, this enables a direct comparison across datasets.
> - Our results are as follows:
>
> ### 1) LLama-3.1-8B:
> | Efficiency Metric | Tulu | SmolTalk | TuluTalk |
> |-------------|------|----------|----------|
> | Tokens  ($\downarrow$)    | 835M | 875M     | **708M**     |
> | Exa FLOPs ($\downarrow$)   | 40.1 | 42.0     | **34.0**     |
> | GPU Hours ($\downarrow$)   | 45   | 49       | **38**       |
>
> ### 2) SmolLM2-1.7B:
> | Efficiency Metric | Tulu | SmolTalk | TuluTalk |
> |-------------|------|----------|----------|
> | Tokens  ($\downarrow$)    | 910M | 961M     | **782M**     |
> | Exa FLOPs ($\downarrow$)   | 9.28 | 9.80     | **7.98**     |
> | GPU Hours   ($\downarrow$) | 26   | 28       | **22**       |
>
>
> - We find that the reduction in dataset size translates approximately linearly into efficiency gains.
> - For example, TuluTalk is around 14% smaller than Tulu. For Llama, this results in a proportionate reduction in the number of processed tokens (708M compared to 835M), Exa FLOPs (34 compared to 40), and total GPU hours (38 compared to 45).
> - Similar trends are observed for the SmolTalk dataset and for the SmolLM model.
> - We hope these insights adequately quantify the efficiency gains and we will include them in our final paper.
>
> ## 2. Answer to Questions
>
> ### a) Effective SFT Samples
>
> - Attributing performance gains to individual samples is difficult and part of ongoing research.
> - However, our experiments demonstrate that both quality (Input Quality/Response Reward) and task category significantly influence post-SFT performance, allowing us to identify performance-enhancing characteristics at a coarse level.
> - We consider a fine-grained analysis of sample effectiveness an important direction for future work.
>
> ### b) What are leaderboard scores averaged over?
>
> - OpenLLM Leaderboard 1 refers to the average over MMLU, TruthfulQA, ARC-C, GSM8K, HellaSwag, and WinoGrande.
> - OpenLLM Leaderboard 2 averages over MMLU\_Pro, BBH, GPQA, MuSR, and IF-Eval.
> - This follows the cited sources in [17, 18].
> - We will add some clarification in the final version of our paper.
>
> ### c) Typo in Section 3 and Table 2 -- wrong bolded numbers
>
> We thank the reviewer for pointing out the typo in the title of Section 3 and the incorrect bold formatting in Table 2. We will correct these in the final version of the paper.
>
> ### d) Clarification on Figure 2 (b)
>
> - Figure 2 (b) shows the distribution of multi-turn samples **per task category**.
> - Therefore, bars are not intended to sum to 1 as they represent the percentage of multi-turn samples per category.
> - Their complement thus represents the percentage of single-turn samples.
> - Figure 2(b) is a direct visualization of Tables 6(a) and 7(a) in Appendix C.3.2 ("Turn Type per Task Category").
> - For an alternative perspective, Figures 11 and 12 show a similar per-task-category plot, but with an explicit breakdown of single- and multi-turn proportions.
> - We will add this clarification to avoid any confusion for the readers.
>
> ### e) Comment on Missing Language Column in Uploaded Tulutalk Dataset
>
> - Thank you for catching this important detail! We acknowledge that while the annotated Tulu and Smoltalk datasets do include all language tags, Tulutalk is missing some of these entries which were not essential for curating the mixture.
> - The completion of our language analysis was performed after the main submission when preparing the supplemental materials.
> - We noticed during this period that the initially submitted TuluTalk dataset did not include all language entries.
> - However, at that time, we were prohibited from updating our dataset submission according to the NeurIPS guidelines.
> - We will merge our language labels back into TuluTalk in the final version.

---

> > ### Comment · Reviewer_DYpu · 2025-08-07
> >
> > I appreciate the authors' response. I stand by my score of 5.

---

### Official Review · Reviewer_j5jr · 2025-06-25

**Rating:** 5
**Confidence:** 3

**Summary:**

This paper presents a comprehensive analysis of two major open-source post-training datasets for large language models (LLMs): Tulu-3-SFT-Mix (Tulu) and SmolTalk. The study aims to address the opacity and lack of standardization in post-training dataset construction, which hinders reproducibility and benchmarking.

**Additional Feedback:**

The authors may consider including a small-scale human study to validate the accuracy and consistency of MagPie annotations (especially input quality and reward scores). This would provide a stronger foundation for trust in the curation pipeline and reward-based filtering.

**Dataset Code Accessibility:**

Yes

**Dataset Code Comments:**

The authors publicly release all annotations, code, and the TuluTalk dataset to foster reproducibility and further research.

**Ethical Considerations:**

No, there are no or only very minor ethics concerns

**Final Justification:**

I have no question anymore.

**Limitations Weaknesses:**

- The experiments are restricted to two model sizes—LLaMA-3.1-8B and SmolLM2-1.7B. These choices are justified for reproducibility, but results may not generalize to larger-scale models or alternative architectures.
- While MagPie annotations provide structure, they rely on LLMs as judges (e.g., LLaMA-3.3-70B-Instruct), which may themselves introduce bias or noise into the labels

**Strengths Contributions:**

+ The paper is well-written, and flows smoothly from motivation to analysis, experimentation, and conclusions.
+ This work addresses the systematic evaluation and curation of post-training datasets, a crucial and underexplored challenge in the LLM development pipeline.
+ The authors present a side-by-side comparative study of open-source post-training datasets using a standardized evaluation setup, in a stark contrast to most prior works that either analyze individual datasets in isolation or vary architectures and hyperparameters.
+ The MagPie-based annotation methodology enhances interpretability and dataset transparency

---

> ### Author Rebuttal · Authors · 2025-07-31
>
> Thank you for the positive feedback! We appreciate your recognition of the importance of systematic dataset evaluation and curation. Our responses follow below.
>
> ## 1. Experiments are restricted to two model sizes
>
> - In our initial submission, we focused on systematic comparisons between Tulu and Smoltalk to design the curation recipe that led to Tulutalk.
> - Thus, we evaluated on the two models from their respective papers.
> - Since then, we have expanded our evaluations to Qwen2.5-0B, Qwen2.5-3B, and SmolLM3-3B, covering small-/mid-scale models of different families.
> - The results are as follows:
>
>
> ### a) Qwen2.5-0.5B:
> | Model                | MMLU  | TRUTHFULQA | ARC-C | GSM8K | HELLASWAG | WINOGRANDE | MMLU_PRO | BBH   | GPQA  | MUSR  | IFEVAL | MATH | HUMAN_EVAL | HUMAN_EVAL+ | LB1   | LB2   | Avg   |
> |----------------------|-------|------------|--------|--------|------------|-------------|----------|-------|--------|--------|--------|------|--------------|----------------|--------|--------|--------|
> | Qwen2.5-0.5B-Base         | 46.51 | 39.78      | 32.22 | 34.50 | 39.93      | 56.75       | 16.98    | 31.58 | 27.20 | 34.26 | 17.34  | 4.68 | 28.66       | 26.22         | 41.62 | 22.01 | 31.19 |
> | Qwen2.5-0.5B-Tulu    | 45.67 | 39.99      | 31.83 | 37.30 | 39.10      | 58.33       | 14.14    | 30.79 | 24.33 | 32.01 | 40.60  | 6.50 | 28.66       | 25.00         | 42.04 | 24.73 | 32.45 |
> | Qwen2.5-0.5B-SmolTalk| 43.25 | 40.87      | 31.66 | 41.93 | 38.70      | 57.14       | 13.15    | 29.53 | 25.59 | 32.41 | 35.59  | 5.21 | 27.44       | 24.39         | 42.26 | 23.58 | 31.92 |
> | Qwen2.5-0.5B-**TuluTalk**| 44.65 | 39.46      | 32.59 | 41.47 | 39.18      | 57.83       | 13.89    | 29.70 | 25.59 | 32.54 | 43.24  | 5.59 | 27.44       | 24.39         | 42.53 | 25.09 | **32.68** |
>
>
> ### b) Qwen2.5-3B:
> | Model               | MMLU  | TRUTHFULQA | ARC-C | GSM8K | HELLASWAG | WINOGRANDE | MMLU_PRO | BBH   | GPQA  | MUSR  | IFEVAL | MATH  | HUMAN_EVAL | HUMAN_EVAL+ | LB1   | LB2   | Avg   |
> |---------------------|-------|------------|--------|--------|------------|-------------|----------|-------|--------|--------|--------|--------|--------------|----------------|--------|--------|--------|
> | Qwen2.5-3B-Base          | 65.55 | 48.87      | 52.90 | 70.20 | 55.51      | 71.35       | 32.12    | 46.38 | 28.27 | 43.25 | 27.13  | 15.63 | 38.41       | 32.93         | 60.73 | 32.13 | 44.89 |
> | Qwen2.5-3B-Tulu     | 66.09 | 46.85      | 52.29 | 74.21 | 56.65      | 72.06       | 32.56    | 45.20 | 26.43 | 41.14 | 63.44  | 21.00 | 44.34       | 39.12         | 61.36 | 38.29 | 48.67 |
> | Qwen2.5-3B-SmolTalk | 65.80 | 50.68      | 51.34 | 77.33 | 54.97      | 70.24       | 32.00    | 43.03 | 27.35 | 39.77 | 61.50  | 16.99 | 41.22       | 34.27         | 61.73 | 36.77 | 47.61 |
> | Qwen2.5-3B-**TuluTalk** | 65.03 | 48.76      | 52.32 | 77.10 | 56.40      | 72.09       | 31.62    | 45.88 | 28.19 | 42.83 | 66.75  | 18.67 | 42.68       | 36.85         | 61.95 | 38.99 | **48.94** |
>
>
> ### c) SmolLM3-3B:
> | Model               | MMLU  | TRUTHFULQA | ARC-C | GSM8K | HELLASWAG | WINOGRANDE | MMLU_PRO | BBH   | GPQA  | MUSR  | IFEVAL | MATH  | HUMAN_EVAL | HUMAN_EVAL+ | LB1   | LB2   | Avg   |
> |---------------------|-------|------------|--------|--------|------------|-------------|----------|-------|--------|--------|--------|--------|--------------|----------------|--------|--------|--------|
> | SmolLM3-3B-Base     | 61.37 | 45.91      | 56.06 | 67.10 | 57.24      | 72.77       | 33.55    | 45.57 | 29.70 | 41.80 | 19.52  | 16.92 | 37.20       | 29.88         | 60.08 | 31.18 | 43.90 |
> | SmolLM3-3B-Tulu     | 61.10 | 45.98      | 53.75 | 70.32 | 55.49      | 73.16       | 28.86    | 44.52 | 28.61 | 40.61 | 65.78  | 19.49 | 47.44       | 32.44         | 59.97 | 37.98 | 47.68 |
> | SmolLM3-3B-SmolTalk | 59.96 | 49.87      | 58.53 | 73.36 | 56.22      | 72.38       | 31.14    | 44.68 | 30.54 | 39.38 | 56.89  | 20.62 | 45.24       | 31.10         | 61.72 | 37.21 | 47.85 |
> | SmolLM3-3B-**TuluTalk** | 61.55 | 49.19      | 58.38 | 72.55 | 56.74      | 74.03       | 31.80    | 44.19 | 29.85 | 40.30 | 63.20  | 19.73 | 44.76       | 30.46         | 62.07 | 38.18 | **48.34** |
>
> - The results are in line with the observations in our paper and show that TuluTalk achieves better performance compared to Tulu and Smoltalk, while being a leaner dataset.
> - This demonstrates generalizability across architectures and scales.
> - While we agree that evaluating larger models like Qwen2.5-32B would be informative, our computational setup and budget unfortunately limits us from training larger models.
> - We will include these additional results and corresponding discussions in our final paper.
>
> ## 2. Small-Scale Human Study to Validate Annotations
>
> - We appreciate the reviewer’s attention to this important aspect and agree that LLM judges may potentially introduce bias/noise into the labels.
> - To further validate the accuracy of Magpie annotations, we conduct a systematic, small-scale human study to asses the quality of annotated samples.
> - To this end, we stratified 100 TuluTalk samples by task category and had four authors independently review 25 samples each for Input Quality and Response Reward.
> - We then computed exact‑match agreement and Spearman’s rank correlation $(\rho)$ between the LLM and the human consensus.
> - Our results are as follows:
>
> | Category        | Agreement (%)   | Spearman $(\rho)$ |
> | --------------- | --------------- | ---------- |
> | Input Quality   | **91%**        | **0.85**   |
> | Response Reward | **93%**        | **0.87**   |
>
> - Both Input Quality and Response Reward show high categorical agreement (≥90%) and strong positive rank correlation, indicating that the LLM’s scores are closely aligned with human judgments.
> - We observe that disagreements typically involved one-step differences such as rating input quality as excellent versus good, which may be attributed to subjective interpretation.
> - These results confirm that a capable judge model reliably mirrors expert annotations, especially when the task is fairly coarse‑grained as it is in our case.
> - In the final version, we will expand upon these results and will add the details of the manual inspection.
>
> In addition to this manual audit, we provided a comparative study between Llama- and Qwen-based judges in Appendix C.1.3 (“Choice of Judge Model“).
> - Therein, we investigated labels for Input Quality on a stratified subset of 30k samples for Tulu and SmolTalk and found that the Llama judge produced a more balanced distribution of annotations (see Figures 7a and 7b in our paper).
> - In our final version, we will provide additional concrete examples.
> - This further validates our choice of judge model.
>
> We would also like to mention that the empirical performance improvements in our experiments further corroborate the validation of the Magpie pipeline.
>
> Generally, we agree that avoiding bias in LLM-as-a-judge frameworks is difficult. Developing techniques to reduce the noise, e.g., by employing an ensemble of multiple judge models, is an interesting future work.

---

### Official Review · Reviewer_U6E3 · 2025-07-02

**Rating:** 5
**Confidence:** 3

**Summary:**

The paper analyzes two commonly used post-training dataset, Tulu-3-SFT-Mix and SmolTalk, and discusses their similarities, composition and differences through a thorough annotation process with the MagPie pipeline. Tulu mainly contains single-turn conversations focused on math, coding and science, while the SmolTalk distribution is spread around conversational tasks like editing, brainstorming, planning etc and contains several-turn conversations. Through some ablation studies involving number of high quality samples, impact of multi-turn dialogues, and mixing between the two datasets, the authors find that a smaller and curated mixture dataset can outperform both larger datasets on several benchmarks.

**Additional Feedback:**

Do you think other datasets can be curated and "mixed" to produce an even higher quality one? For example by integrating TuluTalk with data that are currently not present there but can be in other SFT datasets.

**Dataset Code Accessibility:**

Yes

**Dataset Code Comments:**

The annotated TuluTalk, Tulu and SmolTalk datasets have been released on Huggingface (the link to Tulu and SmolTalk are inside the TuluTalk page) and include several annotations like category, conversation depth, quality etc.

Additionally, the code to perform the annotation has been released in a Github repository with the extension to the Magpie pipeline (template enforcement and multi-turn conversation support).

**Ethical Considerations:**

No, there are no or only very minor ethics concerns

**Final Justification:**

Thank you for thoroughly clarifying the annotation process and for expanding the experiments with different models, this is a great sign of generalization. I think the clarification on the annotation process and the new experiments make the paper stronger overall, so I increased my score to 5.

**Limitations Weaknesses:**

- The paper’s core analysis heavily relies on MagPie and it would be very helpful to have a brief discussion on the validation of the pipeline. For example:
    - Did you also perform a manual inspection on a small subset of data to assess the annotation quality of the MagPie judge?
    - How did you check that 97% of the samples were reliably tagged? Does this refer only to parser success or also measures the label quality?
- The final results on the TuluTalk dataset are interesting. However, it would be valuable to see some results on diverse models and scales, to help understanding scaling and generalization (even if those models were the ones used in the Tulu and SmolTalk papers)
- A minor point: in table 2, in a few entries, the bolded value does not correspond to the highest in the row (like row 1 for Llama-3.1-8B and last row for SmolLM-1.7B). This might be a typo or otherwise some clarifications would be helpful
- Another minor point: since the MagPie pipeline is an important tool in the analysis, it’d be helpful to have a brief overview of it also in the main body of the paper.
- Small typo in the title of section 3 (Annorations -> Annotations)

**Strengths Contributions:**

- The analysis of the two datasets is comprehensive, including category distribution, conversational structure, multi-turn category distribution etc
- Showing that a smaller but higher quality mixture dataset can result in superior performance, can have a significant impact in saving training time and computational resources
- Releasing annotated datasets is a good contribution to the community
- The paper is well written and the tables and plots are helpful in showing the results and illustrating the different characteristics of the Tulu and SmolTalk datasets

---

> ### Author Rebuttal · Authors · 2025-07-31
>
> Thank you for your time and feedback! We are pleased that you recognize our comprehensive analysis and careful dataset annotation as strong contributions to the community. We hope to address your suggestions below.
>
> ## 1. Manual Inspection of Annotation Quality
>
> - At the time of submission, we spot checked only a handful of samples to assure annotation quality.
> - We agree that performing a systematic manual inspection of a small subset of annotated samples would be beneficial.
> - To this end, we stratified 100 TuluTalk samples by task category and had four authors independently review 25 samples each for Input Quality and Response Reward.
> - We then computed exact‑match agreement and Spearman’s rank correlation $(\rho)$ between the LLM and the human consensus.
> - Our results are as follows:
>
> | Category        | Agreement (%)   | Spearman $(\rho)$ |
> | --------------- | --------------- | ---------- |
> | Input Quality   | **91%**        | **0.85**   |
> | Response Reward | **93%**        | **0.87**   |
>
> - Both Input Quality and Response Reward show high categorical agreement (≥90%) and strong positive rank correlation, indicating that the LLM’s scores are closely aligned with human judgments.
> - We observe that disagreements typically involved one-step differences such as rating input quality as excellent versus good, which may be attributed to subjective interpretation.
> - These results confirm that a capable judge model reliably mirrors expert annotations, especially when the task is fairly coarse‑grained as it is in our case.
> - In the final version, we will expand upon these results and will add the details of the manual inspection.
>
> In addition to this manual audit, we provided a comparative study between Llama- and Qwen-based judges in Appendix C.1.3 (“Choice of Judge Model“).
> - Therein, we investigated labels for Input Quality on a stratified subset of 30k samples for Tulu and SmolTalk and found that the Llama judge produced a more balanced distribution of annotations (see Figures 7a and 7b in our paper).
> - In our final version, we will provide additional concrete examples.
> - This further validates our choice of judge model.
>
> We would also like to mention that the empirical performance improvements in our experiments further corroborate the validation of the Magpie pipeline.
>
> Generally, we agree that avoiding bias in LLM-as-a-judge frameworks is difficult. Developing techniques to reduce the noise, e.g., by employing an ensemble of multiple judge models, is an interesting future work.
>
> ### Clarification of 97% Tagging Rate
>
> - Indeed, this refers to parser success, i.e., only 3% of the samples could not be tagged due to inconsistent fromatting or other errors not caught by our additional measures.
> - We discuss these issues in Appendix C.1.2 ("Reliable and Error Tolerant Prompts") and will clarify the details in the final paper.
>
> ## 2. Additional Results for Diverse Models and Scales
>
> - In our initial submission, we focused on systematic comparisons between Tulu and Smoltalk to design the curation recipe that led to Tulutalk.
> - Thus, we evaluated on the two models from their respective papers.
> - Since then, we have expanded our evaluations to Qwen2.5-0B, Qwen2.5-3B, and SmolLM3-3B, covering small-/mid-scale models of different families.
> - The results are as follows:
>
>
> ### a) Qwen2.5-0.5B:
> | Model                | MMLU  | TRUTHFULQA | ARC-C | GSM8K | HELLASWAG | WINOGRANDE | MMLU_PRO | BBH   | GPQA  | MUSR  | IFEVAL | MATH | HUMAN_EVAL | HUMAN_EVAL+ | LB1   | LB2   | Avg   |
> |----------------------|-------|------------|--------|--------|------------|-------------|----------|-------|--------|--------|--------|------|--------------|----------------|--------|--------|--------|
> | Qwen2.5-0.5B-Base         | 46.51 | 39.78      | 32.22 | 34.50 | 39.93      | 56.75       | 16.98    | 31.58 | 27.20 | 34.26 | 17.34  | 4.68 | 28.66       | 26.22         | 41.62 | 22.01 | 31.19 |
> | Qwen2.5-0.5B-Tulu    | 45.67 | 39.99      | 31.83 | 37.30 | 39.10      | 58.33       | 14.14    | 30.79 | 24.33 | 32.01 | 40.60  | 6.50 | 28.66       | 25.00         | 42.04 | 24.73 | 32.45 |
> | Qwen2.5-0.5B-SmolTalk| 43.25 | 40.87      | 31.66 | 41.93 | 38.70      | 57.14       | 13.15    | 29.53 | 25.59 | 32.41 | 35.59  | 5.21 | 27.44       | 24.39         | 42.26 | 23.58 | 31.92 |
> | Qwen2.5-0.5B-**TuluTalk**| 44.65 | 39.46      | 32.59 | 41.47 | 39.18      | 57.83       | 13.89    | 29.70 | 25.59 | 32.54 | 43.24  | 5.59 | 27.44       | 24.39         | 42.53 | 25.09 | **32.68** |
>
>
> ### b) Qwen2.5-3B:
> | Model               | MMLU  | TRUTHFULQA | ARC-C | GSM8K | HELLASWAG | WINOGRANDE | MMLU_PRO | BBH   | GPQA  | MUSR  | IFEVAL | MATH  | HUMAN_EVAL | HUMAN_EVAL+ | LB1   | LB2   | Avg   |
> |---------------------|-------|------------|--------|--------|------------|-------------|----------|-------|--------|--------|--------|--------|--------------|----------------|--------|--------|--------|
> | Qwen2.5-3B-Base          | 65.55 | 48.87      | 52.90 | 70.20 | 55.51      | 71.35       | 32.12    | 46.38 | 28.27 | 43.25 | 27.13  | 15.63 | 38.41       | 32.93         | 60.73 | 32.13 | 44.89 |
> | Qwen2.5-3B-Tulu     | 66.09 | 46.85      | 52.29 | 74.21 | 56.65      | 72.06       | 32.56    | 45.20 | 26.43 | 41.14 | 63.44  | 21.00 | 44.34       | 39.12         | 61.36 | 38.29 | 48.67 |
> | Qwen2.5-3B-SmolTalk | 65.80 | 50.68      | 51.34 | 77.33 | 54.97      | 70.24       | 32.00    | 43.03 | 27.35 | 39.77 | 61.50  | 16.99 | 41.22       | 34.27         | 61.73 | 36.77 | 47.61 |
> | Qwen2.5-3B-**TuluTalk** | 65.03 | 48.76      | 52.32 | 77.10 | 56.40      | 72.09       | 31.62    | 45.88 | 28.19 | 42.83 | 66.75  | 18.67 | 42.68       | 36.85         | 61.95 | 38.99 | **48.94** |
>
>
> ### c) SmolLM3-3B:
> | Model               | MMLU  | TRUTHFULQA | ARC-C | GSM8K | HELLASWAG | WINOGRANDE | MMLU_PRO | BBH   | GPQA  | MUSR  | IFEVAL | MATH  | HUMAN_EVAL | HUMAN_EVAL+ | LB1   | LB2   | Avg   |
> |---------------------|-------|------------|--------|--------|------------|-------------|----------|-------|--------|--------|--------|--------|--------------|----------------|--------|--------|--------|
> | SmolLM3-3B-Base     | 61.37 | 45.91      | 56.06 | 67.10 | 57.24      | 72.77       | 33.55    | 45.57 | 29.70 | 41.80 | 19.52  | 16.92 | 37.20       | 29.88         | 60.08 | 31.18 | 43.90 |
> | SmolLM3-3B-Tulu     | 61.10 | 45.98      | 53.75 | 70.32 | 55.49      | 73.16       | 28.86    | 44.52 | 28.61 | 40.61 | 65.78  | 19.49 | 47.44       | 32.44         | 59.97 | 37.98 | 47.68 |
> | SmolLM3-3B-SmolTalk | 59.96 | 49.87      | 58.53 | 73.36 | 56.22      | 72.38       | 31.14    | 44.68 | 30.54 | 39.38 | 56.89  | 20.62 | 45.24       | 31.10         | 61.72 | 37.21 | 47.85 |
> | SmolLM3-3B-**TuluTalk** | 61.55 | 49.19      | 58.38 | 72.55 | 56.74      | 74.03       | 31.80    | 44.19 | 29.85 | 40.30 | 63.20  | 19.73 | 44.76       | 30.46         | 62.07 | 38.18 | **48.34** |
>
> - The results are in line with the observations in our paper and show that TuluTalk achieves better performance compared to Tulu and Smoltalk, while being a leaner dataset.
> - This demonstrates generalizability across architectures and scales.
> - While we agree that evaluating larger models like Qwen2.5-32B would be informative, our computational setup and budget unfortunately limits us from training larger models.
> - We will include these additional results and corresponding discussions in our final paper.
>
> ## 3. Minor Points
>
> ### a) Typos and Tables
>
> We thank the reviewer for pointing out the typo in the title of Section 3 and the incorrect bold formatting in Table 2. We will correct these in the final version of the paper.
>
> ### b) Brief Overview of Magpie in the Main Body
>
> In the final paper, we will introduce Magpie at the beginning of Section 3 as a self-synthesis annotation pipeline and will briefly outline its main components.
>
> ### c) Can other datasets be curated and "mixed" to produce an even higher quality one?
>
> - Yes, other datasets can be effectively curated as our curation recipe is general and model-agnostic, and can thus be applied to any SFT datasets in principle.
> - Applying our curation recipe on other/new datasets, e.g., SmolTalk-v2, and analyzing the performance of resulting mixtures is an interesting future work.
> - You are more than welcome to use our recipe with a different dataset to train and study other models.

---

### Comment · Area_Chair_Sh4A · 2025-08-05
**Mandatory Acknowledgement**

NeurIPS 2025 Datasets and Benchmarks Track Submission1990 Reviewer U6E3

Pls help submit your Mandatory Acknowledgement asap and provide your feedback!

---

### Note · Authors · 2025-08-14

**We sincerely thank the reviewers for the careful review of our submission and provide a brief summary to facilitate discussion with the AC!**

In our work, we
* conducted a comprehensive analysis of two prominent large-scale open-source SFT datasets (Tulu3, SmolTalk)
* performed data annotations for input quality, difficulty, response reward, turn structure, task category, safety, and others
* designed a data curation recipe that produces **TuluTalk - an up to 23% smaller mix that outperforms on key benchmarks**
* fully open-sourced all annotated datasets

We especially want to thank the reviewers for recognizing our strong contributions:
* "*_The paper makes a convincing case that we can get similar performance with less fine tuning_*"
* "*_Releasing annotated datasets is a good contribution to the community_*"
* "*_Systematic evaluation and curation of post-training datasets [is] a crucial and underexplored challenge in LLM development_*"
* "*_A side-by-side comparative study [is] in stark contrast to most prior works that analyze datasets in isolation or vary architectures/hyperparameters_*"

The reviewers' comments have helped us fix minor typos and improve the paper during rebuttal by adding
* a small-scale systematic human study, showing **our annotations positively correlate with human judgement**
* more results for diverse models and scales (Qwen2.5-0.5B, Qwen2.5-3B, SmolLM3-3B), showing **our recipe generalizes**
* insights on efficiency, showing a reduction in data translates linearly into a **reduction of GPU hours and FLOPs**

We believe the rebuttal has strengthened our contributions and we hope to have fully addressed the reviewers' questions, retaining or even **improving the strong scores** with these additional results!

Thank you again!

---

### Decision · Program_Chairs · 2025-09-18

**Decision:**

Accept (spotlight)

**Comment:**

Overall the reviewers are quite positive. They like the comprehensive comparison between Tulu and SmolTalk, and the new curated Tulutalk dataset that’s smaller but still strong. People highlight the release of annotated data as a real plus for the community.

A few minor issues were pointed out (typos, table bolding errors, brief explain of MagPie pipeline, efficiency quant numbers), but nothing blocking. Authors already addressed most of them with extra exps and even a small human validation study.

Main strenght is the clear message: better quality > just more data, and this work gives practical recipes for future SFT dataset design. Weakness is only that eval is limited to certain model sizes, but added results help a lot.

Given the clarity, novelty, and potential impact, I’d say it’s above accept level, fits well as spotlight.